# Tertiary lymphoid structures and B cells determine clinically relevant T cell phenotypes in ovarian cancer

Intratumoral tertiary lymphoid structures (TLSs) have been associated with improved outcome in various cohorts of patients with cancer, reflecting their contribution to the development of tumor-targeting immunity. Here, we demonstrate that high-grade serous ovarian carcinoma (HGSOC) contains distinct immune aggregates with varying degrees of organization and maturation. Specifically, mature TLSs (mTLS) as forming only in 16% of HGSOCs with relatively elevated tumor mutational burden (TMB) are associated with an increased intratumoral density of CD8[+] effector T ($T_{EFF}$) cells and TIM3[+]PD1[+], hence poorly immune checkpoint inhibitor (ICI)-sensitive, CD8[+] T cells. Conversely, CD8[+] T cells from immunologically hot tumors like non-small cell lung carcinoma (NSCLC) are enriched in ICI-responsive TCF1[+] PD1[+] T cells. Spatial B-cell profiling identifies patterns of in situ maturation and differentiation associated with mTLSs. Moreover, B-cell depletion promotes signs of a dysfunctional CD8[+] T cell compartment among tumor-infiltrating lymphocytes from freshly isolated HGSOC and NSCLC biopsies. Taken together, our data demonstrate that – at odds with NSCLC – HGSOC is associated with a low density of follicular helper T cells and thus develops a limited number of mTLS that might be insufficient to preserve a ICI-sensitive TCF1[+]PD1[+] CD8[+] T cell phenotype. These findings point to key quantitative and qualitative differences between mTLSs in ICI-responsive vs ICI-irresponsive neoplasms that may guide the development of alternative immunotherapies for patients with HGSOC.

It is now widely accepted that most – if not all – tumors originate and develop into clinically manifest entities by evading immunosurveillance[1–3]. Consistent with this notion, a variety of clinically successful regimens including numerous chemotherapeutics[4], radiation therapy[5,6] and targeted anticancer agents[7] have been shown to mediate therapeutically relevant immunostimulatory effects[8]. Moreover, several indicators of the immunological tumor contexture have been shown to influence disease outcome in hundreds of cohorts of patients with cancer[9–11]. These parameters encompass not only the quantity and quality of tumor-infiltrating immune cells, their spatial organization, and their mutual interactions, but also the propensity of malignant cells to be recognized by the immune system, for instance owing to an elevated tumor mutational burden (TMB)[9,10].

While tumor-draining lymph nodes play a major role in the initiation of tumor-targeting immunity[12], many solid neoplasms contain so-called "tertiary lymphoid structures" (TLSs), which in their mature form are spatially organized clusters of CD8[+] effector T ($T_{EFF}$) cells, B lymphocytes, and CD21[+]CD23[+] follicular dendritic cells (DCs), generally served by high-endothelial venules (HEVs)[13–15]. Mature TLSs (mTLSs) represent privileged sites for local antigen presentation by

✉ e-mail: fucikova@sotio.com

DCs, contributing to the generation of tumor-targeting CD8[+] effector memory T (T$_{EM}$) cells as well as memory B cells and antibody-producing plasma cells, both of which originate from structurally defined germinal centers (GCs)[15,16]. Supporting a key relevance for TLSs in natural and immunotherapy-driven immunosurveillance, intratumoral TLS density has been associated with improved disease outcome in several cohorts of patients with solid tumors[16], including (but not limited to) melanoma[17], non-small cell lung carcinoma (NSCLC)[18], colorectal carcinoma[19] and breast carcinoma[20].

Immune checkpoint inhibitors (ICIs) have literally revolutionized the clinical management of various oncological indications, including several settings in which a high prevalence of TLSs at baseline has been associated with improved disease outcome[21–26]. Along similar lines, ICIs appear to be particularly active in patients bearing tumors with genetic, functional and/or immunological features that a priori are supportive of anticancer immunity[27]. Such features include not only an abundant recruitment of immune effector cells that persist in an ICI-activatable state in the tumor microenvironment[28], but also the propensity of neoplastic cells to be recognized and eliminated by the immune system[29]. Accordingly, genomic instability (especially in the form of microsatellite instability) and the consequent accrual of non-synonymous DNA mutations (which increase the likelihood of malignant cells to present novel antigenic determinants) have also been associated with superior ICI sensitivity across several tumor types[30,31]. Similar findings have been obtained for the expression levels of CD274 (best known as PD-L1), an immunosuppressive ligand expressed by malignant and myeloid cells in response to ongoing anticancer immunity[32]. At odds with melanoma, renal cell carcinoma and NSCLC, epithelial ovarian carcinoma (EOC) is poorly sensitive to ICIs employed as standalone immunotherapeutic agents, most likely due to a relatively low TMB coupled with indolent anticancer immunity and active immunosuppression[33–36]. Although the clinical and biological relevance of TLSs[37], B cells[38,39], plasma cells[40] and humoral adaptive immunity[41,42] for patients with EOC have been previously reported, the precise immune contexture of TLSs developing in the EOC microenvironment, their impact on the phenotypic profile of intratumoral T cells, and their influence on sensitivity to immunotherapy remain to be investigated in detail.

Here, we harness spatial transcriptomics, immunofluorescence microscopy, and flow cytometry to characterize TLSs in patients with high-grade serous ovarian carcinoma (HGSOC). Our data demonstrate that mTLSs are forming in a limited number of HGSOCs with a relatively elevated TMB (considering that HGSOC has a low-to-intermediate TMB as compared to other tumors) and are associated with an increased intratumoral density of CD8[+] effector T (T$_{EFF}$) cells. Conversely, an ICI-resistant TIM3[+]PD1[+] phenotype as supported by less mature TLSs is prevalent in HGSOC samples. Moreover, NSCLCs develop a significantly higher frequency of mTLSs, which are predominantly associated with ICI-sensitive TCF1[+]PD1[+]CD8[+] T$_{EFF}$ cells within mTLS areas and the entire tumor microenvironment (TME). Taken together, our data delineate key numerical and functional differences between mTLSs in ICI-responsive vs ICI-irresponsive tumors that may inspire the development of alternative immunotherapies for HGSOC.

## Results

### HGSOC contains TLSs at different maturation states

Tumor samples from a retrospective series of 209 patients with HGSOC (Study groups 1, 2 and 3 from 2 independent cohorts; Supplementary Tables 1, 2 and Supplementary Fig. 1) who did not receive neoadjuvant chemotherapy were analyzed for early TLS (eTLS) and mature TLS (mTLS) using immunofluorescence microscopy based on CD4, CD8, CD20, CD21, CD23, DC-LAMP and GZMB (Fig. 1A, Supplementary Fig. 2A-C). TLS with primary (CD21[+] follicular dendritic cells [DCs]) and secondary (CD21[+]CD23[+] follicular

DCs) follicles were defined as mTLSs, using criteria previously employed for lung carcinoma, colorectal carcinoma and melanoma (Fig. 1A, Supplementary Fig. 2B, C)[43–45]. Both eTLS and mTLSs were predominantly localized at invasive margins or in the tumor stroma as compared to the tumor core (Supplementary Fig. 3A, B). eTLSs were detected in 122/209 tumors (58%) and mTLSs in 33/209 (16%) (Fig. 1B). There was significant variability in the relative amount of eTLSs and mTLSs across HGSOC samples (Fig. 1B), independent of the patient cohort. eTLSs were more abundant in early (Stage I + II) vs advanced (Stage III + IV) HGSOC (Fig. 1C). Conversely, we found no differences in the abundance of mTLSs across disease stage (Fig. 1C).

To further characterize the architecture of TLSs at distinct maturation states, we harnessed spatial transcriptomics coupled with deconvolution analysis based on "metagene" markers that estimate cell type abundance[46]. Although genetic signatures of T cells, cytotoxic T cells, T$_{H}$1 cells, T follicular helper (T$_{FH}$) cells, activated and naive B cells were more abundant in the proximity of mTLSs (Fig. 1D), signatures of monocytes, DCs, macrophages, natural killer (NK) cells, memory B cells and T$_{H}$2 cells were diffusely distributed throughout tumor sections (Fig. 1E). These findings were consistent across 3 different samples containing TLSs at various maturation stages (Supplementary Fig. 3C, D), corroborating the association between TLSs and B cells, T$_{H}$1 cells, and effector CD8[+] T cells[24].

To confirm our findings with an independent technology, we determined the density of key TLS immune cell populations, including mature DC-LAMP[+] DCs, CD20[+] B cells, CD8[+] T cells and effector GZMB[+]CD8[+] T cells within individual eTLSs and mTLSs, as well as non-TLS (nTLS) areas, in the entire tumor tissue of 68 patients with HGSOC (Study cohort 1) using immunofluorescence microscopy (Fig. 1A; Supplementary Fig. 2C). In line with transcriptomic data, CD20[+] B cells, DC-LAMP[+] DCs, CD8[+] T cells and GZMB[+]CD8[+] T cells were largely localized within individual TLSs as compared to nTLS areas (Fig. 1F). Moreover, DC-LAMP[+] DCs, CD8[+] and GZMB[+]CD8[+] T cells were equally distributed within eTLSs and mTLSs (Fig. 1F), which is in line with the key roles of DC-LAMP[+] DCs and CD8[+] T cells in TLS formation and effector functions, respectively[16,18].

To elucidate the prognostic value of immune cells within TLSs at different maturation states or nTLS areas, we evaluated the relationship between patient overall survival (OS) and the density of CD20[+] B cells, DC-LAMP[+] DCs, CD8[+] T cells and GZMB[+]CD8[+] T cells in the HGSOC microenvironment. Importantly, a high density of CD20[+] B cells and CD8[+] T cells in nTLS areas was invariably associated with improved disease outcome, both when densities were analyzed as continuous variable by univariate Cox regression analysis (Fig. 1G) and when patients were stratified by median values (Fig. 1H). The same did not hold true for DC-LAMP[+] DCs in nTLS areas (Supplementary Fig. 4A) nor for CD20[+] B cells, DC-LAMP[+] DCs, CD8[+] T cells and GZMB[+]CD8[+] T cells within eTLSs and mTLS, with the sole exception of GZMB[+]CD8[+] T cells in mTLSs being associated with non-significant trend towards improved disease outcome (Fig. 1G).

Altogether, these findings suggest that although TLSs represent a unique site in the TME of HGSOC exhibiting a dense accumulation of CD20[+] B cells, T$_{FH}$ cells and effector cells, improved disease outcome is largely associated with immune cell infiltration in nTLS tumor areas.

### mTLS formation is associated with effector CD8[+] T cells and development of antitumor immunity

Inspired by our findings spatially linking TLSs to immune effector cells within the HGSOC microenvironment, we set to harness RNA sequencing (RNAseq) to compare the gene expression profile of 53 tumor samples (Study cohort 1; Methods) containing no TLSs within TME (Cluster 1 [CL1]), eTLSs only (Cluster 2 [CL2]) and eTLSs plus mTLSs (Cluster 3 [CL3]) (based on immunofluorescence microscopy assessments) (Fig. 2A). Differential gene expression analysis

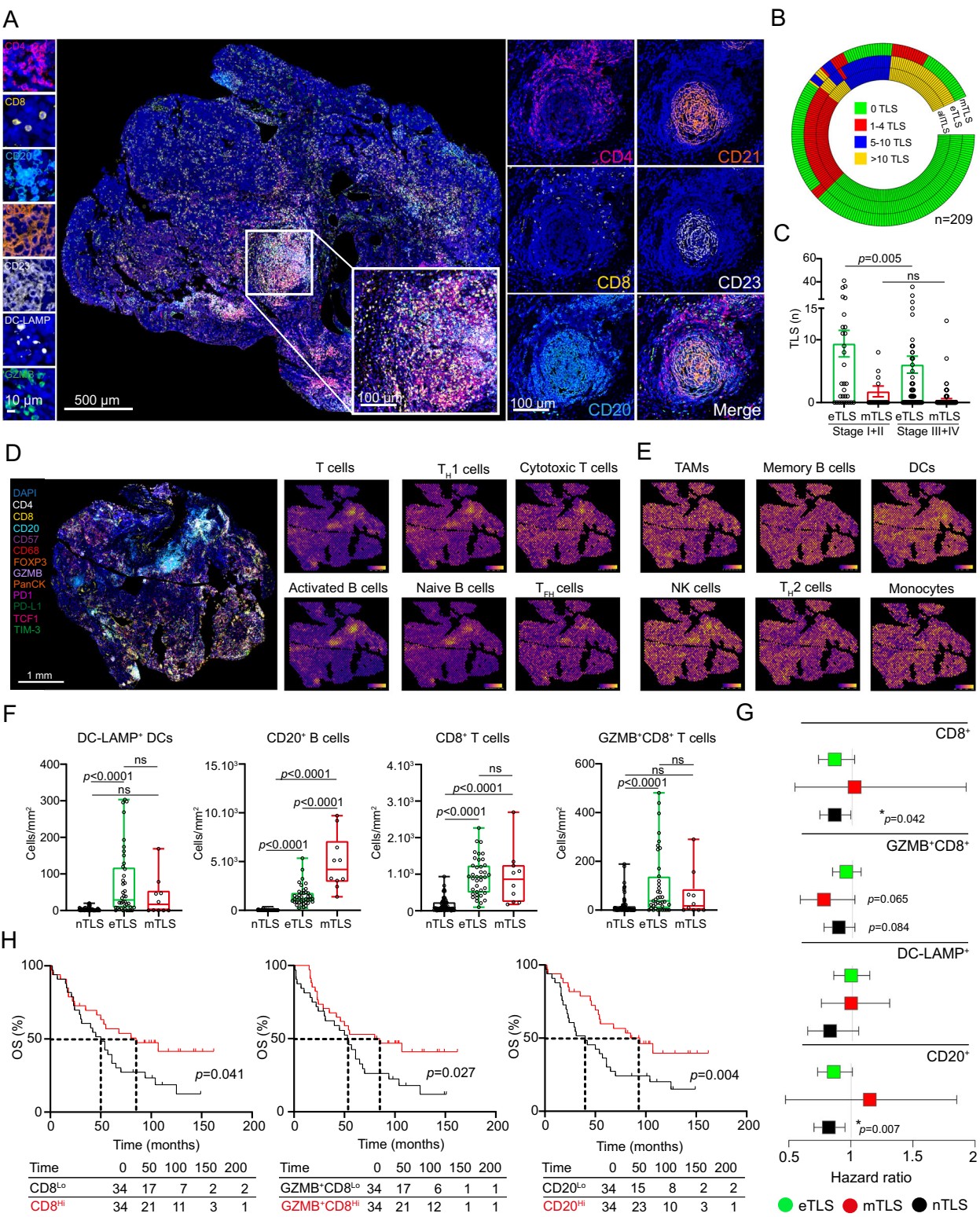

followed by pathway enrichment based on Gene Ontology (GO) terms mainly identified gene sets linked to the immune response, B cells, T cells activation and cytotoxicity as over-represented in tumor samples of patients from cluster 2 and 3 as compared to cluster 1 (Fig. 2A; Supplementary Fig. 4B, C; Supplementary Data 1). Next, we employed "metagene" markers (as per the MCP counter method)[47] to estimate the relative abundance of different immune cell populations in individual clusters (Fig. 2B). In line with whole-transcriptome findings (Fig. 2A), Cluster 2 and 3 samples were

enriched for gene sets associated with B cells, T cells and cytotoxicity as compared to Cluster 1 (Fig. 2B). Moreover Cluster 3 was enriched in transcripts encoding co-inhibitory molecules such programmed cell death 1 (PDCD1; best known as PD1), hepatitis A virus cellular receptor 2 (HAVCR2; best known as TIM3) and cytotoxic T-lymphocyte associated protein 4 (CTLA4) as compared to other clusters (Fig. 2B). To corroborate our findings in a large independent cohort of patients with HGSOC, we collected transcriptomic data from 304 patients included in the TCGA public database (Study

**Fig. 1 | The clinical relevance of spatial immune composition in distinct maturation types of tertiary lymphoid structures (TLS) in HGSOC.**
**A** Representative image of immunofluorescence of CD4, CD8, CD20, CD21, CD23, DC-LAMP and GZMB staining (*immunofluorescence panel 1*). Scale bars 10, 100 and 500 μm. **B** Distribution of early TLS (eTLS) and mature TLS (mTLS) across 209 HGSOC patients (Study cohort 1 and 2; Supplementary Table 1) and **C** across pathologic disease stage of 123 HGSOC patients. (Study cohort 1; Stage I + II: *n* = 34; Stage III + IV: *n* = 89; Supplementary Table 1). Mean and SEM are shown. Statistical significance was calculated by two-sided Mann-Whitney test. *p* values are indicated. ns, not significant. Spatial co-localization of TLS with gene signatures of (**D**) T cells, T$_H$1 cells, cytotoxic T cells, activated and naive B cells, T follicular cells (T$_{FH}$) cells, (**E**) tumor associated macrophages (TAMs), memory B cells, dendritic cells (DCs), natural killer (NK) cells, T$_H$2 cell and monocytes in tumor sample from Study Cohort 1, determined by Visium transcriptomic (repeated in 3 independent HGSOC samples). Immunofluorescence staining for CD4, CD8, CD20, CD57, CD68, FoxP3, GZMB, PanCK, PD1, PD-L1, TCF1 and TIM3 (*immunofluorescence panel 2*) on FFPE TLS$^+$ tumor used for the spatial transcriptomic assay delineate pathologically identified TLS areas. **F** Density of DC-LAMP$^+$ DCs, CD20$^+$ B cells, CD8$^+$ T cells and GZMB$^+$CD8$^+$ T cells in non-TLS areas (nTLS), eTLS and mTLS within 68 tumor samples (Study cohort 1, group 1), as determined by immunostaining (*immunofluorescence panel 1*). Box plots: lower quartile, median, upper quartile; whiskers, minimum, maximum. Statistical significance was calculated by two-sided the Mann–Whitney test. *p* values are indicated. **G** Forest plot displaying univariate Cox analyses of density of CD8$^+$ T cells, GZMB$^+$CD8$^+$ T cells, DC-LAMP$^+$ DCs and CD20$^+$ B cells in non-TLS areas (nTLS), eTLS and mTLS areas of 68 tumor samples (Study cohort 1, group 1). All hazard ratios are obtained from Cox proportional hazard models with adjustment for randomised group only. Lower quartile, mean, upper quartile are shown. **H** Overall survival (OS) of 68 patients (Study cohort 1, group 1) based on median stratification of CD8$^+$ T cells, GZMB$^+$CD8$^+$ T cells and CD20$^+$ B cells density in nTLS areas of tumor samples. Survival curves were estimated by the Kaplan-Meier method, and differences between groups were evaluated using log-rank test. Number of patients at risk and *p* values are reported. Source data are provided as a Source Data file.

cohort 4, Supplementary Fig. 1). Using unsupervised hierarchical clustering based on the aforementioned "metagene" markers[46], we identified three clusters of patients (Fig. 2C). In line with data from Study cohort 1 (Fig. 2B), Cluster 3 was enriched for gene signatures associated with T cells, B cells, cytotoxicity and a TLS-relevant chemokine signature (*CCL2, CCL3, CCL4, CCL5, CCL8, CCL18, CCL19, CCL21, CXCL9, CXCL10, CXCL11, and CXCL13*)[48,49] as compared to Cluster 1 and 2 (Supplementary Fig. 4D). Moreover, Cluster 3 with overexpression of these TLS-related chemokines was also enriched in numerous transcripts that code for co-inhibitory molecules including PD1, TIM3, CTLA4 and others (Fig. 2C; Supplementary Fig. 4D).

To validate these data with another technological approach, we analyzed the immune infiltrate of tumor samples from patients (Study cohort 1, Supplementary Fig. 1) in Clusters 1, 2 and 3 by IHC and immunofluorescence microscopy (Fig. 2D). Confirming transcriptomic observations, we detected a higher density of CD8$^+$ T cells, tumor core CD8$^+$ T cells and GZMB$^+$CD8$^+$ T cells, as well as CD20$^+$ B cells, DC-LAMP$^+$ DCs, NKp46$^+$ NK cells and CD3$^+$FoxP3$^+$ regulatory T (T$_{REG}$) cells in samples from Cluster 3 patients as compared to Cluster 1 (Fig. 2E, F; Supplementary Fig. 4E). Likewise, HGSOC samples from Cluster 3 contained the highest density of PD1$^+$, CTLA-4$^+$ and TIM3$^+$ cells (Fig. 2E).

Next, we set out to determine the prognostic value of TLSs in 209 patients with HGSOC from 2 independent cohorts (Study groups 1, 2 and 3; Supplementary Table 1). First, we evaluated relapse-free survival (RFS) and OS upon stratifying patients based on median number of TLSs, finding that patients with higher-than-median TLSs (TLS$^{Hi}$) had prolonged RFS (*p* = 0.002) and OS (*p* = 0.022) as compared to their TLS$^{Lo}$ counterparts (Fig. 2G). Next, we evaluated RFS and OS upon stratifying 209 patients based on TLS abundance as above: no TLSs (Cluster 1), eTLSs only (Cluster 2) and eTLSs+mTLSs (Cluster 3) as determined by immunofluorescence (see also Fig. 2A). We found that Cluster 2 (patients with eTLSs only) had significantly improved RFS and OS as compared to Cluster 1 (patients with no TLSs; RFS: *p* = 0.0002; OS: *p* = 0.001) and Cluster 3 (patients with eTLSs and mTLSs; RFS: *p* = 0.035; OS: *p* = 0.006) (Fig. 2H). These data were confirmed by univariate and multivariate Cox regression analyses (Table 1; Supplementary Table 3, 4). Similar findings were also obtained when we analyzed early and late stage HGSOC (Supplementary Fig. 5A, B; Supplementary Tables 3, 4) and HGSOC patients cohorts independently from each other (Supplementary Fig. 5C–E), and were further validated in 40 HGSOC patients from validation Study cohort 3 (Supplementary Fig. 5F; Supplementary Table 5).

Taken together, these data suggest that while eTLSs and mTLSs are associated with increased amounts of immune effector cells in HGSOC samples, terminal TLS maturation is associated with poor disease outcome as compared to a less mature TLS state.

## Signs of dysfunctional CD8$^+$ T cells are abundant in HGSOC (but not NSCLC) samples with mature TLSs

As tumor samples from HGSOC patients with mTLSs are associated with increased expression of co-inhibitory receptors (Fig. 2A–C), we quantified the levels of *PDCD1, HAVCR2*, and transcription factor 7 (*TCF7*; encoding a marker of progenitor T cells best known as TCF1) within the HGSOC microenvironment using spatial transcriptomics upon TLS localization on three H&E-stained consecutive FFPE sections from study cohort 1 (Fig. 3A). *PDCD1* and *TCF7* were overrepresented in mTLSs as compared to eTLSs and nTLS areas in 3 independent tumor samples, with a similar but subsignificant trend for *HAVRC2* (Supplementary Fig. 6A).

Next, we analyzed the density of TCF1$^+$PD1$^+$CD8$^+$ T cells, TIM3$^+$PD1$^+$CD8$^+$ T cells and TIM3$^+$PD1$^-$CD8$^+$ T cells in the entire TME, eTLS and mTLS areas using immunofluorescence microscopy alongside spatial analyses in 19 HGSOC samples from Study cohort 1 (Fig. 3B). In line with our transcriptomic data, PD1$^+$CD8$^+$ T cells taken as a whole were more abundant in TLSs (cells/mm$^2$ of TLS area) as compared to the entire TME (including TLSs, cells/mm$^2$ of entire TME area) (Fig. 3C). Moreover, both TIM3$^+$PD1$^+$CD8$^+$ T cells and TCF1$^+$PD1$^+$CD8$^+$ T cells were significantly increased within eTLSs and mTLSs as compared to entire TME (Fig. 3C). We next evaluated the density of TCF1$^+$PD1$^+$CD8$^+$ T cells and TIM3$^+$PD1$^+$CD8$^+$ T cells in different patient clusters (see also Fig. 2; Cluster 1, no TLSs; Cluster 2, only eTLSs; Cluster 3, both eTLSs and mTLSs as previously determined by immunofluorescence). We found a significantly higher density of TCF1$^+$PD1$^+$CD8$^+$ T cells and TIM3$^+$PD1$^+$CD8$^+$ T cells, as well as PD1$^+$CD8$^+$ T cells as a whole, in HGSOC samples from Cluster 2 and 3 as compared to Cluster 1 (Fig. 3D). These findings were largely confirmed by RNA-seq analyses based on gene signatures associated with T cell stemness, T cell effector activity, T cell proliferation and T cell exhaustion/dysfunction[50] (Supplementary Fig. 6 B, C). Specifically, gene signatures associated with T cell stemness, effector functions as well as T cell exhaustion were significantly upregulated in patients from Cluster 3 as compared to other clusters (Supplementary Fig. 6B, C). In addition, we observed a higher prevalence of TIM3$^+$PD1$^+$CD8$^+$ T cells in tumor cores as compared to the HGSOC stroma (Fig. 3D). On the contrary, the population of TCF1$^+$PD1$^+$CD8$^+$ T cells was more abundant in the stroma than in tumor cores, especially in patients from Cluster 3 (Fig. 3D). Of note, TIM3$^+$PD1$^+$CD8$^+$ T cells accounted for the majority of PD1$^+$CD8$^+$ T cells within the TME of HGSOC patients (Fig. 3D). At least in part, these findings might explain the negative prognostic impact of terminally mature TLSs in HGSOC, as a high density of TIM3$^+$CD8$^+$

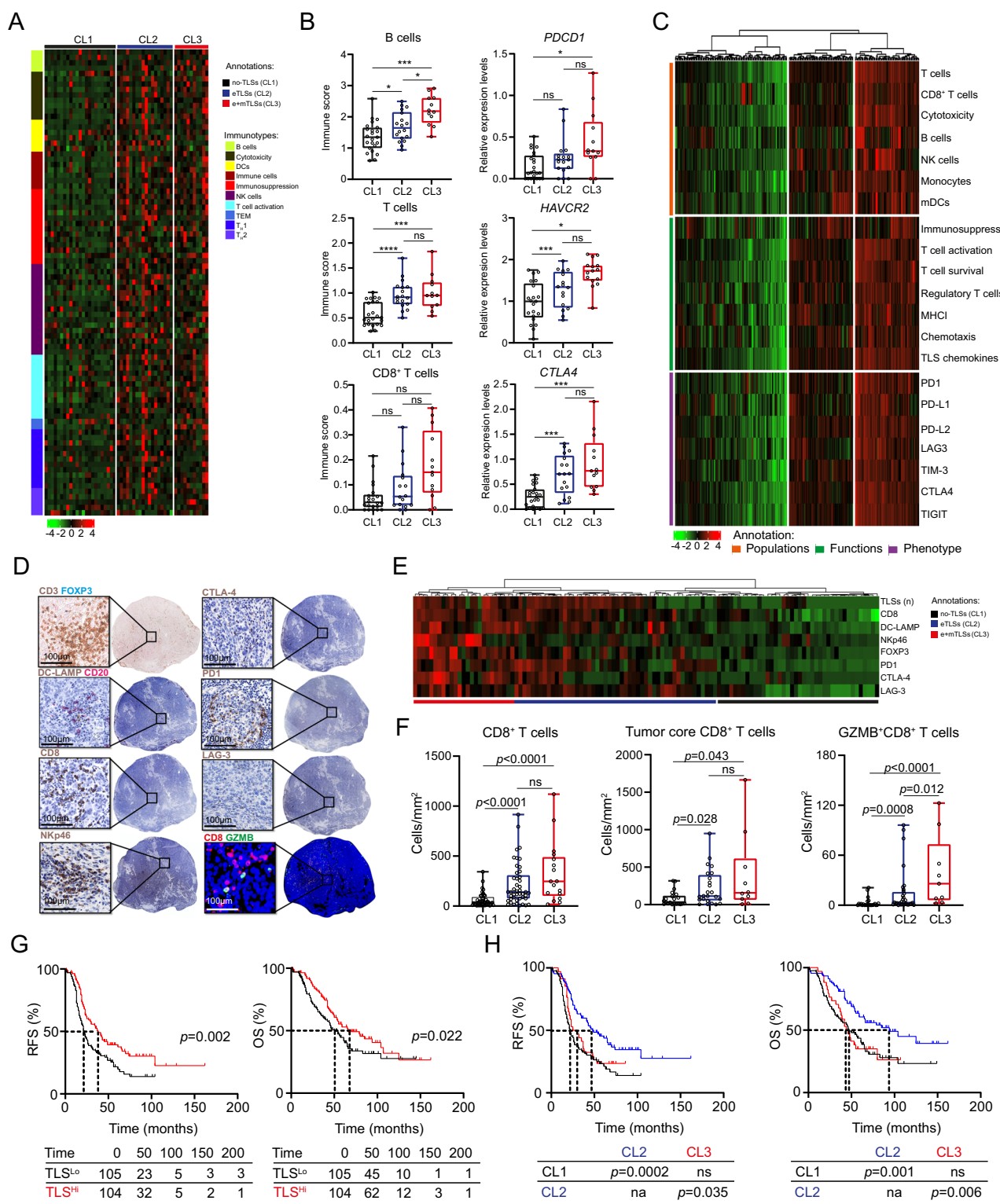

T cells has previously been associated with poor disease outcome in this patient population (Supplementary Fig. 6D)[51]. Conversely, an increased density of TCF1+PD1+CD8+ T cells was associated with non-significnat trend toward favorable prognosis as determined by univariate Cox proportional hazard analyses (Table 1).

Finally, we aimed at comparing the abundance of lymphoid aggregates at different maturation stages and their T cell correlates in samples from patients with HGSOC *vs* NSCLC (Study cohort 5, Supplementary Table 6), as malignancy known to (1) have a high TMB, (2) respond to ICIs, especially when infiltrated by high amounts of

TCF1+PD1+CD8+ T cells or when enriched for TLSs[52–55] (Fig. 3E). We observed a significantly increased abundance of eTLSs and mTLSs in NSCLC samples as compared to HGSOC samples (Fig. 3F; Supplementary Fig. 7A, B).

Furthermore, although the frequency of TIM3+PD1+CD8+ T cells was comparable in HGSOC and NSCLC samples (Fig. 3G), PD1+CD8+ T cells in NSCLC samples were preferentially polarized toward a TCF1+PD1+ T cell phenotype with effector functions (in both the TME as a whole and TLS areas) as compared to HGSOC samples (Fig. 3G–I). Moreover, the density of TCF1+PD1+CD8+ T cells, but not

**Fig. 2 | The clinical impact of TLS maturation on development of antitumor immunity in HGSOC. A** Supervised hierarchical clustering of 53 tumor samples of HGSOC patients (Study Cohort 1) with TLS development (no TLSs, Cluster 1, $n = 23$, only early TLS (eTLS, Cluster 2, $n = 18$), early and mature TLS (eTLS+mTLS, Cluster 3, $n = 12$) based on the expression of 100 genes classified into clusters related to B cells, cytotoxicity, dendritic cells (DCs), immune cells, immunosuppression, natural killers (NK) cells, T cell activation, $T_{EM}$, $T_H1$ and $T_H2$ signatures as determined on RNA sequencing data from Study cohort 1. **B** Gene expression signature associated with B cells, T cells, CD8$^+$ T cells as determined by MCP counter on RNAseq data and relative gene expression levels of *PDCD1*, *HAVCR2*, *CTLA4* as determined on RNAseq data across patients ($n = 53$; Study cohort 1) separated into 3 clusters (clusters determined by immunofluorescence in (**A**), CL1: $n = 23$; CL2: $n = 18$; CL3: $n = 12$). Box plots: lower quartile, median, upper quartile; whiskers, minimum, maximum. Statistical significance was calculated by two-sided Mann–Whitney test. ns not significant. **C** Unsupervised hierarchical clustering of gene signatures related to immune populations (orange), immune functions (green) and immune phenotype (purple) in tumor samples of 304 HGSOC patients from TCGA public database (Study cohort 4). **D** Representative images of double immunohistochemistry for CD3 and FoxP3, DC-LAMP and CD20 cells, single IHC for CD8, NKp46, CTLA-4, PD1,

LAG-3 and double immunofluorescence (IF) staining of CD8 GZMB cells. Scale bar, 100 μm. **E** Unsupervised hierarchical clustering of HGSOC patients ($n = 115$; Study Cohort 1) based on frequency of TLS and densities of CD8$^+$ T cells, DC-LAMP$^+$ DCs, NKp46$^+$ NK cells, FoxP3$^+$ cell, PD1$^+$, CTLA-4$^+$ and LAG-3$^+$ cells as determined by immunostaining. **F** Density of CD8$^+$ (CL1: $n = 44$; CL2: $n = 45$; CL3: $n = 17$), tumor core CD8$^+$ (CL1: $n = 21$; CL2: $n = 24$; CL3: $n = 10$) and GZMB$^+$CD8$^+$ cells (CL1: $n = 24$; CL2: $n = 32$; CL3: $n = 9$) in tumor samples of HGSOC patients (Study Cohort 1) separated into 3 clusters (CL1, no TLS development; CL2, only eTLS development; CL3, both eTLS and mTLS development), as determined by immunostaining. Box plots: lower quartile, median, upper quartile; whiskers, minimum, maximum. Statistical significance was calculated by two-sided Mann–Whitney test. $p$ values are indicated. **G, H** Relapse-free survival (RFS) and overall survival (OS) of 209 HGSOC patients (Study Cohort 1 and 2; Supplementary Table 1) based on median stratification of total TLS (**G**) and based on stratification into 3 clusters (CL1, no TLS development, $n = 87$; CL2, only eTLS development, $n = 86$; CL3, both eTLS and mTLS development, $n = 36$). Survival curves were estimated by the Kaplan–Meier method, and differences between groups were evaluated using log-rank test. Number of patients at risk and $p$ values are reported. *$p < 0.01$; **$p < 0.001$; ***$p < 0.0001$. Source data and exact $p$ values are provided as a Source Data file.

TIM3$^+$PD1$^+$CD8$^+$ T cells, significantly correlated with the number of mTLSs in both NSCLC and HGSOC samples (Fig. 3J).

Altogether, these findings indicate that not only TLS frequency but also TLS maturation influences the phenotypic profile of intratumoral CD8$^+$ T cells in patients with HGSOC. Specifically, an insufficient number of mTLSs in the HGSOC microenvironment appears to be associated with a TIM3$^+$PD1$^+$, rather than a TCF1$^+$PD1$^+$, CD8$^+$ T-cell phenotype, which might contribute to the resistance of patients with HGSOC to ICIs.

## Spatial B-cell profiling identifies in situ maturation and differentiation states associated with high density of $T_{FH}$ cells and TLS formation

As enhanced $T_{FH}$ cell differentiation in a TMB$^{High}$ mouse cancer model has been shown to induce larger TLSs and superior B-cell recruitment[37,56], we next determined the density of CD4$^+$CXCR5$^+$PD1$^+$FoxP3$^-$ $T_{FH}$ cells in HGSOC and NSCLC samples using multiplex immunofluorescence (panel 3) (Fig. 4A). The density of $T_{FH}$ cells was significantly lower in the entire TME of HGSOC samples as compared to NSCLC samples (Fig. 4B). Supporting this notion, we also observed lower frequency of CD4$^+$ T cells, CD8$^+$ T cells and $T_{FH}$ cells, alongside with an overall reduced density of CD21$^+$CD23$^+$ follicular DCs in the mTLSs of HGSOC samples as compared to NSCLC samples (Fig. 4C; Supplementary Fig. 7C). Conversely, the density of CD68$^+$ tumor associated macrophages (TAMs) and PD1$^-$FoxP3$^+$CD4$^+$ regulatory T ($T_{REG}$) cells was comparable in the mTLSs of HGSOC and NSCLC samples (Supplementary Fig. 7C). Taken together, these findings may explain the limited maturity of TLS in the TME of HGSOC, which may be associated with the limited capacity of TLSs to develop GCs with fully mature CD21$^+$CD23$^+$ follicular DCs.

As GC formation within TLSs is crucial for B-cell differentiation, we decided to dissect the localization of specific B cell subsets (as previously defined in single cell transcriptomic analyses of tonsillar B cells)[57] with respect to TLS maturation in HGSOC samples. As expected, all B-cell signatures that we tested were enriched in tumors with abundant TLSs (Study cohort 1) (Fig. 4D, E) and globally associated with TLSs as compared to nTLS areas (Fig. 4F), perhaps with the exception of signatures of mature plasma cells, which at least in some samples were diffused throughout the TME (Fig. 4D). Moreover, we detected gene sets associated with naive B cells preferentially in the proximity of eTLSs and at the margins of mTLSs as compared to other localizations (Fig. 4F). While pre-GC and memory B-cell signatures were predominantly associated with eTLSs, GC-like and plasma cells signatures were associated mTLSs (Fig. 4F). These findings are in line with the notion that mTLSs support B cell maturation, selection and

expansion in situ, culminating with development of mature plasma cells[24].

As TLSs are known to contribute on development of tumor-targeting effector and memory T cell responses[18,40], we next determined the impact of CD19$^+$CD20$^+$ B-cell depletion on the phenotypic and functional profile of tumor-infiltrating lymphocytes (TILs) isolated from freshly resected HGSOCs ($n = 7$; study cohort 6, Supplementary Table 7) and NSCLCs ($n = 7$; study cohort 7, Supplementary Table 8) with a high prevalence of TLSs (Fig. 4G; Supplementary Fig. 8A, B). In this setting, B-cell depletion decreased the percentage of TIM3$^-$PD1$^+$CD8$^+$ T cells while increasing the abundance of TIM3$^+$PD1$^+$CD8$^+$ T cells from both HGSOC and NSCLC samples (Fig. 4H, I). Of note, no impact on T cells phenotypic properties was observed when CD4$^+$ and CD14$^+$ cells were depleted (Supplementary Fig. 9A, B). These findings corroborate the key role of TLS-associated B cells in the preservation of ICI-sensitive T cells.

Taken together, our data suggest that TLS-associated B cells positively influence CD8$^+$ T cells phenotype, ultimately resulting in survival benefits in patients with HGSOC.

## TMB correlates with TLS abundance in HGSOC patients

Both TMB (which is generally considered as a surrogate marker for tumor neoantigens) and TLS frequency have been associated with superior tumor-targeting immunity and ICI sensitivity in cancer patients[10,22,23,25,26,58], suggesting these two parameters may influence each other. To validate this hypothesis, we used the TrueSight-Onco500 panel to determine mutational profile and TMB alongside with Nanostring PanCancer Immune Profiling Panel to investigate the relative expression of 770 genes associated with immune responses in 35 TMB$^{Lo}$ and 33 TMB$^{Hi}$ patients with HGSOC from Study cohort 2 (Supplementary Fig. 1, Supplementary Table 2). Using unsupervised hierarchical clustering based on "metagene" cell markers[46], we identified three clusters of patients including two clusters enriched for gene signatures associated with T cell, B cells and cytotoxicity as well as for transcripts encoding various co-inhibitory receptors (Fig. 5A). Notably, these clusters largely included HGSOC samples with higher-than-median TMB (TMB$^{Hi}$, a non-standard TMB cutoff imposed by low TMB range of this patient cohort: 0-10 somatic mutations per Mb, 2.35 median TMB) (Fig. 5B) and/or TLS number, largely confirming our findings from Study cohort 1 (Fig. 2).

TMB$^{Hi}$ HGSOC samples had increased amounts of CD8$^+$ T cells and CD20$^+$ B cells as compared to their TMB$^{Lo}$ counterparts, as determined by immunofluorescence (Fig. 5C). Moreover, we observed that tumor samples of TMB$^{Hi}$ patients contain higher numbers of eTLSs and mTLSs as compared to their TMB$^{Lo}$

**Table 1 | Univariate Cox proportional hazard analyses on 123 HGSOC patients from Study cohort 1 (Study group 1; Supplementary Table 1)**

| Variable | | HR (95% CI) | *p* value |
|---|---|---|---|
| Age | | 1.02 (1.00–1.04) | 0.034 |
| Stage | I | | |
| | II | 0.70 (0.22–2.21) | 0.546 |
| | III | 2.27 (1.18–4.35) | 0.014 |
| | IV | 1.37 (0.47–3.94) | 0.565 |
| Debulking | R0 | | |
| | R1 | 1.49 (0.58–3.82) | 0.401 |
| | R2 | 1.86 (1.17–2.94) | 0.008 |
| CA125 | | 1.10 (0.97–1.24) | 0.114 |
| CD8$^+$ T cells | | 0.85 (0.73–0.99) | 0.041 |
| CD20$^+$ B cells | | 0.85 (0.75–0.96) | 0.011 |
| DC-LAMP$^+$ DCs | | 0.80 (0.64–1.01) | 0.067 |
| NKp46$^+$ NK cells | | 0.92 (0.73–1.15) | 0.471 |
| TLS clusters | CL1 | | |
| | CL2 | 0.53 (0.32–0.87) | 0.012 |
| | CL3 | 1.06 (0.57–1.96) | 0.851 |
| GZMB$^+$CD8$^+$ T cells | | 0.87 (0.76–0.99) | 0.045 |
| PD1$^+$CD8$^+$ | | 0.86 (0.69–1.06) | 0.172 |
| TCF1$^+$PD1$^+$CD8$^+$ | | 0.75 (0.53–1.04) | 0.098 |
| TIM3$^+$PD1$^+$CD8$^+$ | | 0.86 (0.68–1.07) | 0.173 |

Statistical significance was calculated by Univariate cox proportional hazard analyses. HR and p values are indicated.
*TLS* tertiary lymphoid structures.

counterparts (Fig. 5D). Similar findings were obtained when we analyzed the two cohorts of patients with HGSOC that were included in Study cohort 2 independently from each other (Supplementary Fig. 10A–D). Finally, TMB exhibited a positive correlation with TLS gene signatures and homologous recombination deficiency (HRD) score in 304 HGSOC patients from the TCGA public database (Supplementary Fig. 10E). In line with this notion, we observed a similar trend toward positive correlation (although sub-significant) between TMB and a TLS-relevant chemokine gene signature across 12 cancer types from the TCGA public database (Supplementary Fig. 10F). Despite positive correlation between TMB levels and TLS formation, we failed to observe a similar impact of individual somatic mutations on HGSOC-relevant genes such as *BRCA1*, *BRCA2* and *TRP53* on TLS formation and activation using immuno-fluorescence and TrueSightOnco500 panel in Study cohort 2 samples (Fig. 5E, F).

To validate our previous findings documenting the impact of TMB and TLS formation on T cell phenotypic profile in the HGSOC microenvironment, we next determined the density of TCF1$^+$PD1$^+$ and TIM3$^+$PD1$^+$CD8$^+$ T cells in TMB$^{Lo}$ and TMB$^{Hi}$ patients using multiplex immunofluorescence (Panel 2) (Fig. 5G). Supporting our data, we observed increased density of both TCF1$^+$PD1$^+$CD8$^+$ ($p = 0.007$) and TIM3$^+$PD1$^+$CD8$^+$ T cells ($p = 0.007$) in TMB$^{Hi}$ (>2.35 somatic mutations per megabase) patients as compared to TMB$^{Lo}$ counterparts (Fig. 5H). Nevertheless, effector CD8$^+$ T cells in TMB$^{Hi}$ tumor preferentially exhibited a TIM3$^+$PD1$^+$rather than a TCF1$^+$PD1$^+$ phenotype (Fig. 5H).

Together, these data corroborate a potential association between TMB, TLS formation and effector T cell phenotype in the HGSOC microenvironment.

### TLSs and ICI sensitivity in mouse models of TMB$^{Lo}$ and TMB$^{Hi}$ ovarian cancer

To experimentally dissect the link between TMB level, TLS development and sensitivity to ICIs, we harnessed two mouse models of ovarian cancer that are syngeneic to C57BL/6J mice and exhibit significantly different level of somatic mutations and TMB, namely, ID8 cells and *Brca1$^{-/-}$ Trp53$^{-/-}$/Myc/Hras* SO1 cells[59] (Fig. 6A, B) to generate tumors in immunocompetent female C56BL/6 mice for TLS analyses (Fig. 6C). Immunofluorescence analysis of tumors collected after intraperitoneal implantation demonstrated that TMB$^{Hi}$ SO1 tumors (day 20) develop a higher number of advanced lymphoid aggregates (LAs) than TMB$^{Lo}$ ID8 (day 60) lesions, the most mature of which do not form at all in the latter (Fig. 6D–F). Similar to our findings in human tumor samples, TMB$^{Hi}$ mouse ovarian tumors with high LAs frequency were associated with an increased density of effector T cells with a TCF1$^+$PD1$^+$CD8$^+$, TIM3$^-$PD1$^+$CD8$^+$ and TIM3$^+$PD1$^+$CD8$^+$ T cell phenotypes as compared to their TMB$^{Lo}$ counterparts (Fig. 6G). Moreover, in line with our findings in samples from HGSOC patients, effector PD1$^+$CD8$^+$ T cells preferentially displayed a TIM3$^+$PD1$^+$CD8$^+$ phenotype in the TMB$^{Hi}$ ovarian cancer model (Fig. 6G).

Next, we experimentally dissected the impact of TMB and TLS development on therapeutic responses to an ICI targeting PD1 (Fig. 6C). Anti-PD1 therapy provided a significant survival benefit to TMB$^{Hi}$ SO1 lesions ($p = 0.012$; Fig. 6H) but not TMB$^{Lo}$ ID8 tumors (Fig. 6I). Moreover, as compared to control conditions, PD1 blockage resulted in a significant increase of TCF1$^+$PD1$^+$CD8$^+$ T cells in the TMB$^{Hi}$ SO1 model, while the frequency of TIM3$^+$PD1$^+$CD8$^+$ and TIM3$^-$PD1$^+$CD8$^+$ T cells subsets remained unchanged (Fig. 6J, K). Finally, SO1 tumors responding to PD1 blockage contained an increased frequency of IFNG$^+$ and Ki67$^+$ CD8$^+$ T cells as compared to control lesions (Fig. 6L).

With the caveats imposed by the use of two distrinct cellular systems, these findings suggest that an elevated TMB is associated with the development of experimental HGSOCs that contain increased amounts of TLSs at different maturation stages and exhibit improved sensitivity to ICIs.

## Discussion

TLSs are ectopic lymphoid organs that develop in peripheral tissues, including tumors, upon exposure to inflammatory signals[16]. In multiple cancer types, TLSs provide privileged sites for the local presentation of tumor antigens to T cells and B cells, resulting in their proliferation and acquisition of effector and memory functions[16,18,60]. In line with a key role for TLSs in natural and (immuno)therapy-driven immunosurveillance, an expanding clinical literature demonstrates that a high density of intratumoral TLSs, B cells or plasma cells, as well as the presence of antibodies against tumor-associated antigens, is associated with favorable disease outcome not only in primary and metastatic HGSOC[37–42,61], but also in other oncological indications[22,23,25,26,62,63].

Here, we harnessed five independent patient cohorts to define the immunobiology and prognostic relevance of TLSs in HGSOC (which is resistant to ICIs). We found that while HGSOC-associated eTLSs and mTLSs contain high level of effector and memory CD8$^+$ T cells, they are much less common than eTLSs and mTLSs associated with NSCLC (which is sensitive to ICIs). Moreover, we documented that the most prominent impact on disease outcome originates from the abundance of immune effector cells within TME associated with TLS formation (Figs. 1, 2). These data corroborate the notion that TLSs contribute to the development of anticancer immunity in situ[19,40,64]. On the other hand, our data linking the HGSOC (but not the NSCLC) environment to a predominance of ICI-insensitive TIM3$^+$PD1$^+$CD8$^+$ T cells phenotype over (ICI-sensitive) TCF1$^+$PD1$^+$CD8$^+$ T cells[52,53,65–67] suggest that – at odds with NSCLC – the sparcity and limited maturation of TLSs in HGSOC fail to preserve an ICI-sensitive TCF1$^+$PD1$^+$CD8$^+$ T cell compartment, rather allowing for the acquisition of a TIM3$^+$PD1$^+$ phenotype that has been associated with ICI resistance (Fig. 3). With respect to this notion, mTLS with GC formation mainly occurred at the invasive margin of HGSOC lesions as compared to a largely intratumoral localization in

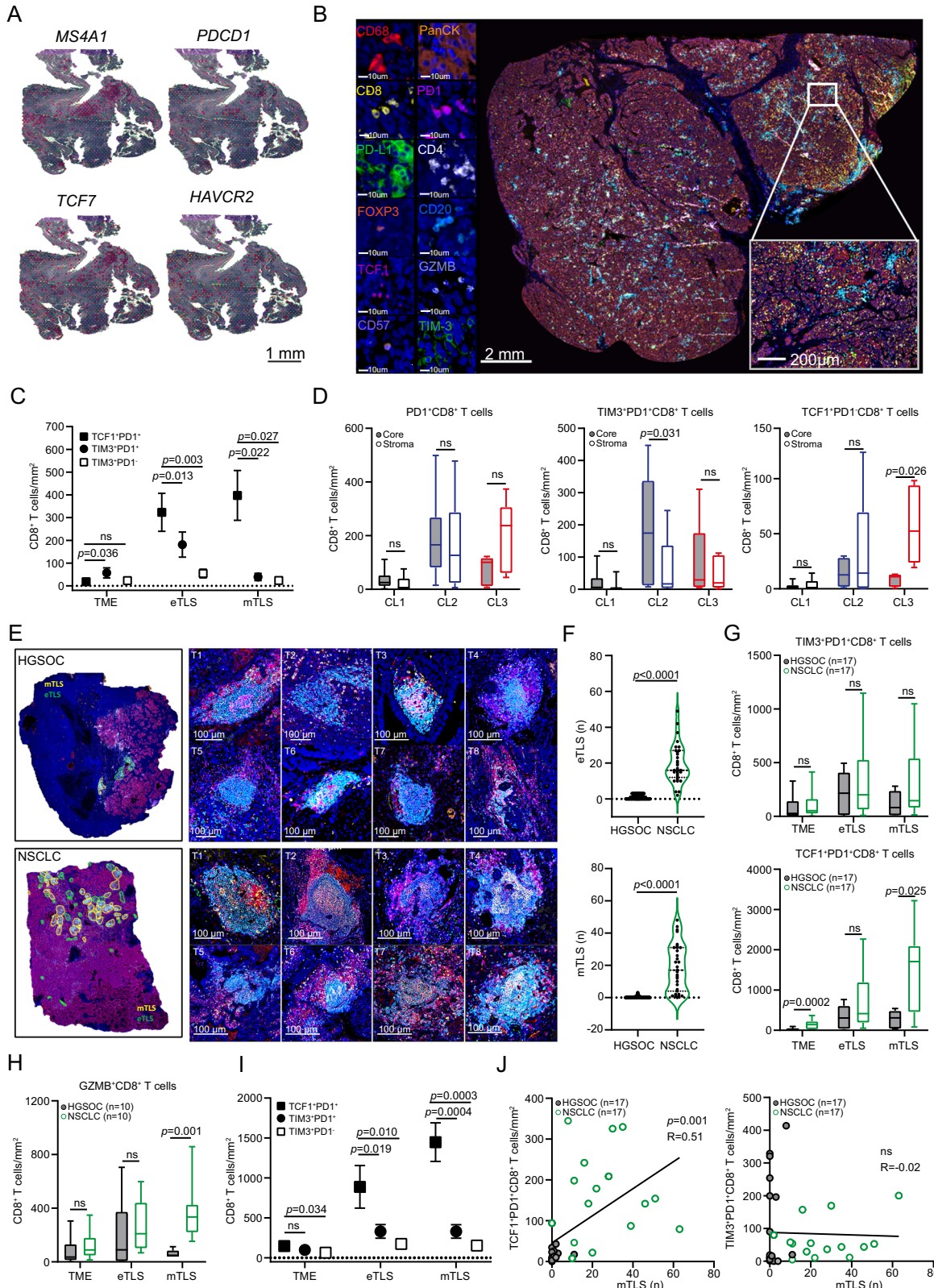

NSCLC samples. In addition, HGSOC-associated TLSs are not only less frequent but also less developed, with apparent lower density of CD4⁺ T cells, GZMB⁺CD4⁺ T cells, GZMB⁺CD8⁺ T cells, T_FH cells and particularly follicular DCs, which might contribute to insufficient B cell and T cell activation within TME, as previously reported[37,68]. Further data documenting the redistribution of immune cells in distinct TLSs of the same HGSOC lesionare urgently awaited.

Thus, our results in HGSOC and NSCLC samples demonstrate that both the frequency and the level of maturation of TLSs impact the phenotypic and functional properties of intratumoral T cells. These observations may (at least partially) explain the limited sensitivity of patients with HGSOC to conventional ICIs targeting the PD1/PD-L1 axis, as low level of neoantigens might lead to reduced level of T_FH cells, insufficient TLS development without fully

**Fig. 3 | TLS frequency and maturation impact the T cell phenotype in HGSOC.**
**A** Spatial co-localization of TLSs with *MS4A1*, *PDCD1*, *TCF7* and *HAVCR2* genes in
TLS^Hi tumor sample from Study Cohort 1. **B** Representative image of immuno-
fluorescence of CD68, CD8, PD-L1, FoxP3, TCF1, CD57, PanCK, PD1, CD4, CD20,
GZMB and TIM3 staining (*immunofluorescence panel 2*). Scale bar, 2 mm, 10 μm and
200 μm. **C** Density of TCF1$^+$PD1$^+$CD8$^+$ T cells, TIM3$^+$PD1$^+$CD8$^+$ and TIM3$^+$PD1$^-$CD8$^+$
T cells within complete tumor microenvironment (TME, including TLSs), early TLS
(eTLS) and mature TLS (mTLS) of 19 HGSOC tumor samples (Study cohort 1, $n$ = 19).
Mean and SEM are shown. Statistical significance was calculated by two-sided
Wilcoxon matched-pairs signed rank test. $p$ values are indicated. **D** Density of
PD1$^+$CD8$^+$, TIM3$^+$PD1$^+$CD8$^+$ and TCF1$^+$PD1$^+$CD8$^+$ T cells within tumor stroma and
tumor core of 19 HGSOC patients (Study Cohort 1) separated into 3 clusters (CL1, no
TLS development, $n$ = 8; CL2, only eTLS development, $n$ = 6; CL3, both eTLS and
mTLS development, $n$ = 5, as previously determined in Fig. 2) Box plots: lower
quartile, median, upper quartile; whiskers, minimum, maximum. Statistical sig-
nificance was calculated by two-sided Wilcoxon matched-pairs signed rank test.
$p$ values are indicated. Representative images (**E**) and violin plot (**F**) showing the
frequency of eTLS and mTLS in 209 HGSOC and 31 non-small cell lung carcinoma

(NSCLC) (Supplementary Table. 6). Statistical significance was calculated by two-
sided Mann–Whitney test. $p$ values are indicated. **G, H** Density of TIM3$^+$PD1$^+$CD8$^+$,
TCF1$^+$PD1$^+$CD8$^+$ T cells and GZMB$^+$CD8$^+$ T cells within complete tumor micro-
environment (TME, including TLS), eTLS and mTLS of HGSOC (Study cohort 1) and
NSCLC samples (Study cohort 5) as determined by immunofluorescence. Number
of patients involved in respective analyses are indicated. Box plots: lower quartile,
median, upper quartile; whiskers, minimum, maximum. Statistical significance was
calculated by two-sided Mann–Whitney test. $p$ values are indicated. **I** Density of
TCF1$^+$PD1$^+$CD8$^+$ T cells, TIM3$^+$PD1$^+$CD8$^+$ and TIM3$^+$PD1$^-$CD8$^+$ T cells within complete
tumor microenvironment (TME, including TLS), early TLS (eTLS) and mature TLS
(mTLS) in 17 NSCLC tumor samples (Study cohort 5). Mean and SEM are shown.
Statistical significance was calculated by two-sided Wilcoxon matched-pairs signed
rank test. $p$ values are indicated. **J** Correlation between frequency of mTLS and
TIM3$^+$PD1$^+$CD8$^+$ and TCF1$^+$PD1$^+$CD8$^+$ T cells in TME of 17 HGSOC and 17 NSCLC
patients as determined by multispectral immunofluorescence. $p$ values are indi-
cated. R, Pearson correlation coefficient. Source data are provided as a Source
Data file.

activated GCs, and hence the generation of B cells unable to pre-
serve an ICI-sensitive T cell phenotype[69,70]. Moreover, they align with
previous findings from our group demonstrating a key role for TIM3
in the establishment of local immunosuppression in the HGSOC
environment[51].

We also demonstrated that TLS-associated B cells are important
for the preservation of ICI-sensitive TCF1$^+$PD1$^+$CD8$^+$ T cells in HGSOC
and NSCLC, and that their interactions with CD8$^+$ T cells favorably
influence disease outcome. Similar observations have been reported in
patients with other cancer types[71,72], potentially suggesting that while
mature DCs are key for the activation of tumor-targeting immunity in
lymph nodes, B cells dominate intratumoral antigen presentation of
therapeutic relevance for ICI sensitivity[73].

Finally, we showed that the TMB is linked to TLS formation in
both human HGSOC and mouse ovarian cancer models, despite the
fact that HGSOC bears a low-to-intermediate TMB[10]. This is parti-
cularly relevant as TMB has recently been shown to predict superior
ICI responsiveness in patients that receive conventional ICIs such a
NSCLC[58], and the same holds true for intratumoral TLS abundance
and frequency of T$_{FH}$ cells[22,23,25,26]. Since our data originate from
different cellular systems, it will be important to validate these
findings in purely syngeneic TMB$^{Hi}$ *vs* TMB$^{Lo}$ preclinical models
of HGSOC.

Overall, these findings and observations delineate a model with
prognostic implications in which: (1) a high TMB is associated with
high T$_{FH}$ abundance and PD1$^+$ expression, supporting the develop-
ment of abundant intratumoral mature TLSs that preserve (ICI-
responsive) TCF1$^+$PD1$^+$CD8$^+$ T cells (as in the NSCLC setting); (2) a
low TMB is linked with an overall "cold" (ICI-unresponsive) TME (as
in HGSOC patients with lower-than-median TMB); but, most inter-
estingly, (3) an intermediate TMB is associated with limited amounts
of TLSs (predominantly localized in marginal zones of tumor sam-
ples) that might be insufficient to preserve TCF1$^+$PD1$^+$CD8$^+$ T cells.
Whether the latter setting may involve at least some degree of B cell
dysfunction remains unclear. Along similar lines, it remains to be
clarified whether NSCLC patients with TMBs in the lower end of the
spectrum for this oncological indication (and hence comparable to
HGSOC patients with TMB in their higher end of the spectrum) also
bear intermediate levels of mature TLSs that fail to sustain
TCF1$^+$PD1$^+$CD8$^+$ T cells and whether this might explain their insen-
sitivity to conventional ICIs.

Irrespective of these and other unresolved questions, our findings
delineate a clinically relevant link between TMB and TLS formation
that, in the HGSOC setting, appears to underlie the generation of a
dysfunctional T cell population insensitive to conventional ICIs but
potentially responsive to TIM3 blockers.

## Methods
### Clinical samples and patient characteristics
All tissue samples and health-related data in our study were collected
after ethical review and approval of the Ethics Committee listed below.

**Study cohort 1 (HGSOC).** A retrospective series of 183 formalin-fixed
paraffin-embedded (FFPE) tumor samples were obtained from
patients with HGSOC who underwent primary surgery in the absence
of neo-adjuvant chemotherapy between 2008 and 2014 at University
Hospital Hradec Kralove ($n$ = 123; study group 1) and University
Hospital Motol (n = 60; study group 2) (Czech Republic) (Supple-
mentary Fig. 1). Baseline characteristics of these patients are sum-
marized in Supplementary Table 1. Pathology staging was performed
according to the 8th TNM classification from 2017, and histologic
types were determined according to the current WHO
classification[74]. Written informed consent was obtained from
patients before inclusion in the study. Data on long-term clinical
outcome were obtained retrospectively by interrogation of muni-
cipality registers or families. The protocol was approved by the local
Ethical Committee (201607S14P).

**Study cohort 2 (HGSOC).** A retrospective series of 79 formalin-fixed
paraffin embedded (FFPE) tumor samples from HGSOC who under-
went primary surgery in the absence of neo-adjuvant chemotherapy
were enrolled in NCT02107937 study as a control group without
maintenance therapy (study group 3, $n$ = 26) and with maintenance
dendritic cell based immunotherapy ($n$ = 53; study group 4) (Supple-
mentary Fig. 1). Baseline characteristics of these patients are sum-
marized in Supplementary Table 2. Pathology staging was performed
according to the 8th TNM classification from 2017, and histologic
types were determined according to the current WHO classification[74].
Written informed consent was obtained from patients before inclusion
in the study. The protocol was approved by the local Ethical
Committees.

**Study cohort 3 (HGSOC).** A retrospective series of 40 FFPE tumor
samples were obtained from patients with HGSOC who underwent
primary surgery in the absence of neo-adjuvant chemotherapy
between 2008 and 2014 at University Hospital Basel, Switzerland
(Supplementary Fig. 1; Supplementary Table 5). Written informed
consent was obtained from patients before inclusion in the study. The
protocol was approved by the local Ethical Committee
(#EKNZ2017–01900).

**Study cohort 4 (HGSOC).** RNA-seq data from 304 patients with
HGSOC were identified in The Cancer Genome Atlas (TCGA) public

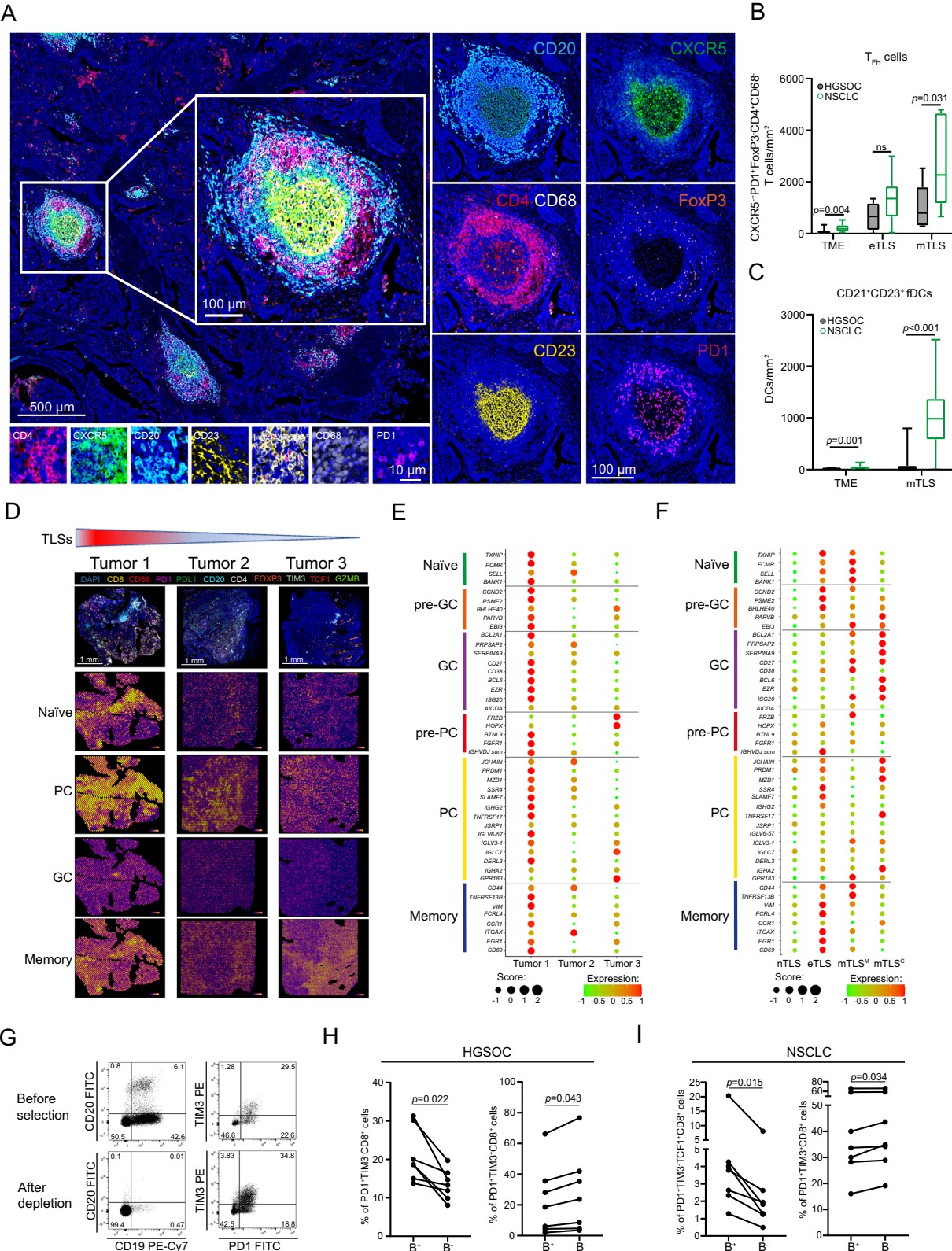

database (https://cancergenome.nih.gov/) (Supplementary Fig. 1). TCGA data was downloaded from the UCSC Xena Data Hub (https://xenabrowser.net/datapages/). Normalized TCGA RNA-sequencing (RNA-seq) data were log-transformed.

**Study cohort 5 (NSCLC).** A retrospective series of 31 formalin-fixed paraffin-embedded (FFPE) tumor samples were obtained from patients with stage III adenocarcinoma NSCLC who underwent primary surgery in the absence of neo-adjuvant chemotherapy between 2014 and 2022 at University Hospital Hradec Kralove (Supplementary Fig. 1; Supplementary Table 6). Written informed consent was obtained from patients before inclusion in the study. The protocol was approved by the local Ethical Committee (201607S14P).

**Fig. 4 | In situ activated intratumoral B cells impact CD8⁺ T cells phenotype in HGSOC and NSCLC.** Representative image (**A**) and box plot showing the density of CXCR5⁺PD1⁺FoxP3⁻CD68⁻CD4⁺ $T_{FH}$ cells (**B**) in the complete tumor microenvironment (TME), eTLS and mTLS of HGSOC (*n* = 17) and NSCLC (*n* = 10) patients and **C** box plot showing density of CD23⁺CD20⁺ follicular dendritic cells (fDCs) in TLS^{Lo} and TLS^{Hi} HGSOC (*n* = 16) and NSCLC (*n* = 14) patients. Box plots: lower quartile, median, upper quartile; whiskers, minimum, maximum. Statistical significance was calculated by two-sided Mann–Whitney test. *p* values are indicated. **D**, **E** Spatial co-localization and dot plot showing expression profile of genes signatures of B cells subtypes, e.g. naive, pre-plasma cells (pre-PC), plasma cells (PC), pre-germinal center (pre-GC), germinal center (GC) and memory B cells within 3 individual tumor samples (Study cohort 1) with decreasing frequency of TLS, as determined by Visium spatial transcriptomic. **F** Dot plot showing expression profile of genes signatures of B cell subtypes, e.g. naive, pre-germinal center (pre-GC), germinal center (GC), pre-plasma cells (pre-PC), plasma cells (PC) and memory B cells within non-TLS areas (nTLS), early TLS (eTLS), margin (mTLS^M) and central (mTLS^C) area of mTLS in one selected TLS^{Hi} tumor sample as determined by Visium spatial transcriptomic. **G-I** Representative dot plot and flow cytometry analyses for frequency of TIM3⁻PD1⁺CD8⁺ and TIM3⁺PD1⁺CD8⁺ T cells before and after CD19⁺CD20⁺ B cells depletion from native HGSOC (*n* = 7; Study cohort 6) (**H**) and NSCLC (*n* = 7; Study cohort 7) (**I**) tumor tissue. Statistical significance was calculated by two-sided Wilcoxon matched-pairs signed rank test. *p* values are indicated. Source data are provided as a Source Data file.

**Study cohort 6 (HGSOC).** An additional series of samples from 12 HGSOC patients was prospectively collected in the absence of neo-adjuvant chemotherapy at University Hospital Kralovske Vinohrady (Prague, Czech Republic) (Supplementary Fig. 1). This study was conducted in accordance with the Declaration of Helsinki and the protocol was approved by the local Ethical Committee (Progress UK, Q40/11). Written informed consent was obtained from patients before inclusion in the study. Baseline characteristics of these patients are summarized in Supplementary Table 7.

**Study cohort 7 (NSCLC).** An additional series of samples from 7 stage III adenocarcinoma NSCLC patients was prospectively collected in the absence of neo-adjuvant chemotherapy at University Hospital Motol (Prague, Czech Republic) (Supplementary Fig. 1; Supplementary Table 8). This study was conducted in accordance with the Declaration of Helsinki and the protocol was approved by the local Ethical Committee. Written informed consent was obtained from patients before inclusion in the study.

**Immunohistochemistry (IHC) and immunofluorescence analyses**
Immunostaining with antibodies specific for CD3, CD8, CD20, FoxP3, programmed cell death 1 (PDCD1, best known as PD1), lymphocyte activating gene 3 (LAG-3), cytotoxic T lymphocyte-associated protein 4 (CTLA4), lysosomal associated membrane protein 3 (LAMP3; best known as DC-LAMP), NKp46 was performed according to conventional protocols[51]. Briefly, tumor specimens were fixed in neutral buffered 10% formalin solution and embedded in paraffin as per standard procedures. In brief, 4 μm-thick tissue sections were deparaffinized and rehydrated in a descending alcohol series (100, 96, 70, and 50%), followed by antigen retrieval with Target Retrieval Solution (Leica) in pH 6.0 (for CD3, FoxP3) in EDTA pH 8.0 (for CD8, CD20, DC-LAMP and NKp46) in TRIS EDTA pH 9.0 (for CTLA-4, LAG-3 and PD1) in a pre-heated water bath (97 °C, 30 min). Sections were allowed to cool down to RT for 30 min. Endogenous peroxidase or alkaline phosphatase was blocked with 3% $H_2O_2$, levamisole (Vector), or blocking solution Bloxall (Vector), respectively, for 10–15 min. Thereafter, sections were treated with Protein Block (DAKO) for 15 min and incubated with primary antibodies, followed by the revelation of enzymatic activity (Supplementary Table 9). Sections were counterstained with hematoxylin (DAKO) for 30 s. Images were acquired using a Leica Aperio AT2 scanner (Leica). *Immunofluorescence panel 1* (CD4, CD8, CD20, CD21, CD23, DC-LAMP, and GZMB) were performed according to conventional protocols[51] (Supplementary Table 9; Supplementary Fig. 1). Briefly, 4-μm-thick FFPE tissue sections were deparaffinized and rehydrated in a descending alcohol series (100, 96, 70, and 50%), followed by antigen retrieval with Target Retrieval Solution (Leica) in EDTA pH 8.0 with a heated water bath (97 °C, 30 min). Sections were allowed to cool down to RT for 30 min, then treated with Signal Enhancer (Thermo Fisher Scientific) for 30 min and Blocking Buffer (Thermo Fisher Scientific) for 1 h. The DC-LAMP-specific antibody (Dendritics, 1:350) was incubated overnight at 4 °C, the CD8-specific

antibody (Abcam,1:60) for 90 min at RT, the CD20-specific antibody (Dako, 1:300) for 1 h at RT. Thereafter, slides were incubated with appropriate HRP Polymer secondary antibodies for 1 h at RT, followed by Tyramide Signal Amplification (Thermo Fisher Scientific) (Supplementary Table 9). Finally, sections were treated with TrueBlack Lipofuscin Autofluorescence Quencher (Biotium) for 30 s and mounted with ProLong Gold Antifade Reagent containing DAPI (Thermo Fisher Scientific) (Supplementary Table 9). Staining specificity was determined using appropriate isotype controls. Images of whole tumor sections were acquired using a Leica Aperio AT2 scanner (Leica). Thereafter the same sections were restrained with antibodies specific for CD4, CD21 and CD23 using sequential IHC protocol (Supplementary Table 9; see Supplementary Methods)[75]. Between every staining step the slides were scanned, and final image was composed by the HALO10 software (Indica labs) using the registration algorithm.

*Immunofluorescence panel 2* with antibodies specific for CD8, PD1, CD4, FoxP3, CD20, GZMB, CD68, CD274 (best known as PD-L1) was performed according to the manufacturer's instructions (ULTIVUE) (Supplementary Table 9). Thereafter the same sections were restrained with antibodies specific for for CD4, TCF1, GZMB, CD57, TIM3 and PanCyto using sequential IHC protocol. Further details are provided in Supplementary Materials and Methods. *Immunofluorescence panel 3* (CXCR5, PD1, FoxP3, CD23, CD20, CD4 and CD68) were performed using sequential IHC (Supplementary Table 9; for full details see Supplementary Methods). Sections were treated with Protein Block (DAKO) for 15 min and incubated with primary antibodies anti-human CXCR5, followed by the revelation of enzymatic activity (Supplementary Table 9). Sections were counterstained with hematoxylin (DAKO) for 30 s. Images were acquired using a Leica Aperio AT2 scanner (Leica). Thereafter the same sections were restrained with antibodies specific for PD1, FoxP3, CD23, CD20, CD4 and CD68 using sequential IHC protocol (Supplementary Table 9). Between every staining step the slides were scanned, and final image was composed by the HALO10 software (Indica labs) using the registration algorithm.

**Sequential immunohistochemistry (IHC) protocol for detection follicular T cells**
Immunostaining with antibodies specific for CXCR5, programmed cell death 1 (PDCD1, best known as PD1), FoxP3, CD23, CD20, CD4, and CD68 was performed according to sequential protocol. In brief, 4 μm-thick tissue sections were deparaffinized and rehydrated in a descending alcohol series (100, 96, 70, and 50%), followed by antigen retrieval with Target Retrieval Solution (Leica) in pH 9 (for first sequential staining antibody CXCR5) in a preheated water bath (97 °C, 30 min). Sections were allowed to cool down to RT for 30 min. Endogenous peroxidase or alkaline phosphatase activity was blocked with blocking solution Bloxall (Vector), for 10 min. Thereafter, sections were treated with Normal Horse Serum 2,5% (Vector) for 20 min and incubated with anti-CXCR5 primary antibodies (1:500, 60 min) (Supplementary Table 9), followed by the revelation of enzymatic activity (AEC substrate, Vector). Sections were counterstained with hematoxylin (DAKO) for 30 s. Images were acquired using a Leica

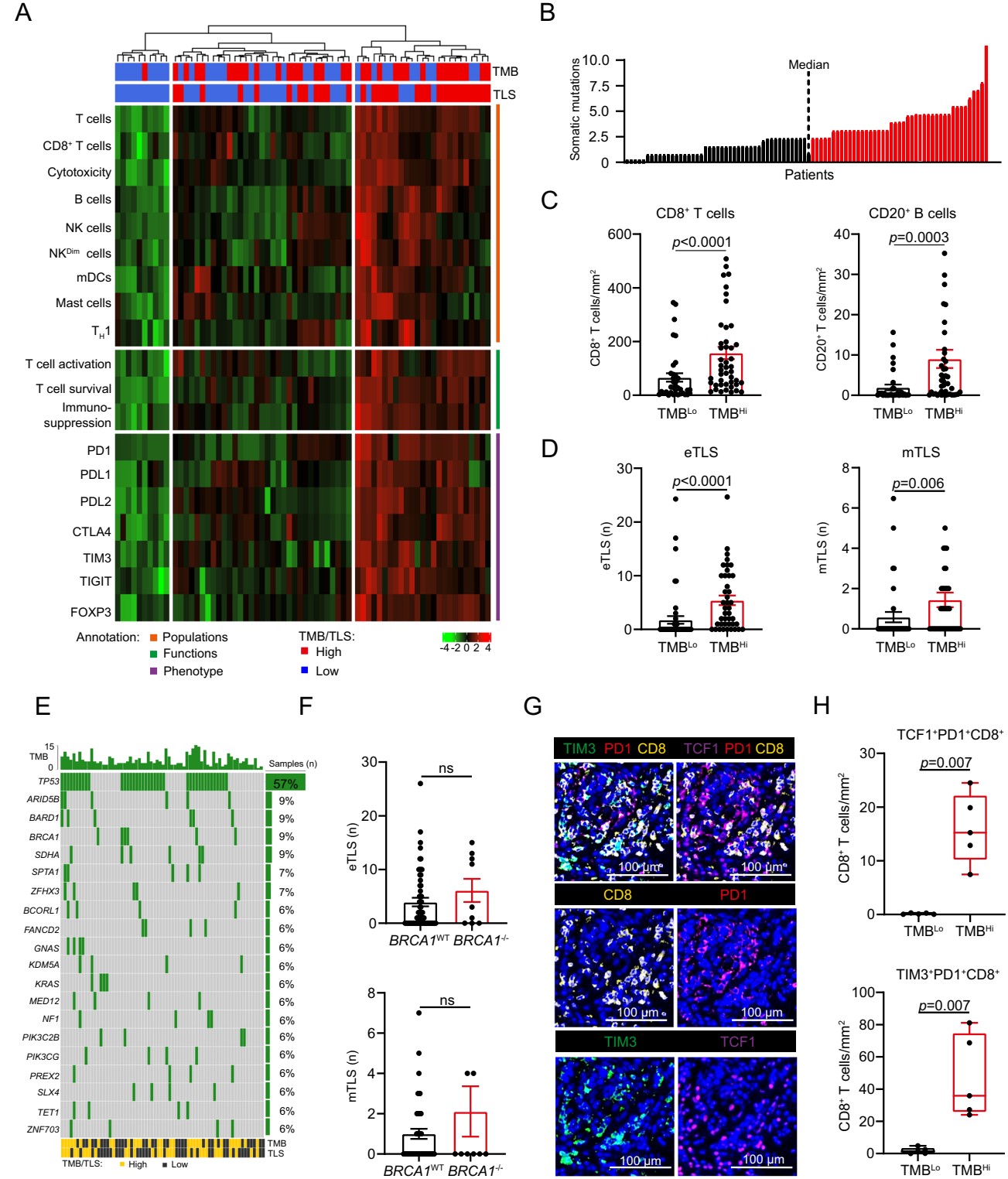

Aperio AT2 scanner (Leica). Thereafter the same sections were decolorated and antibody was removed by heating in pH 9 in microwave for 30 min. Sections were allowed to cool down to RT for 30 min. The staining process was repeated for additional markers, including anti-PD1 (1:50, 120 min), anti-FOXP3 (1:50, 120 min), anti-CD23 (1:200, 60 min), anti-CD20 (1:50, 60 min), anti-CD4 (1:50, 120 min) and anti-CD68 (1:500, 60 min) (Supplementary Table 9). Final image was composed from individual staining sequential steps by the HALO10 software (Indica labs) using the deconvolution and registration algorithm.

**B cells depletion and flow cytometry analyses**

As previously described, total live mononuclear cells were isolated from fresh tumor specimens (Supplementary Table 7, 8)[51]. Tumor-derived single cells suspensions were split into halves. One half was depleted of B cells using EasySep™ HLA Chimerism Whole Blood B Cell Positive Selection Kit (EasySep) according to the manufacturer's instructions. The second half was subjected to the same procedures without addition of microbeads. After magnetic separation, both cell suspensions were cultured in RPMI 1640 supplemented 10% inactivated FCS, L-glutamine and penicillin–streptomycin (Invitrogen) in 24-

**Fig. 5 | TMB positively correlates with the formation of TLS structures in HGSOC. A** Unsupervised hierarchical clustering of gene signatures related to immune populations (orange), immune functions (green) and immune phenotype (purple), as determined by the PanCancer Immune Profiling Panel from Nanostring and further annotated by tumor mutational burden (TMB) determined by True-SightOnco500 and TLS numbers as determined by immunofluorescence staining in tumor samples (*n* = 68; Study cohort 2). **B** Distribution of TMB in 79 tumor samples (Study cohort 2), with median display. Dot plot representing density of CD8⁺ T cells and CD20⁺ B cells (**C**) and number of early TLS (eTLS) and mature TLS (mTLS) (**D**) in TMB$^{Lo}$ (*n* = 32) and TMB$^{Hi}$ (*n* = 44) tumor samples, as determined by median stratification (Study cohort 2). Mean and SEM are shown. Statistical significance was

calculated by two-sided Mann–Whitney test. *p* values are indicated. **E** Oncoplot showing the profile of somatic mutations in 79 tumors annotated by TMB and number of TLSs as determined by median stratification. **F** Number of eTLS and mTLS in *BRCA1*$^{WT}$ (*n* = 70) and *BRCA1*$^{-/-}$ (*n* = 9) HGSOC patients (*n* = 79; Study cohort 2). Mean and SEM are shown. Statistical significance was calculated by the Mann–Whitney test. *p* values are indicated. Representative images (**G**) and box plots (**H**) showing density of TCF1⁺PD1⁺CD8⁺ and TIM3⁺PD1⁺CD8⁺ T cells in 5 TMB$^{Lo}$ and 5 TMB$^{Hi}$ tumor samples (Study Cohort 2), as determined by median stratification. Box plots: lower quartile, median, upper quartile; whiskers, minimum, maximum. Statistical significance was calculated by two-sided Mann–Whitney test. *p* values are indicated. Source data are provided as a Source Data file.

well plates for 1 day without any additional stimuli. The purity and cell yield of the magnetic separation was assessed using panel of fluorescent primary antibodies or appropriate isotype controls for 20 min at 4 °C in the dark, followed by washing and acquisition on a Fortessa flow cytometer (BD Biosciences). The phenotype of CD8⁺ T cells was asseseed at day 2 using panel of fluorescent extracelluar and intracelluar primary antibodies or appropriate isotype controls for 20 min at 4 °C in the dark, followed by washing and acquisition on a Fortessa flow cytometer (BD Biosciences) (Supplementary Table 10). Flow cytometry data were analyzed with the FlowJo software (TreeStar) (Supplementary Fig. 8).

## Quantitative evaluation of TLSs and cell densities

Only TLS made up of more than 50 cells were included in the analysis (Supplementary Fig. 2). Tumor samples with cellular aggregates that contain more than 50 cells on hematoxylin and eosin (H&E) were further analyzed by IHC using anti-CD20 and -CD23 antibodies. In the absence of CD21⁺ and CD23⁺ positivity, the TLS was identified as early TLS (Supplementary Fig. 2A). In addition, eTLS were defined as aggregates of minimum size of 250μm, with majority of cells being CD20⁺ B cells in close proximity contacts (min 3 μm), with presence of CD4⁺, CD8⁺ T cells. TLS were defined as "mature" when at least one CD21⁺ and CD23⁺ dendritic cell was detected in the TLS (Supplementary Fig. 2B), further using immunofluorescence panel 1. Cell density (cells/mm²) for CD8⁺ T cells, CD3⁺FoxP3⁺ cells, CTLA-4⁺ cells, DC-LAMP⁺ DCs, CD20⁺ B cells, GZMB⁺CD8⁺ T cells, LAG3⁺ cells, NKp46⁺ cells, PD1⁺ TIM3⁻CD8⁺ T cells, CXCR5⁺PD1⁺FoxP3⁻CD4⁺CD68⁺ T$_{FH}$ cells, CD21⁺CD23⁺fDCs, GZMB⁺CD4⁺ T cells, GZMB⁺CD8⁺ T cells, PD1TIM3⁺CD8⁺ T cells, TCF1⁺PD1⁺CD8⁺ T cells, and TIM3⁺PD1⁺CD8⁺ T cells was quantified in whole tumor sections and TLS areas (Supplementary Fig. 2C) by the HALO10 software (Indica labs) using the HighPlex FL 4.1.0.3 and classifier algorithm. Quantitative assessments were performed by three independent investigators (JF, LK, JR) and independently reviewed by either of two expert pathologists (JL, AR).

## DNA/RNA isolation from FFPE

RNA and DNA were isolated using AllPrep DNA/RNA FFPE Kit (Qiagen) according to manufacturer instructions. RNA concentration and purity were determined using a NanoDrop 2000c (Thermo Scientific, Germany). Samples were stored at −80 °C until further use.

## Bulk RNA sequencing and analysis

Raw FASTQ sequencing files were aligned to human reference genome (build h19) with bowtie2 (version 2.3.2) and tophat2 (version 2.1). Expression levels as raw "counts" were calculated from aligned reads with mapping quality ≥10 using htseq-count (version 0.6.0). Differential gene expression analyses were performed using DESeq2 (version 1.24.0) in R. Unsupervised hierarchical clustering heatmaps were used for differentially expressed genes (DEGs) using the R package ComplexHeatmap (2.8.0) based on the Euclidean distance and Ward2 clustering method. R package Circlize (0.4.15) was used for the circular layout representations of TLS structures. R package ggplot2 (3.3.6) was

used for the dotplot representation of scaled gene expression of TLS structures.

## Tumor mutational burden (TMB) analyses in human FFPE samples

The NGS library was prepared using TruSight Oncology 500 DNA kit (Illumina) as per manufacturer's recommendations. Denaturation of NGS libraries was achieved by bead-based normalization, according to the manufacturer protocol. Final denaturation was performed according to the NextSeqSystem Denature and dilute Guide. NGS libraries were sequenced on the NextSeq 550 using NextSeq 500/550 High Output kit v2.5 (300 cycles) in pair-end mode 2 × 101 bp. The TMB analysis was done using the TSO500 LocallApp. TMB analysis step generates TMB metrics from the annotated small variant JSON file and the gVCF file that is generated from the small variant filtering analysis step. The annotated JSON file is used to retrieve information regarding individual variants, such as allele counts in public databases and resulting consequences at a transcript level. The gVCF is used to evaluate the effective panel size denominator. To remove germline variants from the TMB calculation, the software uses a combination of public database filtering and post-database filtering strategy that uses allele frequency information and variants in close proximity. First, the component excludes any variant with an observed allele count ≥10 in any of the GnomAD exome, genome, and 1000 genomes database. To filter germline variants that are not observed in the database, the software identifies variants on the same chromosome with an allele frequency within a certain range. If a given variant is not filtered out based on occurrence in the databases, variants on the same chromosome with similar allele frequencies will be grouped, and if 5 or more similar variants are found to have been filtered, the variant of interest is removed from the TMB Calculation. Additionally, variants with an allele frequency ≥ 90% are removed from the TMB calculation as well. The TMB is calculated as follows: TMB = Eligible Variants / Effective panel size. Eligible Variants (numerator) including variants not removed by the filtering strategy; variants in the coding region (Seq Cds); variant Frequency ≥ 5%; Coverage ≥ 50×; SNVs and Indels (MNVs excluded); nonsynonymous and synonymous variants; variants with COSMIC count ≥ 50 excluded. Effective Panel Size (denominator) include. Total coding region with coverage > 50× and excluding low confidence regions in which variants are not called.

## Tumor mutational burden (TMB) analyses in murine cancer cells

DNA exome of BR5 and ID8 cell lines was sequenced by Novaseq (Illumina). Read files were mapped onto mouse genome version GRCm38_68[76] using STAR aligner (ver. 2.7.0c)[77]. The variants were identified using STRELKA (ver. 2.9.10)[78]. All variants were annotated by ENSEMBL BIOMART (Release 102, November 2020). The TMB is reported as the number of mutations per megabase (mut/Mb).

## Spatial transcriptomics

For the analysis of human tumor TLSs by the Visium spatial gene expression assay (10x Genomics), 4 selection steps were used on

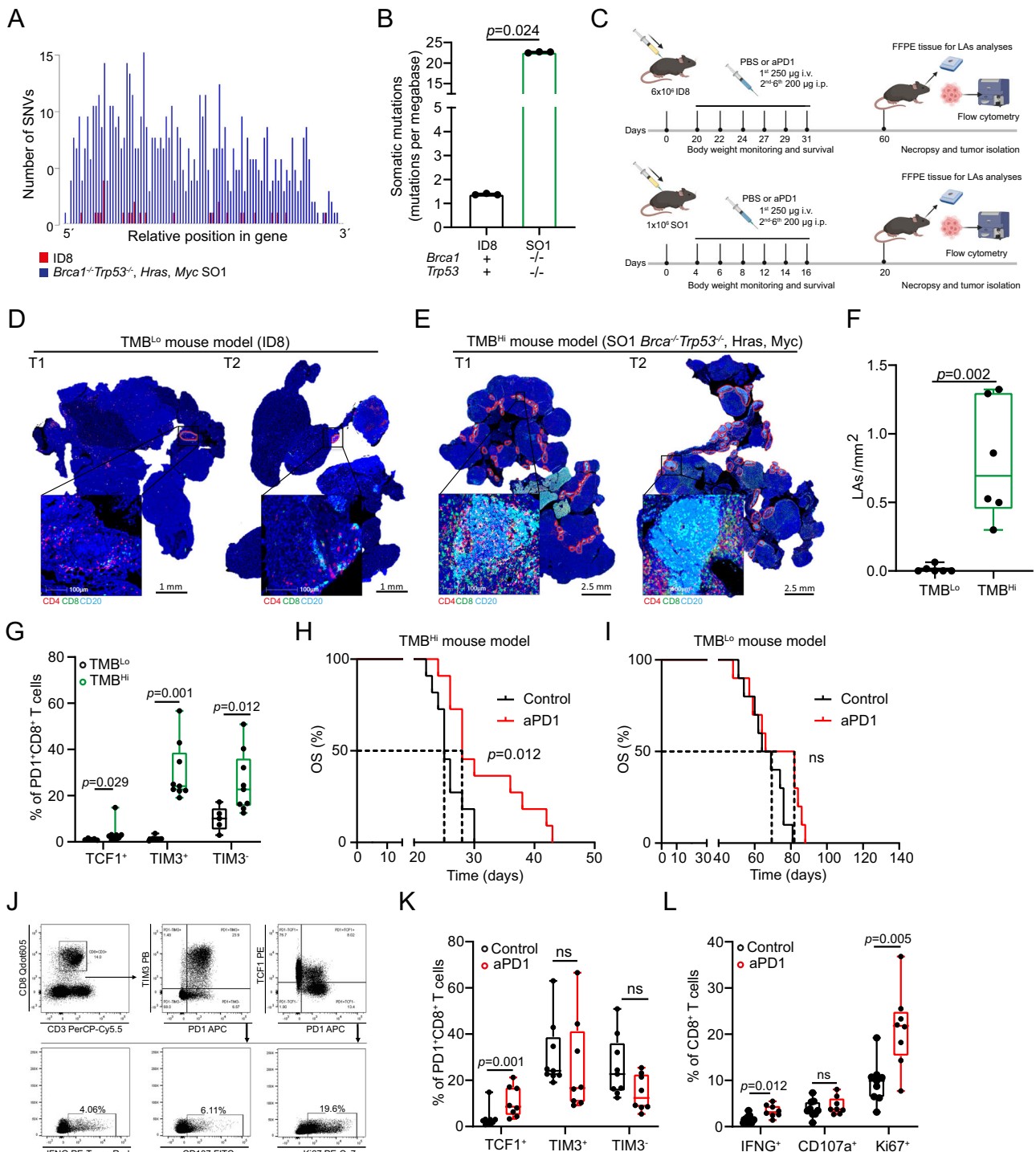

**Fig. 6 | TLSs and ICI sensitivity in mouse models of TMB^Lo and TMB^Hi ovarian cancer. A**, **B** Barplots showing single-nucleotide variants (SNVs) and the somatic mutations prevalence (mutations per megabase) in ID8 (*n* = 3) and *Trp53^-/- BRCA1^-/-*, *Myc, Hras* SO1 (*n* = 3) C57BL/6 syngeneic mouse ovarian cancer cell lines. Mean and SEM are shown. Statistical significance was calculated by multiple T-test. *p* values are indicated. **C** Experimental design for the analysis of lymphoid aggregates (LAs) development and efficacy of anti-PD1 therapy in TMB^Lo ID8 and TMB^Hi SO1 experimental syngeneic mouse models. Created with BioRender.com. Representative immunostaining for CD4, CD8, CD20 and CD21 and a box plot showing density of lymphoid aggregates within TMB^Lo ID8 (*n* = 6) (**D**) and TMB^Hi SO1 (*n* = 6) (**E**) ovarian tumors (**F**). Scale bars 1 and 2.5 mm. Box plots: lower quartile, median, upper quartile; whiskers, minimum, maximum. Statistical significance was calculated by two-sided Mann–Whitney test. *p* values are indicated. **G** Box plot showing percentage of TCF1^+PD1^+CD8^+, TIM3^+PD1^+CD8^+ and TIM3^-PD1^+CD8^+ T cells in tumor samples of TMB^Lo ID8 (*n* = 5) and TMB^Hi SO1 (*n* = 9) experimental models. Box plots: lower quartile, median, upper quartile; whiskers, minimum, maximum. Statistical significance was calculated by two-sided Mann–Whitney test. *p* values are indicated. Overall survival (OS) of TMB^Hi SO1 (*n* = 22; control: 11, aPD1: 11) (**H**) and TMB^Lo ID8 (*n* = 20; control: 10, aPD1: 10) (**I**) experimental models after anti-PD1 therapy. Survival curves were estimated by the Kaplan–Meier method and differences between groups were evaluated using log-rank test. Representative dot plot (**J**) and flow cytometry analyses for percentages of TCF1^+PD1^+CD8^+, TIM3^+PD1^+CD8^+ and TIM3^-PD1^+CD8^+ T cells (**K**) and IFNG^+, CD107^+ and Ki67^+CD8^+ T cells (**L**) in tumor samples of the TMB^Hi SO1 experimental model in the presence (*n* = 8) or absence (*n* = 9) of anti-PD1 therapy. Box plots: lower quartile, median, upper quartile; whiskers, minimum, maximum. Statistical significance was calculated by two-sided Mann–Whitney test. *p* values are indicated. Source data are provided as a Source Data file.

samples with available FFPE material: (1) TLS were identified by IHC for CD20 and DC-LAMP on whole FFPE sections, (2) IHC data were confirmed by multispectral immunofluorescence panel 1 and 2 (see section immunofluorescence) on whole FFPE sections, (3) TLS positivity was validated by the expression of the so-called "12-chemokines signature" (*CCL2, CCL3, CCL4, CCL5, CCL8, CCL18, CCL19, CCL21, CXCL9, CXCL10, CXCL11*, and *CXCL13*)[48,49] from bulk RNAseq data, and (4) finally, to estimate the relative abundance of immune cell populations, we used "metagene" markers[46,57]. The markers were applied on the SCT normalized scores of each tumor. Selected samples were evaluated for size, shape and overall status for application on the Visium slides (10x Genomics) to provide optimal coverage of $6.5 \times 6.5$ mm$^2$ capture areas. Only FFPE tumor blocks with relevant TLS status and DV.200 over 50 were selected for the Visium FFPE assay. Spatial transcriptomics assay, Visium spatial gene expression slides and reagents were used according to the manufacturer's instructions (10x Genomics). Libraries were prepared with Truseq Illumina libraries and sequenced on NovaSeq (Illumina) at a minimum sequencing depth of 25,000 read pairs per spatial spot.

### Visium data processing
Visium pre-processed data were imported into R via Seurat V.4.0.1. Spatial spots featuring more than 30% of mitochondrial genes and less than 300 genes were filtered out, as they identified necrotic or damaged tissue areas which were validated by pathologist. Genes with counts in less than 5 spatial spots were discarded. Spots featuring folded were removed. Raw counts were normalized with the SCTransform function of Seurat using the "assay=spatial" parameter.

**Immune abundances estimation.** The spatial immune infiltrates of each tumor were estimated with metagene markers (1) which computes abundances scores of immune and stromal populations. B cell phenotype was defined based on scRNAseq B cell signature (2).

**TLS annotations on the Visium data.** Spatial spots belonging to TLS were defined by IHC and IF staining on the consecutive slide for FFPE tumor. Selection of spatial spots belonging to TLS was performed manually using the software Loupe Browser 6 (10x genomics)

### Next-generation sequencing data analysis
As previously described[79], hierarchical clustering analysis was conducted for differentially expressed genes (DEG) using the ComplexHeatmap package in R, based on the Euclidean distance and ward.D2 clustering method. The MCP-counter R package was used to estimate the abundance of tissue-infiltrating immune cell populations on bulk RNAseq data[46,47].

### Mice
Female C57BL/6 mice, aged 8–12 weeks were from the Institute of Microbiology, Czech Academy of Sciences (Prague, Czech Republic). Housing was in individually ventilated cages. Food and water was provided ad libitum, Teklad Global 18% Protein Rodent Diet, irradiated. Water in sterile prefilled bottles. Dark/Light cycle on a 12 h automated schedule. Temperature $21 \pm 2$ °C and humidity $50 \pm 10\%$. Animals were randomly assigned to experimental groups, no blinding was performed during these experiments.

### Mouse tumor models
SO1 (kindly provided by Dr. Sandra Orsulic, University of California, Los Angeles) murine ovarian cancer cells were cultured in the Dulbecco's Modification of Eagle's Medium (DMEM) (Sigma) supplemented with 10% fetal bovine serum (FBS; Gibco), antibiotics (100 U mL$^{-1}$ penicillin sodium and 100 μg mL streptomycin sulfate, (Gibco)). ID8 (kindly provided by prof. Ian A. McNeish, Imperial College London), murine ovarian cancer cells were cultured in DMEM (Sigma)

supplemented with 4% fetal bovine serum (FBS, Sigma Aldrich), 1% Insulin -Transferin-Selenium (ITS; Sigma) and antibiotics. All cells were cultured at 37 °C with 5% CO$_2$ before being harvested and suspended in serum-free DMEM (Sigma) for tumor injection. All cell lines were routinely checked for Mycoplasma spp. contamination by the PCR-based LookOut® Mycoplasma PCR Detection kit (Millipore Sigma). $1 \times 10^6$ SO1 and $6 \times 10^6$ ID8 tumor cells were mixed at a 1:1 ratio with cold Cultrex BME (Biotechne) and injected intraperitoneally into the fat tissue close the ovary of 8–12-week-old C57BL/6 mice. For all mice experiments, animals were routinely monitored for disease progression and overt toxicity such as >20% decline in body weights, lethargy, immobility, fur ruffling, and fur loss. Mice were euthanized upon experimental endpoint. Mice reached a humane endpoint and were humanely euthanized upon >50% body weight increase due to ascites formation or institutional compassionate euthanasia criteria were met. Maximal tumor size/burden was not exceeded. Animal experiments strictly followed a protocol approved the Institutional Animal Care and Use Committee of the Academy of Sciences of the Czech Republic, and conducted in compliance with local and European guidelines

### Statistical analysis
Survival analyses were estimated by Cox proportional hazard regressions and the Kaplan–Meier method, where differences between the groups of patients were calculated using the log-rank test. For Log-rank tests, the prognostic value of continuous variables was assessed using median cutoff of intra-tumoral immune cell densities, TLS numbers and TMB value. The Mann–Whitney test was used to compare the density of tumor-infiltrating cells among patient groups. The Wilcoxon test was used to compare the frequency of immune markers before and after depletion experiment. The Fisher exact test was used to compare patient distribution across subgroups. The enrichGo function from ClusterProfiler was used to identify enriched GO terms based on hypergeometric distribution. $p$ values were adjusted for multiple comparisons using the Benjamini–Hochberg correction. All analyses were performed with Prism 8.4.2 (GraphPad) and R software V.4.1.0 (http://www.r-project. org/). $p$ values < 0.05 were considered statistically significant.

### Reporting summary
Further information on research design is available in the Nature Portfolio Reporting Summary linked to this article.

## Data availability
The raw data and datasets, including bulk RNAseq data of 50 HGSOC patients from study cohort 1, spatial transcriptomic data from study cohort 1, whole exome sequencing of ID8 and SO1 cell line generated in this study have been deposited in the European Nucleotide Archive repository and Sequence Read Archive (SRA-NCBI), under the access number PRJEB56495, and SAMN38195788, SAMN38195789. The ovarian cancer publicly available data used in this study are available in the TCGA database via the Xena browser: https://xenabrowser.net/datapages/?cohort=TCGA%20Ovarian%20Cancer%20(OV)&removeHub=https%3A%2F%2Fxena.treehouse.gi.ucsc.edu%3A443.
The raw data and dataset on patients samples from NCT02107937 clinical study (study cohort 2), including nanostring data and tumor mutational burden analyses data (TruSight Oncology 500) that support findings of this study, are not publicly accessible due to the lack of consent from study subjects to deposit these data. The datasets can be made available from Sotio Biotech a.s. upon request to M.He. The response to access request will be provided within 1 month and data will be available for 1 year once access has been granted. Additional individual de-identified participant data related to the study cohorts and full imaging datasets can be shared upon request to the corresponding author (J.F.). The remaining data are available within the

Article, Supplementary Information or Source Data file. Source data are provided with this paper.

## Code availability

All analyses reported in this study used the statistical software R (v.3.6.1). All codes employed or generated during the current study are available publicly at GitHub - pepap/sotio-NatComm-2023.

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

## Acknowledgements

This study was sponsored by Sotio Biotech a.s., Prague. J.La. and A.Ry. were supported by the project BBMRI-CZ LM2023033, Cooperatio Program (research area DIAG), and the European Regional Development Fund-Project BBMRI-CZ.: Biobank network – a versatile platform for the research of the etiopathogenesis of diseases, No: EF16_013/0001674. L.Ga. is/has been supported (as a PI unless otherwise indicated) by one R01 grant from the NIH/NCI (#CA271915), by two Breakthrough Level 2 grants from the US DoD BCRP (#BC180476P1; #BC210945), by a grant from the STARR Cancer Consortium (#I16-0064), by a Transformative Breast Cancer Consortium Grant from the US DoD BCRP

(#W81XWH2120034, PI: Formenti), by a U54 grant from NIH/NCI (#CA274291, PI: Deasy, Formenti, Weichselbaum), by the 2019 Laura Ziskin Prize in Translational Research (#ZP-6177, PI: Formenti) from the Stand Up to Cancer (SU2C), by a Mantle Cell Lymphoma Research Initiative (MCL-RI, PI: Chen-Kiang) grant from the Leukemia and Lymphoma Society (LLS), by a Rapid Response Grant from the Functional Genomics Initiative (New York, US), by startup funds from the Dept. of Radiation Oncology at Weill Cornell Medicine (New York, US), by industrial collaborations with Lytix Biopharma (Oslo, Norway), Promontory (New York, US) and Onxeo (Paris, France), as well as by donations from Promontory (New York, US), the Luke Heller TECPR2 Foundation (Boston, US), Sotio Biotech a.s. (Prague, Czech Republic), Lytix Biopharma (Oslo, Norway), Onxeo (Paris, France), Ricerchiamo (Brescia, Italy), and Noxopharm (Chatswood, Australia). H.Ta. and S.Or. were supported by the National Cancer Institute grant R01CA149385. S.Or. was also supported by the National Cancer Institute grant R01CA208753 and the Veterans Administration Merit Award VA-ORD BX006020. L.Ro. and M.J.H. were supported by program PROGRES Q28 Charles University research (oncology). P.Du. was supported by Ministry of Health, CZ, research project RVO 64165 and by the project BBMRI.cz, reg n. LM2023033. V.H.S. and F.Ja. were supported by Personalized Health and Related Technologies Pioneer Project #715 (to V.H.S. and F.aJ.) and SNF project grant #207975 to V.H.S. M.Ko. was supported by the project National Institute for Cancer Research (Programme EXCELES, ID Project No. LX22NPO5102) - Funded by the European Union - Next Generation EU. I.A.M. is supported by Ovarian Cancer Action (grant 006) and an NIHR Senior Investigator Award and receives infrastructure support from the Imperial NIHR Biomedical Research Centre and Imperial College Healthcare Tissue Bank. J.P. was supported by Czech Academy of Sciences RVO 68378050 and by the project LM2023036 Czech Centre for Phenogenomics provided by Ministry of Education, Youth and Sports of the Czech Republic.

## Author contributions

Concept and design: L.Ka., M.He., J.Fu.; development of methodology: L.Ka., J.Ra., M.He., T.La., J.Pa., H.Ta., S.Or.; acquisition of data: L.Ka., J.Ra., M.He., T.La., J.To., J.Pa., J.Dr., K.Mo., A.Fi., S.Vo., M.Ko., H.Ta., F.Ja., S.Or.; provision of critical samples and reagents: J.La., A.Ry., P.Du., R.Ko., T.Br., P.Sk., L.Ca., M.Ko., J.Pr., I.Pr., V.Ko., A.Ta., R.Li., F.C.M., J.V.J, V.H.S., F.Ja., I.A.M., M.J.H., L.Ro., D.Ci.; analysis and interpretation of data: L.Ka., J.Ra., M.He., T.La., J.Pa., F.Ja., J.Fu.; writing, review, and/or revision of the manuscript: L.Ka., J.Ra., M.He., S.Or., L.Ga., R.Sp., J.Fu.; study supervision: J.Fu.

## Competing interests

L.G. is/has been holding research contracts with Lytix Biopharma, Onxeo and Promontory, has received consulting/advisory honoraria from Boehringer Ingelheim, AstraZeneca, OmniSEQ, Onxeo, The Longevity Labs, Inzen, Sotio, Promontory, Noxopharm and the Luke Heller TECPR2 Foundation, and holds Promontory stock options. R.Sp. is minority shareholder of Sotio Biotech a.s. A.R. declare advisory services and invited lectures for Amgen, AstraZeneca, BMS, Eli-Lilly, Janssen-Cilag, MSD, Roche. All other authors have no conflicts to declare.

## Additional information

Lenka Kasikova[1,24], Jana Rakova[1,24], Michal Hensler[1], Tereza Lanickova[1,2], Jana Tomankova[1], Josef Pasulka [1], Jana Drozenova[3], Katerina Mojzisova[1], Anna Fialova [1], Sarka Vosahlikova[1], Jan Laco[4], Ales Ryska[4], Pavel Dundr[5], Roman Kocian[6], Tomas Brtnicky [7], Petr Skapa[8], Linda Capkova[8], Marek Kovar [9], Jan Prochazka [10], Ivan Praznovec[11], Vladimir Koblizek[12], Alice Taskova[13], Hisashi Tanaka [14], Robert Lischke[15], Fernando Casas Mendez[16], Jiri Vachtenheim Jr [15], Viola Heinzelmann-Schwarz[17], Francis Jacob [17], Iain A. McNeish[18], Michal J. Halaska[19], Lukas Rob[19], David Cibula[6], Sandra Orsulic [20], Lorenzo Galluzzi [21,22,23], Radek Spisek[1,2] & Jitka Fucikova [1,2] ✉

[1]Sotio Biotech a.s., Prague, Czech Republic. [2]Department of Immunology, Charles University, 2nd Faculty of Medicine and University Hospital Motol, Prague, Czech Republic. [3]Department of Pathology, 3rd Faculty of Medicine and University Hospital Kralovske Vinohrady, Prague, Czech Republic. [4]The Fingerland Department of Pathology, Charles University, Faculty of Medicine and University Hospital Hradec Kralove, Hradec Kralove, Czech Republic. [5]Department of Pathology, 1st Faculty of Medicine, Charles University and General University Hospital, Prague, Czech Republic. [6]Department of Gynaecology, Obstetrics and Neonatology, General University Hospital in Prague, 1st Faculty of Medicine, Charles University, Prague, Czech Republic. [7]Department of Gynecology and Obstetrics, 1st Faculty of Medicine, Charles University, University Hospital Bulovka, Prague, Czech Republic. [8]Department of Pathology and Molecular Medicine, 2nd Faculty of Medicine, Charles University and University Hospital Motol, Prague, Czech Republic. [9]Laboratory of Tumor Immunology, Institute of Microbiology of the Czech Academy of Sciences, Prague, Czech Republic. [10]Czech Center for Phenogenomics, Institute of Molecular Genetics of the Czech Academy of Sciences, Vestec, Czech Republic. [11]Department of Gynecology and Obstetrics, Charles University, Faculty of Medicine and University

Hospital Hradec Kralove, Hradec Kralove, Czech Republic. [12]Department of Pneumology, University Hospital Hradec Kralove, Hradec Kralove, Czech Republic. [13]Department of Thoracic Surgery, Charles University, 3rd Faculty of Medicine and Thomayer University Hospital, Prague, Czech Republic. [14]Departments of Surgery and Biomedical Sciences, Samuel Oschin Comprehensive Cancer Institute, Cedars-Sinai Medical Center, West Hollywood, CA, USA. [15]3rd Department of Surgery, First Faculty of Medicine, Charles University and University Hospital Motol, Prague, Czech Republic. [16]Oncology and Pneumology Department, 2nd Faculty of Medicine, Charles University and University Hospital Motol, Prague, Czech Republic. [17]Ovarian Cancer Research, Department of Biomedicine, University Hospital Basel and University of Basel, Basel, Switzerland. [18]Ovarian Cancer Action Research Centre, Department of Surgery and Cancer, Imperial College London, London, UK. [19]Department of Gynecology and Obstetrics, Charles University, 3rd Faculty of Medicine and University Hospital Kralovske Vinohrady, Prague, Czech Republic. [20]Department of Obstetrics and Gynecology, David Geffen School of Medicine, University of California, Los Angeles, Los Angeles, CA, USA. [21]Department of Radiation Oncology, Weill Cornell Medical College, New York, NY, USA. [22]Sandra and Edward Meyer Cancer Center, New York, NY, USA. [23]Caryl and Israel Englander Institute for Precision Medicine, New York, NY, USA. [24]These authors contributed equally: Lenka Kasikova, Jana Rakova. ✉e-mail: fucikova@sotio.com

