## [Peer Review File · Nature Communications]

Tertiary lymphoid structures and B cells determine clinically relevant T cell phenotypes in ovarian cancerREVIEWERS' COMMENTS:

Reviewer #1 (Remarks to the Author): with expertise in ovarian cancer, TLS, cancer immunology

This is an interesting and well presented paper that addresses the role of TLS in the immune response to high-grade serous ovarian cancer (HGSOC). This is a timely study, given the high level of interest in TLS, B cells, plasma cells (PC) and associated immune cells in the cancer immunotherapy field currently. The authors make the unexpected and counter-intuitive claim that the most mature TLS (i.e., those with secondary follicles, or sTLS) are associated with higher levels of (to use their term) "terminally exhausted" CD8 T cells. They claim that HGSOC differs from NSCLC and melanoma in this regard, and they argue this could explain why HGSOC has a lower response rate to checkpoint blockade. If true, this would imply that TLS are only beneficial to anti-tumor immunity until they reach a certain stage of maturation, after which they become detrimental. This would have important implications for immunotherapies designed to induce TLS, and for cancer immunotherapy in general.

Strengths:

1. Important and timely topic
2. Relatively larger HGSOC cohorts
3. State of the art methods such as spatial transcriptomics

Major issues:

1. In general, the paper overlooks the considerable body of work by others regarding B cells, PCs, TLS and antibody responses in HGSOC (e.g., groups like Conejo-Garcia, Nelson, Balkwill, Shulman). Consequently, several of the key results are in fact confirmatory yet are not presented as such. The paper could be improved by summarizing this prior work in the introduction and better emphasizing the novel findings of the present study.
2. The authors' finding that TLS are associated with higher TMB appears to contradict a prior report from Kroeger et al using TCGA data (PMID: 26763251); this discrepancy should be discussed. Moreover, given the narrow range of TMB values obtained (from 0-10 per Mb, median 2.2), and given that only 1-10% of mutations are thought to give rise to neoantigens, it seems unlikely (and hence a larger burden of proof) that TMB can explain higher numbers

of TLS in HGSOC. This important issue should be validated in an independent cohort and discussed.

3. The authors use terms like "exhausted" and "terminally exhausted" based on molecular phenotypes alone, without showing any functional data. These are still evolving and controversial concepts in the field. Therefore such terms could be mentioned as hypotheses in the Discussion, but should not be used as definitive nomenclature in the Results.

4. On a related note, the authors assume "exhausted" and "terminally exhausted" phenotypes are detrimental to anti-tumor immunity. This is an oversimplified view. There is considerable evidence that T cells with a seemingly exhausted phenotype are in fact associated with favorable properties and good outcomes in HGSOC (see Duraiswamy PMID: 34739845; Laumont PMID: 33963000) and other cancers. Thus, these issues should be presented and discussed with more objective, nuanced language.

5. The authors' 4 clusters from RNA-seq should be better described to give readers a general sense of how they differ, without needing to study the heatmap. In general, the sections describing these clusters could be made more clear and succinct.

6. Lines 350-357 and Fig 2g,h: It is unclear whether these data are based on IHC/IF enumeration of TLS or RNA-seq based clusters. The wording on lines 350-351 is particularly unclear in this regard. Moreover, it appears from Fig 2h that sTLS hi patients have worse RFS, but not OS. If this is the case, this needs to be made more clear in the text as it bears on the critical issue of whether sTLS are detrimental to patient survival.

7. The comparisons between HGSOC and NSCLC appear to be based on only 10 cases of the latter (ditto for melanoma). If so, this is a grossly inadequate sample size on which to hinge any major conclusions, especially given the molecular and histological diversity of NSCLC. Moreover, the melanoma results are only briefly mentioned, and CRC results are shown in Figs. 3g & h but not mentioned. Any key findings from this section of the paper need to be validated in larger cohorts with care taken to match clinicopathological parameters as best as possible.

8. The B cell depletion experiments in Figs 4d-g require additional controls. It appears that B cells were the only cell type depleted? If so, perhaps the observed effects are attributable to the depletion process itself? It will be necessary to show that depletion of other cell type(s) has no effect such that B cells are uniquely linked to these observations.

9. The murine studies do not add to the paper, as the two models differ not only in terms of

TMB but also tumor line and, most importantly, background strain. Moreover, bona fide TLS are rare in murine models (in general), so further evidence would be required to convince readers that such models faithfully recapitulate the human scenario.

10. Likewise, the DCVAC results do not add to the paper. It appears that responses to this vaccine, for unknown reasons, are stronger in TIL-low cases. To show that these non-responding cases are also TLS-low (as would be expected given they are TIL-low), and to then implicate TLS as detrimental to the vaccine, seems highly speculative.

11. Finally, the paper would be strengthened by providing one or more compelling hypotheses for why intermediate numbers of TLS would be detrimental to anti-tumor immunity, whereas high numbers of TLS would be beneficial. Likewise, why would sTLS be worse than pTLS, and how do the authors square this with other studies showing the opposite?

Minor issues:

1. Lines 78-79 and 434: These statements about TLS are speculative, so the wording should be more guarded.
2. Lines 396-397 are unclear. What does it mean to be a "hub for the spatial (re)distribution" of T cells?

Reviewer #2 (Remarks to the Author): with expertise in TLS, imaging

Article Title: Mature tertiary lymphoid structures are associated with clinically relevant T cell exhaustion in ovarian cancer.

The authors have submitted a well written, clear, and concise article that is relevant to the journal and of interest to the scientific community, as tertiary lymphoid structures are an ongoing area of research in clinical and translational cancer studies. A strength of this paper is that the studies performed in this manuscript generally were performed on large cohort of patient samples. They also performed extensive analyses which include traditional immunohistochemistry (IHC), multiplex immunofluorescent IHC, analyses of tumor mutational burden, Nanostring gene expression profiling, and spatial transcriptomics.

The author's work demonstrates that high-grade serous ovarian carcinoma (HGSOC) contain TLS. Mature TLS, found in HGSOC with high tumor mutations burden (TMB), were associated

with and increased density of intratumoral CD8+ T cells. The authors work found that TLS were predominantly localized in the invasive margins or in the tumor stroma of HGSOc. Furthermore, TLS were observed in 117 out of 180 evaluated tumors, and the majority of these were early TLS or primary follicle-like TLS, while mature secondary follicle like TLS were detected in 19/180 samples (10%). This study found that tumor immune cell densities were correlated with improved disease outcome, but this did not hold true from immune cell densities found in TLS. However, they found that patients with TLS-high (5 TLS or more) tumors had prolonged RFS and OS compared to those with TLS-low tumors. Patients that had eTLS and/or lower than median p- and s-TLS had improved RFS and OS compared to patients with no TLS.

Mature TLS in HGSOc were also associated with higher numbers of terminally exhausted T cells expressing TIM3 and PD1, when compared to CD8 T cells in non-small cell lung carcinoma. They found that B cell depletion promoted terminal exhaustion in CD8+ T cells from freshly isolated HGSOc biopsies, suggesting that B cells are helping in the preservation of T effector functions. Finally, they found that dendritic cell-based immunotherapy was more beneficial in HGSOc patients with reduced expression of mature TLS and terminally exhausted CD8 + T cells.

The data presented in this manuscript represent intriguing findings that are relevant to the field and add to ongoing questions surrounding tertiary lymphoid structures and their prognostic value. A limitation of this study is that a mechanism for these findings were not explored. Questions remain about how B cells limit exhaustion in TLS that are less mature, and what is lost or gained during TLS maturation that then mediates more terminally exhausted T cells in mature TLS of HGSOc, relative to NSCLC. Furthermore, questions remain about why T cells become terminally exhausted, and why they are higher in HGSOc relative to NSCLC. A limitation of this work is that the study did not demonstrate exhaustion of T cells in HGSOc or resistance to ICI treatment in-vitro.

My specific questions/comments to the authors are below:

In this study TLS immune cell populations were evaluated and compared to outside TLS (oTLS) areas in the entire tumor tissues.

Can the authors state how the TLS regions were defined? Specifically, how were the TLS T-cell areas segment/defined away from the rest of the tumor associated cells in these

analyses. How was the T cell border at the TLS site defined?

The study evaluated TLS in non-small cell lung cancer (NSCLC) as well as melanoma. The data identified that HGSOC had fewer exhausted progenitor T cells than NSCLC. The conclusion is that ICI resistance is mediated by an environment dominated by terminally exhausted T cells.

What about the melanoma specimens, were similar findings seen in melanomas as well as NSCLC? These data were not shown. Although some assessments of TLS were performed in melanoma.

A substantial finding in this paper is that HGSOC harbor more terminally exhausted T cells than other cancers that are responsive to ICI, and that mature TLS in HGSOC facilitate an increase in terminally exhausted T cells.

Can the authors demonstrate that the terminally differentiated cells in HGSOC are indeed exhausted/dysfunctional, and more resistant to ICI in-vitro?

Minor points:

In line 274 the sentence states sTLS could be only detected, I believe the sentence should be sTLS could only be detected.

In the abstract a sentence starting in line 46 and ending in line 51 is rather large, it would be easier to read if it could be broken up into two if possible?

Also, this sentence states that CD8+ T cells from HGSOC were compared to those in immunologically hot tumors like non-small cell lung cancer. Can the authors state more concretely that they compared these to non-small cell lung cancer etc.?

In line 316 the sentence states that cluster 3 was lower than median and cluster 4 were higher than median, however the median was not stated here. Can you please state what the median value is. Is the median 5 here too, as stated later for another assay in line 349.

Reviewer #3 (Remarks to the Author): with expertise in TLS, computational

In this manuscript, Kasikova et al. explore the presence, composition and clinical impact of tertiary lymphoid structures in high-grade serous ovarian carcinoma (HGSOG), using a

variety of methods. The topic is very timely, as TLS gather more and more interest in the field, and the authors used state-of-the-art technologies. While I appreciate the huge work this manuscript incorporates, there are some critical technical and methodological flaws that make me question the findings, notably regarding the definition of what is considered a TLS in these tumors.

1) The definition of TLS used in the methods section is problematic for 2 reasons. (1) it does not involve T cells at all. All the current literature on TLS considers them to be lymphoid aggregates with both B cells and T cells. Here, the authors do not analyze CD3 for T cells and therefore, there is a high risk of aggregates being considered as TLS without T cells. (2) the definition is very flexible with the size of such aggregates, going as low as 10 cells. I do not believe 10 cells are sufficient for a group of cells to be called a TLS. Altogether, There is probably a strong inflation of the number of TLSs that are identified throughout the manuscript.

2) Only CD8+ T cells are considered, and not CD4+ T cells, which are usually the main T cell population within TLS, and the main T cell population interacting with TLS B cells in mature TLS are follicular helper T cells, which are also CD8-. I would strongly encourage the authors to also look at CD4+ T cells, or at the very least at a pan-T cell marker such as CD3.

3) PD-1 is not an exhaustion marker. It is expressed by most activated T cells and allows them to enter exhaustion when bound to PD-L1 or PD-L2. In this sense, it does not make sense to label PD-1+TCF1+CD8+ as exhausted. On the contrary, as they have been showed to favor longer survival notably in melanoma (see Sade-Feldman et al., Cell, 2018), it is likely that they would instead have some kind of effector function.

4) The distinction of samples between pTLS + sTLS hi/lo does not seem meaningful. Indeed, a small number of sTLS could be considered a much more mature TLS presence than many pTLS without sTLS. Instead, I would encourage the authors to consider a split between pTLS only and at least one sTLS. The current split would amount to considering pTLS and sTLS as a similar entity, even though sTLS have a much broader array of potential functions in the

TME.

5) The prognostic impact relies on a low number of patients, and is a central point of the manuscript. It needs to be validated on a larger, independent cohort. Perhaps using the TCGA dataset would be useful here, since it has already been divided into clusters with TLS meaning in Fig. 2C.

6) TLS in HGSOE have already been studied in a previous study that is not referenced nor discussed. See Kroeger et al., *Clinical Cancer Research* 2016.

7) Some claims do not seem to be sufficiently backed by data but are rather guesses as to potential mechanisms. These need to be either dimmed or properly backed by mechanistic experiments. Those include:

a. Lines 308-310: It is not sure how relevant the intra-TLS densities are, as the simple presence of TLS, regardless of how dense it is in some cell populations could be more impactful. Also, there is a very low variability for oTLS cell densities, so the authors are at risk of overinterpreting on a few outliers.

b. Lines 396-397: there is no data in the manuscript that can adequately suggest that TLS redistribute any cell population.

c. Lines 408-412: It is unlikely that all PD-1+CD8+ T cells are either TIM3+ or TCF1+, but I guess a number of PD-1+CD8+ T cells are either double positive or double negative for TIM3 and TCF1. Therefore, summing these 2 subpopulations to account for 100% makes is irrelevant.

d. Lines 454-456: A simple colocalization between cells is not enough to claim that B cells influence CD8+ T cells phenotypes.

8) The number of TLS found in the orthotopic mouse model seems extremely high. There are very few reports of any TLS in orthotopic mouse models (it is usually thought that this is because TLS induction takes too long for the span of these experiment), and only a handful of intrinsic models usually develop TLSs. Having an orthotopic mouse models develop

dozens of mature TLS is simply unheard of and makes me question the robustness of the classification of aggregates as TLS.

POINT-BY-POINT REPLY TO REVIEWER N° 1

Reviewer n° 1 commented:

This is an interesting and well-presented paper that addresses the role of TLS in the immune response to high-grade serous ovarian cancer (HGSOC). This is a timely study, given the high level of interest in TLS, B cells, plasma cells (PC) and associated immune cells in the cancer immunotherapy field currently. The authors make the unexpected and counter-intuitive claim that the most mature TLS (i.e., those with secondary follicles, or sTLS) are associated with higher levels of (to use their term) "terminally exhausted" CD8 T cells. They claim that HGSOC differs from NSCLC and melanoma in this regard, and they argue this could explain why HGSOC has a lower response rate to checkpoint blockade. If true, this would imply that TLS are only beneficial to anti-tumor immunity until they reach a certain stage of maturation, after which they become detrimental. This would have important implications for immunotherapies designed to induce TLS, and for cancer immunotherapy in general.

Strengths:

1. Important and timely topic
2. Relatively larger HGSOC cohorts
3. State of the art methods such as spatial transcriptomics

Our response: We are indebted to Reviewer #1 of *Nature Communications* for the enthusiastic evaluation of our work. We have undertaken a major revision of the paper involving (1) a thorough re-analysis of available data including data on TLS structures and their impact on TME using novel 8-plex immunofluorescence data, (2) inclusion of additional validation results from new patient cohort and experimental models that we believe considerably strengthened our conclusions, as detailed herein.

And raised these major critiques:

1. In general, the paper overlooks the considerable body of work by others regarding B cells, PCs, TLS and antibody responses in HGSOC (e.g., groups like Conejo-Garcia, Nelson, Balkwill, Shulman). Consequently, several of the key results are in fact confirmatory yet are not presented as such. The paper could be improved by summarizing this prior work in the introduction and better emphasizing the novel findings of the present study.

Our response: We apologize for the lack of details summarizing available work regarding B cells, plasma cells, TLS and antibody response in HGSOC in the original version of the manuscript. Inspired by the suggestion from Reviewer #1, we have amply discussed in more detail available findings from others in the introduction and discussion of the revised version of manuscript.

2. The authors' finding that TLS are associated with higher TMB appears to contradict a prior report from Kroeger et al using TCGA data (PMID: 26763251); this discrepancy should be discussed. Moreover, given the narrow range of TMB values obtained (from 0-10 per Mb, median 2.2), and given that only 1-10% of mutations are thought to give rise to neoantigens, it seems unlikely (and hence a larger burden of proof) that TMB can explain higher numbers of TLS in HGSOC. This important issue should be validated in an independent cohort and discussed.

Our response: We thank Reviewer #1 for bringing up this point to our attention. TMB was analyzed using TrueSight ONCO500 DNA kit (Illumina) in tumor samples of 80 patients with HGSOC (Study group 2) collected from 2 independent clinical sites (n=27; n=53). TMB was presented as the number of somatic mutations per megabase. TMB in tumor samples from both study groups ranged from 0 to 10 mut/Mb with a median value of 2.2 mut/Mb which is in line with previously published data (Sha et al, Nature Discovery, 2020) and further confirmed on 304 ovarian cancer patients from the TCGA database (Fig. 1A of this rebuttal letter). This implies that the “conventional” 10 mut/Mb cut-off value cannot be applied to stratify ovarian cancer patients into 2 groups, which explain our choice to use a median cutoff approach for further correlation of TMB with TLS presence (Figure 5 of revised version of manuscript). Inspired by constructive Reviewer #1 comment, we provide separate analyses on somatic mutations profile (Fig. 1B of this rebuttal letter); impact of TMB on anti-tumor immunity development using Nanostring analyses and immunofluorescence staining (Fig. 1C-E of this rebuttal letter); and early TLS (eTLS) and mature TLS (mTLS) formation in both independent patient cohorts (Fig. 1D, E of this rebuttal letter) while also implementing novel TLS-like 8-plex immunofluorescence panel for eTLS and mTLS

classification (CD8, CD4, CD20, CD21, CD23, DC-LAMP and GZMB). Our separate analyses on independent patient cohorts further confirm our previous findings documenting the impact of higher than median TMB on anti-tumor immunity development and eTLS and mTLS formation (Fig. 1C-E of this rebuttal letter). In line with this notion, we have confirmed the positive correlation between high TMB and homologous recombination deficiency (HRD) status and TLS-like gene signature (*CCL2, CCL3, CCL4, CCL5, CCL8, CCL18, CCL19, CCL21, CXCL9, CXCL10, CXCL11, CXCL13*) (as previously published *Meylan et al, Immunity, 2022*) on 304 HGSOE samples from the TCGA public database (Fig. 1F of this rebuttal letter). Moreover, we observed a trend for positive correlation between TMB and the TLS-chemokine gene signature across 12 cancer types from the TCGA public database (Fig. 1G of this rebuttal letter). However, we failed to observe a similar impact of individual somatic mutations, such as *BRCA1* mutations, on TLS formation (determined by immunofluorescence staining) and activation in tumor samples from study cohort 2 (Fig. 1H of this rebuttal letter).

To experimentally dissect the link between TMB level and TLS development, we harnessed a mouse model of ovarian cancer (ID8) cells that is syngeneic to C57BL/6J mice and exhibit a low number of somatic mutations and limited TMB (Fig. 1I, J of this rebuttal letter). Since contrarily to expectations *Trp53^{-/-}Brca1^{-/-}* ID8 cells as generated by CRISPR/Cas9 gene editing, kindly provided by prof. Ian McNeish (*Walten et al, CCR, 2016*) failed to exhibit a significant increase in overall TMB (Fig. 1I, J of this rebuttal letter), we selected another mouse model of ovarian cancer that is also syngeneic to C57BL/6J mice but exhibits high TMB (*Brca1^{-/-}Trp53^{-/-}, Myc, Hras* SO1 cells), kindly provided by prof. Sandra Orsulic (*Beach et al, Oncotarget, 2016*) (Fig. 1I, J of this rebuttal letter). The TMB^{Lo} ID8 and TMB^{Hi} SO1 mouse ovarian cancer cell lines were used to generate tumors in immunocompetent female syngeneic C57BL/6J mice for subsequent TLS analyses (Fig. 1K of this rebuttal letter). Immunofluorescence analysis of tumors collected 20 and 60 days after intraperitoneal implantation of cancer cells into C57BL/6J mice demonstrated that the TMB^{Hi} SO1 tumors contained a higher proportion of advanced lymphoid aggregates, the most mature of which did not form in the TMB^{Lo} ID8 parental tumors (Fig. 1L of this rebuttal letter).

Taken together, our data indicate that increased level of TMB correlate with formation of TLS structures as demonstrated in 3 independent cohorts of HGSOE patients, experimental mice studies and supported by previous observation in experimental breast carcinoma models (Hollern et al, *Cell, 2019*). These analyses were included in Figure 5, 6 and Supplemental Figure 10 of the revised version of manuscript.

Figure 1. TMB positively correlates with formation of mature TLS structures in HGSOC. (A) Violin plot showing the somatic mutations prevalence in 2 independent HGSOC study cohorts (Study group 2 and TCGA

public database) and NSCLC cohort (TCGA). **(B)** Oncoplot showing the profile of somatic mutations in 2 independent cohorts of 27 and 53 HGSOc patient samples, further annotated by TMB determined by TrueSighOnco500 and TLS numbers as determined by immunofluorescence staining. **(C)** Unsupervised hierarchical clustering of gene signatures related to immune populations (orange), immune functions (green) and immune phenotype (purple), as determined by PanCancer Immune Profiling from Nanostring and further annotated by TLS and TMB in 2 independent cohorts of 27 and 53 HGSOc patients. **(D, E)** Density of CD8⁺ T cells, CD20⁺ B cells, early TLS (eTLS) and mature TLS (mTLS) numbers in TMB low (TMB^{Lo}) and high (TMB^{Hi}) tumors from 2 independent cohorts of HGSOc patients (study group 2). Box plots: lower quartile, median, upper quartile; whiskers, minimum, maximum. Statistical significance was calculated by the Mann-Whitney test. *p* values are indicated. **(F)** Correlation between the TLS gene signature (*CCL2, CCL3, CCL4, CCL5, CCL8, CCL18, CCL19, CCL21, CXCL9, CXCL10, CXCL11, CXCL13*) and somatic mutations and homologous recombination deficiency (HRD) score in 304 HGSOc patients from the TCGA public database (Study group 4). R, Pearson correlation coefficient. **(G)** TLS gene signature and TMB score in low grade gliomas (LGG), glioblastoma (GBM), uterine carcinosarcoma (UCS), kidney renal clear cell carcinoma (KIRC), kidney renal papillary cell carcinoma (KIRP), uterine corpus endometrial carcinoma (UCEC), ovarian carcinoma (OV), rectum adenocarcinoma (READ), colon adenocarcinoma (COAD), lung adenocarcinoma (LUAD), lung squamous cell carcinoma (LUSC), skin cutaneous melanoma (SKCM) patients from the TCGA public database. **(H)** eTLS and mTLS numbers as determined by immunofluorescence staining in *BRCA1*^{WT} and *BRCA1*^{-/-} tumor samples as determined by TrueSighOnco500 in 80 HGSOc tumor samples (Study group 2). **(I)** Barplots showing single-nucleotide variants (SNVs) and **(J)** the somatic mutations prevalence (mutations per megabase) in ID8, *Brcal*^{-/-} *Trp53*^{-/-} ID8 and *Brcal*^{-/-} *Trp53*^{-/-}, *Myc, Hras* SO1 cell lines. Statistical significance was calculated by the Mann-Whitney test. *p* values are indicated. **(K, L)** Representative immunostaining for CD4, CD8 and CD20 and box plot showing density of lymphoid aggregates within TMB^{Lo} and TMB^{Hi} derived mouse ovarian tumors. Scale bars 1 and 2.5 mm. Box plots: lower quartile, median, upper quartile; whiskers, minimum, maximum. Statistical significance was calculated by Mann-Whitney test. *p* values are indicated.

3. The authors use terms like "exhausted" and "terminally exhausted" based on molecular phenotypes alone, without showing any functional data. These are still evolving and controversial concepts in the field. Therefore such terms could be mentioned as hypotheses in the Discussion, but should not be used as definitive nomenclature in the Results.

Our response: We apologize for the strong phrasing and lack of details focusing on functional properties of so called “progenitor” and “terminally” exhausted T cells in the original version of the manuscript. This terminology classifying TCF1⁺PD1⁺ cells as progenitors and TIM3⁺PD1⁺ cells as terminally exhausted CD8⁺ T cells was adapted based on previously published studies: Wang *et al*, *Nature*, 2023; Miller *et al*, *Nature Immunology*, 2019; Koh *et al*, *EJC*, 2022; Connolly *et al*, *Science immunology*, 2021. As per the suggestion of Reviewer #1, we have avoided using these terms and instead objectively reported cell phenotype throughout the results section. We have instead alluded to the possibility that TCF1⁺PD1⁺ and TIM3⁺PD1⁺ cells may represent progenitor and terminally exhausted CD8⁺ T cells, respectively, in the discussion of the paper.

4. On a related note, the authors assume "exhausted" and "terminally exhausted" phenotypes are detrimental to anti-tumor immunity. This is an oversimplified view. There is considerable evidence that T cells with a seemingly exhausted phenotype are in fact associated with favorable properties and good outcomes in HGSOE (see Duraiswamy PMID: 34739845; Laumont PMID: 33963000) and other cancers. Thus, these issues should be presented and discussed with more objective, nuanced language.

Our response: We agree with Reviewer #1 that T cells under distinct exhausted phenotypes contribute differently to the clinical outcome. Thus, besides scientifically relevant studies mentioned by Reviewer #1 mainly focusing on clinical relevance of entire PD1⁺CD8⁺ T cells population, there is accumulating literature documenting distinct role of TCF1⁺PD1⁺ and TIM3⁺PD1⁺ subsets in solid carcinomas. Notably, numerous studies link TCF1⁺PD1⁺ cells to favorable disease outcome and response to CPIs in: malignant melanoma (Siddiqui *et al*, *Immunity*, 2019; Miller *et al*, *Nature Immunol*, 2019; Sade-Feldman, *Cell*, 2018); triple-negative breast carcinoma (Wang *et al*, 2023, *Nature*); NSCLC (Koh *et al*, *EJC*, 2022); HNSCC (Wang *et al*, *IJOS*, 2022); esophageal carcinoma (Ma *et al*, *Cancer Science*, 2012); B cell lymphoma (Tabanelli *et al*, *Blood Advances*, 2022) and ovarian carcinoma (Anadon *et al*, *Cancer Cell*, 2022). Conversely, a high prevalence of TIM3⁺PD1⁺ T cells correlate with poor prognosis in: triple negative breast cancer (Egelstan *et al*, *JCI insight*, 2022); hepatocellular carcinoma (Ma *et al*, *JITC*, 2019); colon cancer (Sahuishi *et al*, *JEM*, 2010); large B cell lymphoma (Roussel *et al*, *Blood Advances*, 2020) and NSCLC (Thommen *et al*, *Nat Med*, 2018). In addition, in line with previous findings from Duraiswamy *et al* (*Cancer Cell*, 2021), here we demonstrate that a high density of PD1⁺CD8⁺ T cells also correlate with favorable disease outcome ($p=0.001$) in HGSOE patients using immunofluorescence staining (Fig. 2A, B of this rebuttal letter). However, further phenotypic distribution of PD1⁺CD8⁺ cells into TCF1⁺PD1⁺ and TIM3⁺PD1⁺

demonstrate the former to be associated with favorable outcome and latter with poor outcome (OS: TCF1⁺PD1⁺: $p=0.042$; OS: TIM3⁺PD1⁺: $p=0.045$) as determined by immunofluorescence staining (Fig. 2C, D of this rebuttal letter). We have discussed these findings in more objective and nuanced language in the revised version of manuscript. These analyses were included in Supplemental Figure 7D, E of the revised version of manuscript.

Figure 2. The clinical relevance of distinct T cells phenotypes in HGSOc. (A) Representative image of PD1⁺CD8⁺, TCF1⁺PD1⁺CD8⁺ and TIM3⁺PD1⁺CD8⁺ T cells subset analyzed by immunofluorescence staining. (B-D) Overall survival (OS) of HGSOc patients based on median stratification of PD1⁺CD8⁺ T cells (B), TCF1⁺PD1⁺CD8⁺ (C) and TIM3⁺PD1⁺CD8⁺ (D) from study cohort 1. Survival curves were estimated by Kaplan-Meier method, and differences between groups were evaluated using log-rank test. Number of patients at risk and p values are reported.

5. The authors' 4 clusters from RNA-seq should be better described to give readers a general sense of how they differ, without needing to study the heatmap. In general, the sections describing these clusters could be made more clear and succinct.

Our response: We thank Reviewer #1 for bringing the cumbersome presentation of RNAseq data in Figure 2A to our attention. We set to harness RNAseq to compare the gene expression profile of 50 tumor samples from Study Group 1 containing no TLS (Cluster 1), early TLS (eTLS) only (Cluster 2) and containing eTLS and mTLS

(Cluster 3) (Fig. 3A of this rebuttal letter). The number of eTLS and mTLS were determined using novel well-defined 8-plex immunofluorescence panel (CD8, CD4, CD20, CD21, CD23, DC-LAMP and GZMB) including marker for CD4⁺ T cells, follicular DCs, B cells and mature DCs. We believe that the revised version of heatmap presented in Figure 2A of revised version of manuscript is present in more clear and succinct manner.

Figure 3. The impact of TLS maturation on development of antitumor immunity. (A) Supervised hierarchical clustering of tumor samples of HGSOc patients (Study group 1) with no TLS development (CL1), only early TLS (eTLS) development (CL2) and eTLS and mature TLS (mTLS) development (CL3) determined by immunofluorescence staining, based on the expression of 100 genes classified into clusters related to B cells, cytotoxicity, dendritic cells (DCs), immune cells, immunosuppression, natural killers (NK) cells, T cell activation, TEM, T_H1 and T_H2 signatures as determined on RNA sequencing data from Study Group 1.

6. Lines 350-357 and Fig 2g,h: It is unclear whether these data are based on IHC/IF enumeration of TLS or RNA-seq based clusters. The wording on lines 350-351 is particularly unclear in this regard. Moreover,

it appears from Fig 2h that sTLS hi patients have worse RFS, but not OS. If this is the case, this needs to be made clearer in the text as it bears on the critical issue of whether sTLS are detrimental to patient survival.

Our response: Inspired by comments provided by the Editor and all Reviewers of *Nature Communications*, we have implemented a novel well-defined 8-plex immunofluorescence panel (CD8, CD4, CD20, CD21, CD23, DC-LAMP and GZMB) including marker for CD4⁺ T cells, follicular DCs, B cells and mature DCs as recently used within key studies in the field (*Vanhersecke et al, Nat Cancer, 2021; Helmink et al, Nature, 2020; Petitprez et al, Nature, 2020*) to optimally distinguish between early TLS (eTLS) and mature TLS (mTLS) (with GC formation and CD21⁺CD23⁺ fDCs) (Fig. 4A of this rebuttal letter). The quantification of eTLS and mTLS was done in HALO software with manual depiction of TLS based on described criteria and further confirmed by two independent pathologists from the team in all tumor samples from 3 independent study groups to strengthen our data provided within entire manuscript. The novel TLS classification was used for revised Kaplan Meier survival analyses on 210 HGSOc patients from study group 1 and 2 (Fig. 4B-F of this rebuttal letter). We have confirmed the positive prognostic role of TLSs in HGSOc, using median separation between eTLS+mTLS high and low patients (RFS: $p=0.002$; OS: $p=0.016$; Fig. 4B of this rebuttal letter). However, stratifying patients in the 3 clusters reflecting, Cluster 1 no TLS present, Cluster 2 only eTLSs present and Cluster 3 both eTLS and mTLSs present in the TME, we found that cluster 2 (patients with eTLS only) had significantly improved RFS and OS as compared to Cluster 1 (patients with no TLS) (Fig. 4C of this rebuttal letter). Importantly, the same did not hold true for Cluster 3 (patients with eTLS and mTLS development; Fig. 4C of this rebuttal letter). These findings were largely confirmed by univariate COX regression analyses (Table 1 of revised version of manuscript). Similar findings were also obtained when we analyzed the two cohorts of patients with HGSOc that make up Study Group 1 independently from each other (Fig. 4D, E of this rebuttal letter) and from Study Group 2 (Fig. 4F of this rebuttal letter). To further validate our findings, we have collected an independent cohort of 40 HGSOc patients from University of Basel in kind collaboration with prof. Viola Heinzlmann-Schwarz, indicated as Study Group 3 within the revised version of manuscript. Supporting our previous observation, the formation of mTLS was observed only in 10% of tumor samples from respective cohort. The formation of both eTLS and mTLS correlate with favorable progression free survival (PFS) in HGSOc from study cohort 3 (Fig. 4G of this rebuttal letter). Moreover, in line with our previous findings, we observed that Cluster 2 patients had significantly improved RFS as compared to Cluster 1. Importantly, the same did not hold true for Cluster 3 (Fig. 4H of this rebuttal letter).

Taken together, in line with previously published data in multiple clinical cohorts, TLS formation in HGSOC correlates with favorable prognosis. However, the formation of mature TLS in TME correlate with unfavorable prognosis, which might be associated with higher prevalence of co-inhibitory molecules and exhausted phenotype of T cells within TME of respective patients as comprehensively documented in Figure 3 of revised version of manuscript. Provided analyses were included in Figure 1, 2 and Supplemental Figure 6 of revised version of manuscript.

A

B

Cohort 1 (HGSOc; n=123)

C

Cohort 2 (HGSOc; n=60)

D

Cohort 3 (HGSOc; n=27)

E

Cohort 4 (HGSOc; n=40)

F

G

H

Figure 4. The clinical relevance of TLS formation and maturation in 4 independent cohorts of HGSOc patients. (A) Representative image of immunofluorescence of CD4, CD8, CD20, CD21, CD23, DC-LAMP and GZMB staining (immunofluorescence panel 1). Scale bars 10, 100 and 500 μ m. **(B, C)** Relapse-free survival

(RFS) and overall survival (OS) of 210 HGSOC patients (Study Group 1+2) based on median stratification of total TLS **(B)** and **(C)** based on stratification into 3 clusters with no TLS development (CL1), only eTLS development (CL2) and both eTLS and mTLS development (CL3). **(D-F)** RFS and OS of patients from individual patients' cohorts based on stratification into 3 clusters. **(G, H)** RFS and OS of 40 HGSOC patients from study group 3 (Swiss cohort) based on median stratification of total TLS **(G)** and based on stratification into 3 clusters **(H)**. Survival curves were estimated by the Kaplan-Meier method, and differences between groups were evaluated using log-rank test. Number of patients at risk and *p* values are reported.

7. The comparisons between HGSOC and NSCLC appear to be based on only 10 cases of the latter (ditto for melanoma). If so, this is a grossly inadequate sample size on which to hinge any major conclusions, especially given the molecular and histological diversity of NSCLC. Moreover, the melanoma results are only briefly mentioned, and CRC results are shown in Figs. 3g & h but not mentioned. Any key findings from this section of the paper need to be validated in larger cohorts with care taken to match clinicopathological parameters as best as possible.

Our response: We fully agree with Reviewer #1 that the data comparing TLS frequencies and T cell phenotypes in NSCLC and HGSOC required validation in larger patient cohorts. Thus, eTLSs vs mTLSs formation determined by TLS-8-plex immunofluorescence staining was demonstrated on HGSOC patients from study group 1 and 2 (n=210) and stage III adenocarcinoma NSCLC patients (n=40) (Fig. 5A, B of this rebuttal letter). Moreover, the impact of TLS on detailed T cell phenotype within TLS structures and entire TME determined by 12-plex immunofluorescence analyses was demonstrated on stage III HGSOC patients (n=20) and stage III adenocarcinoma NSCLC patients (n=20) (Fig. 5C-E of this rebuttal letter). Adding more patients in the final analyses supports our previous observation documenting significantly higher frequency of eTLS and mTLS in NSCLC as compared to HGSOC patients (Fig. 5B of this rebuttal letter). In addition, the formation of eTLS and mainly mTLS in HGSOC and NSCLC was associated with high density of TCF1⁺PD1⁺CD8⁺ T cells within these structures (Fig. 5C, D of this rebuttal letter). Finally, mTLS development in HGSOC and NSCLC positively correlated with increased frequency of TCF1⁺PD1⁺CD8⁺ T cells and not with TIM3⁺PD1⁺CD8⁺ T cells within entire TME as documented by multiplex immunofluorescence staining (Fig. 5E of this rebuttal letter), which also

largely reflect higher prevalence of TCF1⁺PD1⁺CD8⁺ T cells within NSCLC with significantly higher frequency of mTLS formation. These analyses were included in Figure 3 of the revised version of manuscript.

Figure 5. TLS formation and maturation impact the T cell phenotype in HGSOC and NSCLC. (A, B) Representative images (A) and violin plot (B) showing the density of eTLS and mTLS in HGSOC (n=210) and

NSCLC (n=40). Statistical significance was calculated by the Mann-Whitney test. *p* values are indicated. **(C, D)** Density of TCF1⁺PD1⁺CD8⁺, TIM3⁺PD1⁺CD8⁺ and TIM3⁺PD1⁻CD8⁺ cells in entire TME, in early TLS (eTLS) and mature TLS (mTLS) of stage III tumor samples of HGSOC (no=20, study group 1) and stage III NSCLC adenocarcinoma (no=20, study group 6) as determined by immunofluorescence. Box plots: lower quartile, median, upper quartile; whiskers, minimum, maximum. Statistical significance was calculated by the Mann-Whitney test. *p* values are indicated. **(E)** Correlation between number of mTLS and density of TCF1⁺PD1⁺CD8⁺ and TIM3⁺PD1⁺CD8⁺ T cells in HGSOC (n=20, study group 1) and NSCLC (n=20). R, Pearson correlation coefficient.

8. The B cell depletion experiments in Figs 4d-g require additional controls. It appears that B cells were the only cell type depleted? If so, perhaps the observed effects are attributable to the depletion process itself? It will be necessary to show that depletion of other cell type(s) has no effect such that B cells are uniquely linked to these observations.

Our response: Inspired by the constructive critique of Reviewer #1, the impact of CD4⁺ T cells and CD14⁺ cells depletion on tumor-infiltrating lymphocytes (TILs) isolated from freshly resected HGSOCs (n=5) was determined by flow cytometry (Fig. 6A, B of this rebuttal letter). In this setting, CD4⁺ T cells and CD14⁺ cells depletion did not significantly impact the percentage of TIM3⁺PD1⁺CD8⁺ T cells and/or TIM3⁻PD1⁺CD8⁺ T cells (Fig. 6A, B of this rebuttal letter). However, supporting our previous observation in HGSOC, B-cell depletion decreased the percentage of TIM3⁺PD1⁺CD8⁺ T cells while increasing the abundance of TIM3⁻PD1⁺CD8⁺ T cells isolated from freshly resected TLS^{Hi} stage III adenocarcinoma NSCLC (n=7) (Fig. 6C, D of this rebuttal letter). Taken together, our findings indicate that B cells depletion from TLS^{Hi} HGSOC and NSCLC patients severely impact the phenotypic properties of TILs toward ICIs-nonresponsive TIM3⁺PD1⁺ phenotype. Thus, our findings support previous studies, documenting the impact of B cells on both cancer prognosis and response to ICIs (*Fridman et al, JEM, 2020; Laumont et al, NRC, 2022; Kim et al, CCR, 2022; Petitprez et al, Nature, 2020*). These novel analyses were included in Figure 4 and Suppl. Figure 9 of revised version of manuscript.

Figure 6. The impact of intratumoral CD4⁺, CD14⁺ cells and CD19⁺CD20⁺ B cells on CD8⁺ T cells phenotype in HGSOc and NSCLC. (A, B) Representative dot plot and flow cytometry analyses for frequency of PD1⁺TIM3⁺ and PD1⁺TIM3⁻ TCF1⁺CD8⁺ T cells before and after CD4⁺ cells depletion (A) and CD14⁺ cells depletion (B) from native HGSOc tumor tissue (n=5). (C, D) Representative dot plot (C) and flow cytometry analyses (D) for frequency of TIM3⁺PD1⁺ and TIM3⁻PD1⁺CD8⁺ T cells before and after CD19⁺CD20⁺ B cells depletion from native stage III NSCLC tumor tissues (n=7). Statistical significance was calculated by the Mann-Whitney test. *p* values are indicated.

9. The murine studies do not add to the paper, as the two models differ not only in terms of TMB but also tumor line and, most importantly, background strain. Moreover, bona fide TLS are rare in murine models (in general), so further evidence would be required to convince readers that such models faithfully recapitulate the human scenario.

Our response: We agree with the critique raised by Reviewer #1 regarding tumor cell lines and background strain used in experimental *in vivo* models. To experimentally dissect the link between TMB level and TLS development, we harnessed the C57BL/6J syngeneic ID8 experimental mouse model with overall low number of somatic mutations and low TMB (Fig. 7A, B of this rebuttal letter). CRISPR/Cas9 gene editing generated *Trp53^{-/-}Brca1^{-/-}* ID8 cells (Walton *et al*, *Can Res*, 2016) failed to exhibit a significant increase in TMB as compared to parental ID8 (Fig. 7A, B of this rebuttal letter). Thus, we used the C57BL/6J syngeneic genetically engineered mouse ovarian cancer cell line SO1 (genotype *Brca1^{-/-}, Trp53^{-/-}, Myc, Hras*) (Beach *et al*, *Oncotarget*, 2016), which has a high number of somatic mutations and high overall TMB (Fig. 7A, B of this rebuttal letter). The TMB^{Lo} ID8 and TMB^{Hi} SO1 mouse ovarian cancer cell lines were used to generate tumors in immunocompetent female syngeneic C57BL/6J mice for subsequent TLS analyses (Fig. 7C of this rebuttal letter). Immunofluorescence analysis of tumors collected 20 and 60 days after intraperitoneal implantation of cancer cells into C57BL/6J mice demonstrated that the TMB^{Hi} SO1 tumors contained a higher proportion of advanced lymphoid aggregates (LAs), the most mature of which did not form in the TMB^{Lo} ID8 parental tumors (Fig. 7D-F of this rebuttal letter). Similar to our findings in human tumor samples, mouse TMB^{Hi} tumors with high advanced LAs frequency were associated with increased density of effector T cells with TCF1⁺PD1⁺CD8⁺ T, TIM3⁺PD1⁺CD8⁺ and TIM3⁺PD1⁺CD8⁺ T cell phenotypes as compared to TMB^{Lo} tumors (Fig. 7G of this rebuttal letter). Nevertheless, in line with the findings in human HGSO, the effector PD1⁺CD8⁺ preferentially differentiated into TIM3⁺PD1⁺CD8⁺ T cells in the TMB^{Hi} ovarian cancer model (Fig. 7G of this rebuttal letter).

Next, we experimentally dissected the impact of TMB and TLS development on response rate to anti-PD1 therapy (Fig. 7C of this rebuttal letter). Anti-PD1 therapy provided significant survival benefit in the SO1-derived experimental model ($p=0.012$; Fig. 7H of this rebuttal letter) as compared to the ID8-derived model (Fig. 7I of this rebuttal letter). Moreover, anti-PD1 therapy in the TMB^{Hi} SO1 model resulted in significant increase in TCF1⁺PD1⁺CD8⁺ T cells as compared to control treatment group, although the frequency of TIM3⁺PD1⁺CD8⁺ and TIM3⁺PD1⁺CD8⁺ T cells subsets remained unchanged as determined by flow cytometry (Fig. 7J, K of this rebuttal letter). Supporting these findings, we observed higher frequency of IFNG⁺ and Ki67⁺ CD8⁺ T cells after

anti-PD1 therapy as compared to control treatment group in the TMB^{Hi} SO1 experimental model (Fig. 7L of this rebuttal letter). These novel analyses were included in Figure 6 of the revised version of manuscript.

Figure 7. The efficacy of anti-PD1 therapy in TMB^{Lo} and TMB^{Hi} mouse ovarian cancer models. (A, B) Barplots showing single-nucleotide variants (SNVs) and the somatic mutations prevalence (mutations per megabase) in parental ID8, *Trp53*^{-/-} *Brca1*^{-/-} ID8, and *Trp53*^{-/-} *Brca1*^{-/-}, *Myc*, *Hras* SO1 C57BL/6J syngeneic mouse ovarian cancer cell lines. Statistical significance was calculated by the Mann–Whitney test. *p* values are indicated. (C) Experimental design for the analysis of lymphoid aggregate development and efficacy of anti-PD1 therapy in

TMB^{Lo} ID8 and TMB^{Hi} SO1 experimental syngeneic mice models. **(D-F)** Representative immunostaining for CD4, CD8 and CD20 **(D, E)** and a box plot **(F)** showing density of lymphoid aggregates within TMB^{Lo} ID8 **(D)** and TMB^{Hi} SO1 **(E)** ovarian tumors **(F)**. Scale bars 1 and 2.5 mm. Box plots: lower quartile, median, upper quartile; whiskers, minimum, maximum. Statistical significance was calculated by the Mann–Whitney test. *p* values are indicated. **(G)** Box plot showing percentage of TCF1⁺PD1⁺CD8⁺, TIM3⁺PD1⁺CD8⁺ and TIM3⁻PD1⁺CD8⁺ T cells in tumor samples of TMB^{Lo} ID8 and TMB^{Hi} SO1 experimental models. Box plots: lower quartile, median, upper quartile; whiskers, minimum, maximum. Statistical significance was calculated by the Mann–Whitney test. *p* values are indicated. **(H, I)** Overall survival (OS) of TMB^{Hi} SO1 **(H)** and TMB^{Lo} ID8 **(I)** experimental models after anti-PD1 therapy. Survival curves were estimated by the Kaplan-Meier method and differences between groups were evaluated using log-rank test. **(J-L)** Representative dot plot **(J)** and flow cytometry analyses for percentages of TCF1⁺PD1⁺CD8⁺, TIM3⁺PD1⁺CD8⁺ and TIM3⁻PD1⁺CD8⁺ T cells **(K)** and IFNG⁺, CD107⁺ and Ki67⁺CD8⁺ T cells **(L)** in tumor samples of the TMB^{Hi} SO1 experimental model in the presence or absence of anti-PD1 therapy. Box plots: lower quartile, median, upper quartile; whiskers, minimum, maximum. Statistical significance was calculated by the Mann–Whitney test. *p* values are indicated.

10. Likewise, the DCVAC results do not add to the paper. It appears that responses to this vaccine, for unknown reasons, are stronger in TIL-low cases. To show that these non-responding cases are also TLS-low (as would be expected given they are TIL-low), and to then implicate TLS as detrimental to the vaccine, seems highly speculative.

Our response: We agree with Reviewer #1 that the current data presented in such a way do not add to the paper due to lack of mechanistic studies behind. Thus, inspired by Reviewer #1 and other Reviewers comments, we decided to remove data related to DC-based studies from the revised version of manuscript.

11. Finally, the paper would be strengthened by providing one or more compelling hypotheses for why intermediate numbers of TLS would be detrimental to anti-tumor immunity, whereas high numbers of TLS would be beneficial. Likewise, why would sTLS be worse than pTLS, and how do the authors square this with other studies showing the opposite?

Our response: Please, see our response to Main Point #6.

POINT-BY-POINT REPLY TO REVIEWER N° 2

Reviewer n° 2 commented:

The authors have submitted a well written, clear, and concise article that is relevant to the journal and of interest to the scientific community, as tertiary lymphoid structures are an ongoing area of research in clinical and translational cancer studies. A strength of this paper is that the studies performed in this manuscript generally were performed on large cohorts of patient samples. They also performed extensive analyses which include traditional immunohistochemistry (IHC), multiplex immunofluorescent IHC, analyses of tumor mutational burden, Nanostring gene expression profiling, and spatial transcriptomics. The author's work demonstrates that high-grade serous ovarian carcinoma (HGSOC) contains TLS. Mature TLS, found in HGSOC with high tumor mutations burden (TMB), were associated with and increased density of intratumoral CD8⁺ T cells. The authors work found that TLS were predominantly localized in the invasive margins or in the tumor stroma of HGSOC. Furthermore, TLS were observed in 117 out of 180 evaluated tumors, and the majority of these were early TLS or primary follicle-like TLS, while mature secondary follicle like TLS were detected in 19/180 samples (10%). This study found that tumor immune cell densities were correlated with improved disease outcome, but this did not hold true from immune cell densities found in TLS. However, they found that patients with TLS-high (5 TLS or more) tumors had prolonged RFS and OS compared to those with TLS-low tumors. Patients that had eTLS and/or lower than median p- and s-TLS had improved RFS and OS compared to patients with no TLS. Mature TLS in HGSOC were also associated with higher numbers of terminally exhausted T cells expressing TIM3 and PD1, when compared to CD8 T cells in non-small cell lung carcinoma. They found that B cell depletion promoted terminal exhaustion in CD8⁺ T cells from freshly isolated HGSOC biopsies, suggesting that B cells are helping in the preservation of T effector functions. Finally, they found that dendritic cell-based immunotherapy was more beneficial in HGSOC patients with reduced expression of mature TLS and terminally exhausted CD8⁺ T cells. The data presented in this manuscript represent intriguing findings that are relevant to the field and add to ongoing questions surrounding tertiary lymphoid structures and their prognostic value. A limitation of this study is that a mechanism for these findings were not explored. Questions remain about how B cells limit exhaustion in TLS that are less mature, and what is lost or gained during TLS maturation that then mediates more terminally exhausted T cells in mature TLS of HGSOC, relative to NSCLC. Furthermore, questions remain about why T cells become terminally exhausted, and why they are higher in

HGSOC relative to NSCLC. A limitation of this work is that the study did not demonstrate exhaustion of T cells in HGSOC or resistance to ICI treatment in-vitro.

Our response: We are indebted to the Reviewer #2 for the globally positive comments on our article and bringing to our attention its main weaknesses. Inspired by the constructive comments from the Editor and Reviewers of *Nature Communications*, we have performed additional experiments involving the re-analyses of available data as well as the addition of numerous novel findings, which we believe might provide more details regarding mechanism of our findings which will further strengthened our conclusions, as detailed below.

1. In this study TLS immune cell populations were evaluated and compared to outside TLS (oTLS) areas in the entire tumor tissues. Can the authors state how the TLS regions were defined? Specifically, how were the TLS T-cell areas segment/defined away from the rest of the tumor associated cells in these analyses. How was the T cell border at the TLS site defined?

Our response: Inspired by overall constructive comments provided from Editor and Reviewers of *Nature Communications*, we have implemented a novel well-defined 8-plex immunofluorescence panel (CD8, CD4, CD20, CD21, CD23, DC-LAMP and GZMB) including marker for CD4⁺ T cells, follicular DCs, B cells and mature DCs as recently used within key studies in the field (*Vanhersecke et al, Nat Cancer, 2021; Helmink et al, Nature, 2020; Petitprez et al, Nature, 2020*) to optimally distinguish between eTLS and mTLS (with GC formation and CD21⁺CD23⁺ fDCs) and to set T cell border between TLS and rest of tumors (Fig. 4A and Fig. 8 A, B of the rebuttal letter). The quantification of TLS areas was done in HALO software with manual depiction of TLS based on described criteria and further confirmed by two independent pathologists from the team (Fig. 8C of this rebuttal letter) in all tumor samples from 4 independent study cohorts to strengthen our data provided within entire manuscript.

Figure 8. Spatial distribution of early TLS (eTLS) and mature TLS (mTLS). (A, B) Representative images of eTLS and mTLS using immunofluorescence of CD4, CD8, CD20, CD21, CD23, DC-LAMP and GZMB staining. (C) Representative image of TLS areas quantification in HALO software.

2. The study evaluated TLS in non-small cell lung cancer (NSCLC) as well as melanoma. The data identified that HGSOC had fewer exhausted progenitor T cells than NSCLC. The conclusion is that ICI resistance is mediated by an environment dominated by terminally exhausted T cells. What about the melanoma specimens, were similar findings seen in melanomas as well as NSCLC? These data were not shown. Although some assessments of TLS were performed in melanoma.

Our response: We thank Reviewer #2 for bringing this point to our attention. We fully agree with Reviewer #2 that the data comparing TLS frequencies and T cell phenotypes in NSCLC and HGSOC require validation in larger patient cohorts. As we need to prioritize and to comprehensively focus on large cohorts with matched clinicopathological parameters, we decided to compare stage III HGSOC with stage III adenocarcinoma NSCLC patients. eTLSs vs mTLSs formation determined by TLS-8-plex immunofluorescence staining was demonstrated on 210 HGSOC patients and 40 adenocarcinoma NSCLC patients (Fig. 5A, B of this rebuttal letter). The impact of TLS on detailed T cell phenotype determined by 12-plex immunofluorescence staining was demonstrated on stage III HGSOC patients (n=20) and stage III adenocarcinoma NSCLC patients (n=20) (Fig. 5C of this rebuttal letter). Adding more patients in the final analyses supports our previous observation documenting significantly higher frequency of eTLS and mTLS in NSCLC as compared to HGSOC patients (Fig. 5B of this rebuttal letter). Importantly, the formation of eTLSs and mTLSs in HGSOC and NSCLC correlated with high density of TCF1⁺PD1⁺CD8⁺ T cells within these structures (Fig. 5D of this rebuttal letter). Moreover, our novel findings document that mTLS development in HGSOC and NSCLC positively correlate with increased frequency of TCF1⁺PD1⁺CD8⁺ T cells and not with TIM3⁺PD1⁺CD8⁺ T cells within entire TME as documented by multiplex immunofluorescence staining (Fig. 5E of this rebuttal letter). These findings point to key quantitative and qualitative differences between mTLSs in ICI-responsive vs ICI-irresponsive neoplasms, which largely reflect higher prevalence of TCF1⁺PD1⁺CD8⁺ T cells within NSCLC with significantly higher frequency of mTLS formation. Regarding this respect, recent studies in the field underlie the importance of TCF1⁺PD1⁺CD8⁺ T cells for ICI response in immunologically hot cancer malignancies such as triple negative breast carcinoma by Wang *et al*, *Nature*, 2023, in malignant melanoma by Siddiqui *et al*, *Immunity*, 2019; Miller *et al*, *Nature Immunol*, 2019; Sade-Feldman, *Cell*, 2018; in NSCLC by Koh *et al*, *EJC*, 2022; in HNSCC by Wang *et al*, *IJOS*, 2022. These analyses were included in Figure 3 of the revised version of manuscript.

3. A substantial finding in this paper is that HGSOC harbor more terminally exhausted T cells than other cancers that are responsive to ICI, and that mature TLS in HGSOC facilitate an increase in terminally exhausted T cells. Can the authors demonstrate that the terminally differentiated cells in HGSOC are indeed exhausted/dysfunctional, and more resistant to ICI in-vitro?

Our response: We are indebted to Reviewers #2 for bringing to our attention this point. To this aim, we first investigated the impact of TIM3 expression on the functional properties of CD8⁺ TILs from 10 freshly resected HGSOC (Fig. 9A of this rebuttal letter). Importantly, about one third of tumor-infiltrating CD8⁺ T cells expressed TIM3, near to invariably in conjugation with PD1 which is in line with our multiplex immunofluorescence analyses provided in Figure 3 of revised version of manuscript. Moreover, nonspecific stimulation induced IFNG production and CD107a exposure preferentially in TIM3⁺PD1⁺ cells as compared with their TIM3⁻PD1⁻ counterparts (Fig. 9B of this rebuttal letter). Moreover, incubation with anti-TIM3 and anti-PD1 antibodies (but not with either antibody alone) increased the ability of bulk tumor-infiltrating CD8⁺ T cells to respond to nonspecific stimulation *in vitro* by synthesizing IFNG and cytolytic molecules (i.e., GZMB, PRF) (Fig. 9C of this rebuttal letter). Thus, TIM3⁺PD1⁺ cells from the HGSOC TME stand out as a functionally impaired CD8⁺ T cell subpopulation with limited effector functions.

In addition, we have implemented a novel well-defined 8-plex immunofluorescence panel (CD8, CD4, CD20, CD21, CD23, DC-LAMP and GZMB) to optimally distinguish between early TLS (eTLS) and mature TLS (mTLS) to evaluate their impact on T cell phenotype within HGSOC and NSCLC TME (Fig. 4A of this rebuttal letter). Adding more patients in the final analyses (HGSOC: n=20; NSCLC: n=20) support our previous findings documenting the correlation between eTLS and mainly mTLS presence in the TME on the final density of TCF1⁺PD1⁺CD8⁺ T cells within entire TME and in individual TLS structures in HGSOC and mainly NSCLC (Fig. 9D of this rebuttal letter). These findings might explain why NSCLC with significantly higher frequency of mTLS as compared to HGSOC associate with higher density of TCF1⁺PD1⁺CD8⁺ T cells, while the overall density of TIM3⁺PD1⁺CD8⁺ T cells was relatively comparable (Fig. 9D, E of this rebuttal letter). These analyses were included in Figure 3 of the revised version of manuscript.

Figure 9. TIM3 dictates clinically relevant immunosuppression in the TME of HGSC. (A) Gating strategy for TIM3⁺PD1⁺ and TIM3^{-/-}PD1⁺ cells (Study Group 5). The percentage of cells in each gate is reported. (B) Percentage of IFNG⁺ and CD107⁺ CD8⁺ T cells among TIM3⁺PD1⁺ and TIM3^{-/-}PD1⁺CD8⁺ T cells. (C) Fold change of IFNG⁺ and PRF1/GZMB⁺ CD8⁺ T cells after incubation with anti-PD1, anti-TIM3, and combination of anti-PD1/anti-TIM3 antibodies. (D) Density of TCF1⁺PD1⁺CD8⁺ and TIM3⁺PD1⁺CD8⁺ cells within entire tumor microenvironment (TME), eTLS areas and mTLS areas, respectively in HGSOC (n=20) and NSCLC (n=20) tumor samples as determined by immunofluorescence panel 2. Box plot: lower quartile, median, upper quartile; whiskers, minimum, maximum. Statistical significance was calculated by the Mann-Whitney test. *p* values are indicated. (E) Correlation between number of mTLS and density of TCF1⁺PD1⁺CD8⁺ and TIM3⁺PD1⁺CD8⁺ T cells in HGSOC (n=20, Study Group 1) and NSCLC (n=20). R, Pearson correlation coefficient.

Minor points:

In line 274 the sentence states sTLS could be only detected, I believe the sentence should be sTLS could only be detected.

In the abstract a sentence starting in line 46 and ending in line 51 is rather large, it would be easier to read if it could be broken up into two if possible? Also, this sentence states that CD8⁺ T cells from HGSOC were

compared to those in immunologically hot tumors like non-small cell lung cancer. Can the authors state more concretely that they compared these to non-small cell lung cancer etc.?

In line 316 the sentence states that cluster 3 was lower than median and cluster 4 were higher than median, however the median was not stated here. Can you please state what the median value is. Is the median 5 here too, as stated later for another assay in line 349.

Our response: We thank Reviewer #2 for raising these minor points, which were addressed in the revised version of manuscript.

POINT-BY-POINT REPLY TO REVIEWER N° 3

Reviewer n° 3 commented:

In this manuscript, Kasikova et al. explore the presence, composition and clinical impact of tertiary lymphoid structures in high-grade serous ovarian carcinoma (HGSOG), using a variety of methods. The topic is very timely, as TLS gather more and more interest in the field, and the authors used state-of-the-art technologies. While I appreciate the huge work this manuscript incorporates, there are some critical technical and methodological flaws that make me question the findings, notably regarding the definition of what is considered a TLS in these tumors.

1) The definition of TLS used in the methods section is problematic for 2 reasons. (1) it does not involve T cells at all. All the current literature on TLS considers them to be lymphoid aggregates with both B cells and T cells. Here, the authors do not analyze CD3 for T cells and therefore, there is a high risk of aggregates being considered as TLS without T cells. (2) the definition is very flexible with the size of such aggregates, going as low as 10 cells. I do not believe 10 cells are sufficient for a group of cells to be called a TLS. Altogether, There is probably a strong inflation of the number of TLSs that are identified throughout the manuscript.

2) Only CD8+ T cells are considered, and not CD4+ T cells, which are usually the main T cell population within TLS, and the main T cell population interacting with TLS B cells in mature TLS are follicular helper T cells, which are also CD8-. I would strongly encourage the authors to also look at CD4+ T cells, or at the very least at a pan-T cell marker such as CD3.

Our response to #1 and #2: We thank Reviewer #3 for highly constructive comments. Inspired by overall constructive comments provided from Editor and all Reviewers of *Nature Communications*, we have implemented a novel well-defined 8-plex immunofluorescence panel (CD8, CD4, CD20, CD21, CD23, DC-LAMP and GZMB) including marker for CD4⁺ T cells, follicular DCs, B cells and mature DCs as recently used within key studies in the field (*Vanhersecke et al, Nat Cancer, 2021; Helmink et al, Nature, 2020; Petitprez et al, Nature, 2020*) (Fig. 4A of this rebuttal letter, Fig. 8 of this rebuttal letter).

For further details, please, see our previous responses to Main Point #1 raised by Reviewer #2 of *Nature Communications*.

3) PD1 is not an exhaustion marker. It is expressed by most activated T cells and allows them to enter exhaustion when bound to PD-L1 or PD-L2. In this sense, it does not make sense to label PD1+TCF1+CD8+ as exhausted. On the contrary, as they have been showed to favor longer survival notably in melanoma (see Sade-Feldman et al., Cell, 2018), it is likely that they would instead have some kind of effector function.

Our response: We apologize for the strong phrasing and lack of details focusing on functional properties of so called “progenitor” and “terminally” exhausted T cells in the original version of the manuscript. As per the suggestion of Reviewer #1 and #3, we have avoided using these terms and instead objectively report cell phenotype throughout the results section. We instead alluded to the possibility that TCF1⁺PD1⁺ and TIM3⁺PD1⁺ cells may represent progenitor and terminally exhausted CD8⁺ T cells, respectively, in the discussion of the paper. For further details and additional analyses targeting the clinical relevance of PD1⁺CD8⁺ T cells subsets, please, see our previous response to Main Point #3, 4 (Fig. 2 of this rebuttal letter) raised by Reviewer #1 and Main Point #3 (Fig. 9 of this rebuttal letter) raised by Reviewer #2 of *Nature Communications*.

4) The distinction of samples between pTLS + sTLS hi/lo does not seem meaningful. Indeed, a small number of sTLS could be considered a much more mature TLS presence than many pTLS without sTLS. Instead, I would encourage the authors to consider a split between pTLS only and at least one sTLS. The current split would amount to considering pTLS and sTLS as a similar entity, even though sTLS have a much broader array of potential functions in the TME.

Our response: We thank Reviewer #3 for bringing up this critical point. To present our findings in a clear and succinct manner, we have re-classified early TLS (eTLS) as CD21⁻CD23⁻ TLS structures with no less than 50 CD20⁺ B cells and with presence of CD4⁺ and CD8⁺ T cells. Mature TLS (mTLS) were re-defined as CD21⁺CD23⁺ TLS structures regarding previous reports (Lynch et al, JITC, 2021; Zou et al, JITC, 2023) as shown in Fig 8A-C of this rebuttal letter and presented within 210 HGSOC tumor samples presented within the entire revised manuscript.

5) The prognostic impact relies on a low number of patients, and is a central point of the manuscript. It needs to be validated on a larger, independent cohort. Perhaps using the TCGA dataset would be useful here, since it has already been divided into clusters with TLS meaning in Fig. 2C.

Our response: We are indebted to Reviewers #3 for bringing to our attention this point. This point was also commented on in our previous response to Main Point #6 (Fig. 4 of this rebuttal letter) raised by Reviewer #1 of *Nature Communications*. The prognostic value of eTLS and mTLS formation was evaluated in 210 HGSOC patients in the revised version of manuscript. Additionally, we have collected an independent cohort of 40 HGSOC patients from University of Basel in kind collaboration with prof. Viola Heinzlmann-Schwarz, indicated as study cohort 3 within the revised version of manuscript. Although, TCGA dataset represent a unique source of sequencing data from large cohort of patients, the stratification used within heatmap (Fig. 2C of revised version of manuscript) was done based on TLS-like chemokine profile and does not directly correspond to development and number of eTLS and mTLS within the TME. Thus, we prefer to conclude our findings on 4 independent HGSOC cohorts where eTLS and mTLS were defined based on multiplex immunofluorescence staining.

6) TLS in HGSOC have already been studied in a previous study that is not referenced nor discussed. See Kroeger et al., Clinical Cancer Research 2016.

Our response: We apologize for the lack of details summarizing key-findings presented by Kroeger et al in HGSOC in the original version of the manuscript. Inspired by the suggestions from Reviewer #3, we have amply discussed in more detail available findings from others in the introduction and discussion of the revised version of manuscript.

7) Some claims do not seem to be sufficiently backed by data but are rather guesses as to potential mechanisms. These need to be either dimmed or properly backed by mechanistic experiments. Those include:

a. Lines 308-310: It is not sure how relevant the intra-TLS densities are, as the simple presence of TLS, regardless of how dense it is in some cell populations could be more impactful. Also, there is a very low variability for oTLS cell densities, so the authors are at risk of overinterpreting on a few outliers.

Our response: We agree with Reviewer #3 that our findings might be impacted by the overall presence of TLS in the TME. The density of CD8⁺ T cells, GZMB⁺CD8⁺ T cells, CD20⁺ B cells and DC-LAMP⁺ DCs was analyzed per mm² of either entire TME or eTLS or mTLS areas (mm²) (Fig. 8C of this rebuttal letter) of 68 HGSOC patients

(Study Group 1). These findings were mainly depicted to support our previous spatial transcriptomic analyses presented in Figure 1D, E to demonstrate that TLS represents unique sites within TME with high accumulation of effector immune components.

b. Lines 396-397: there is no data in the manuscript that can adequately suggest that TLS redistribute any cell population.

Our response: We agree with Reviewer #3 that in our current version of manuscript we don't provide any mechanistic data documenting redistribution of immune cells from TLS within TME. Regarding this respect, we removed spatial distribution analyses from the revised version of the manuscript. Finally, we believe that development of novel transcriptomic and imaging technologies deciphering multiparametric analyses, combining both mRNA and protein level, will enable the tracking of spatial redistribution of immune cells from and to TLS structures are highly needed to uncover the power of TLS to redistribute immune cells subsets within TME.

c. Lines 408-412: It is unlikely that all PD1+CD8+ T cells are either TIM3+ or TCF1+, but I guess a number of PD1+CD8+ T cells are either double positive or double negative for TIM3 and TCF1. Therefore, summing these 2 subpopulations to account for 100% makes is irrelevant.

Our response: We agree with Reviewer #1 that summing TCF1+PD1+CD8+ T cells and TIM3+PD1+CD8+ T cells as total PD1+CD8+ T cell population is not relevant, and we might miss some other important subsets. Thus, we decided to remove these original data from the revised version of manuscript, and we demonstrate our findings only as density of T cells subsets within TLS areas and entire TME (Fig. 5C, D of this rebuttal letter and Figure 3 of revised version of manuscript).

d. Lines 454-456: A simple colocalization between cells is not enough to claim that B cells influence CD8+ T cells phenotypes.

Our response: We totally agree that a simple colocalization between cells is not enough to claim the impact of B cells on CD8+ T cells phenotype. Inspired by constructive comments from Reviewer #1 and 3, additional functional studies to strengthen our data from B-cell depletion experiments were performed and concluded as

response to main Point #8 raised by Reviewer #1 (Fig. 6 of this rebuttal letter). Of note, spatial colocalization analyses were removed from the revised version of manuscript.

8) The number of TLS found in the orthotopic mouse model seems extremely high. There are very few reports of any TLS in orthotopic mouse models (it is usually thought that this is because TLS induction takes too long for the span of these experiment), and only a handful of intrinsic models usually develop TLSs. Having an orthotopic mouse models develop dozens of mature TLS is simply unheard of and makes me question the robustness of the classification of aggregates as TLS.

Our response: We totally agree with Reviewer #3 that our current *in vivo* experimental model has various limitations as also correctly pointed out by Reviewer #1. Inspired by overall critique we have implemented novel mice experimental models, novel immunofluorescence staining to determine TLS development in mice tumor samples and novel flow cytometry panels to characterize the T cell responses after anti-PD1 therapy in TLS^{Lo} and TLS^{Hi} tumors. Please, see our previous responses to Main Point #9 raised by Reviewer #1 of *Nature Communications*. These novel analyses were included in Figure 7 of this rebuttal letter and Figure 6 of revised version of manuscript.

REVIEWER COMMENTS

Reviewer #1 (Remarks to the Author):

This is a significantly revised manuscript in which the authors have addressed many of the points raised in the first review. The narrative is now easier to follow, and the key findings are more clearly highlighted. The main conclusion (and most original finding in the paper) is that the presence of so-called early TLS (eTLS) is associated with favorable prognosis in HGSOC, whereas mature TLS (mTLS) are associated with unfavorable prognosis in HGSOC, as opposed to NSCLC. The authors attribute this to their claim that mTLS are smaller and less prevalent in HGSOC than NSCLC. Thus, they propose that TLS fail to fully mature in HGSOC (as opposed to NSCLC), resulting in partially mature TLS (my terminology) that promote a maladaptive TIM3+ T cell phenotype which in turn promotes poor prognosis. Thus, this leads to the unprecedented (to my knowledge) view that, in terms of HGSOC patient survival, it is better to have eTLS in the absence of mTLS. In contrast, in lung cancer and many other cancers, it is better to have mTLS. If so, this places HGSOC as an outlier compared to many other cancers in which TLS have been studied. It also suggests that TLS go through a 'maladaptive stage' midway through their development. In other words, eTLS are 'good', small mTLS are 'bad' and large mTLS are 'good' again. This goes against current thinking in the TLS field, where the weight of evidence suggests that TLS to get 'better and better' (from a prognosis perspective) along their developmental trajectory.

Of course, radical new findings are always welcome in science, but they do demand a higher burden of evidence. In this regard, several issues need to be addressed:

1. The patient cohorts are not uniform with regard to important clinicopathological features. For example, the authors state that eTLS are more prevalent in early-stage tumors, which are fairly abundant (28%) in cohort 1. Could this be why eTLS appear to be prognostically favorable compared to mTLS?

It seems that cohort 2 received a DC vaccine? This should disqualify this as a validation cohort.

Not all cases in cohorts 1 & 2 are HGSOE; other EOC histologies are included, which are known to have distinct immunological and clinical properties.

It's not clear if some of the cases may have received neoadjuvant chemo? If so, that would also be an issue, as NACT has a profound influence on TIL patterns.

In general, the cohorts lack a uniform reporting structure such that it is impossible to identify all the ways in which they may differ.

A final point is that multi-variate analysis needs to be performed to rule out these and other potentially confounding issues.

The above issues are critical for establishing whether or not eTLS are genuinely more prognostically favorable than mTLS.

2. The use of the term 'early' TLS is an over-simplification. Although this term is used loosely in some publications, stricter definitions should be applied, especially in a study such as this, where key findings rest on the definition of eTLS. There is only limited evidence in the literature that eTLS-like aggregates are all on their way to becoming TLS, and accumulating evidence that some of these aggregates may perform unique functions unrelated to TLS per se (discussed in PMID: 35393541). Thus, 'eTLS' might actually refer to several distinct types of aggregate, each with potential prognostic significance.

3. The comparison of TLS in HGSOE and NSCLC is key to the author's main conclusions. However, this is only demonstrated in 4 panels (Figs 3E, 3F, 4B and 4C), which do not address this issue head-on. Fig 3E shows representative images only. Fig 3F shows the numbers of mTLS, which is useful information but does not address size or functional status. Fig 4B and 4C appear to show the densities of Tfh cells and fDCs in total tumor tissue from HGSOE vs. NSCLC. It's not surprising that NSCLC has higher levels of these cells, given that these tumors are generally hotter and (according to the authors) have bigger and more abundant TLS. If the authors wish to demonstrate that TLS are less mature in HGSOE than NSCLC, this would require a careful, detailed analysis of large numbers of TLS from both

tumor types. Ideally, the authors would find a tangible difference in TLS structure or composition. I don't think size is a sufficient distinction on which to rule one TLS as less functional than another; I'm not aware of evidence that large mTLS are more functional than smaller mTLS.

4. As mentioned in my first review, I don't think the mouse models add to this story. Mouse models are notoriously poor for recapitulating phenomena related to TIL-B and TLS. While I recognize there are some exceptions to this rule (e.g. when a given tumor line was engineered to have higher TMB, PMID: 31730857), one can't draw rigorous conclusions by comparing cell line A which naturally has a high TMB to cell line B which naturally has a low TMB. Cell lines differ by a plethora of other factors that could contribute to any of the observed effects.

5. Finally, I think the authors overstate the possibility that mTLS in HGSOC may give rise to less fit CD8 TIL populations and this in turn explains why HGSOC responds poorly to ICI. While this is a reasonable hypothesis that could be stated once or twice in the paper, at present it is overemphasized, giving the impression that the paper provides more evidence in favor of this notion than it actually does.

6. Minor points:

- in Fig 1D, how did they stain for so many markers simultaneously? The methods section does not describe such highly multi-plexed methodology.

- in human tissues, CD4 is expressed by macrophages as well as T cells -- how were these distinguished?

Reviewer #2 (Remarks to the Author)

Dear Author,

I know that you have put a large amount of work into the current submission and congratulate you. I do believe that this paper will be of significance to the TLS field. You

have generally addressed my questions. However, I have a few minor questions remaining.

1. Did you also evaluate for PNAD+ vasculature, a marker for high endothelial venules commonly associated with TLS?
2. I saw images segmenting out TLS via Halo. However, how were the T-cell areas, associated with the B cell follicles, determined in these analyses? Was that done by a measurement, or something else?
3. Did you evaluate the murine advanced lymphoid aggregates for additional TLS markers such as PNAd, CD21, CD23 etc.?

Thank you.

Reviewer #4 (Remarks to the Author): **new reviewer**

Reviewer #3.

Many of the comments have now been addressed satisfactorily. The Authors are commended for their thoughtful and responsive rebuttal. Notwithstanding, please address the following:

- 1) In several locations in this revised manuscript, the Authors mention the use of a TLS-like chemokine profile, which is not adequately described. This is neither described in the Materials and Methods in the Main content nor in the Materials and Methods in the Supplemental content. Moreover, in Figure 2C, no mention is made of it in the legend. Similar for the legend of Supplemental Figure 5D.
- 2) Given 1), in the Introduction section, reference is made to melanoma and is cited as reference #17. However, there is a published study by Messina et al., Scientific Reports volume 2, Article number: 765 (2012) that should be considered for its relevance here.
- 3) It would be worthwhile for the Authors to consider adding text to the Discussion section to reflect their responses to 7)a and 7)b.

4) The Authors could comment on the issue of reliability of analyses done on just single slices off a block of tissue given concerns for the heterogeneity of sampling of a tumor mass (e.g., Gerlinger et al., NEJM 336; 883-892, 2012).

POINT-BY-POINT REPLY TO REVIEWER N° 1

Reviewer n° 1 commented:

This is a significantly revised manuscript in which the authors have addressed many of the points raised in the first review. The narrative is now easier to follow, and the key findings are more clearly highlighted. The main conclusion (and most original finding in the paper) is that the presence of so-called early TLS (eTLS) is associated with favorable prognosis in HGSOC, whereas mature TLS (mTLS) are associated with unfavorable prognosis in HGSOC, as opposed to NSCLC. The authors attribute this to their claim that mTLS are smaller and less prevalent in HGSOC than NSCLC. Thus, they propose that TLS fail to fully mature in HGSOC (as opposed to NSCLC), resulting in partially mature TLS (my terminology) that promote a maladaptive TIM3+ T cell phenotype which in turn promotes poor prognosis. Thus, this leads to the unprecedented (to my knowledge) view that, in terms of HGSOC patient survival, it is better to have eTLS in the absence of mTLS. In contrast, in lung cancer and many other cancers, it is better to have mTLS. If so, this places HGSOC as an outlier compared to many other cancers in which TLS have been studied. It also suggests that TLS go through a 'maladaptive stage' midway through their development. In other words, eTLS are 'good', small mTLS are 'bad' and large mTLS are 'good' again. This goes against current thinking in the TLS field, where the weight of evidence suggests that TLS to get 'better and better' (from a prognosis perspective) along their developmental trajectory. Of course, radical new findings are always welcome in science, but they do demand a higher burden of evidence. In this regard, several issues need to be addressed:

Our response: We are indebted to Reviewer #1 for the overall positive evaluation of our revised version of manuscript. We agree with the view proposed by Reviewer #1, that mature TLSs in HGSOCs are limited in number, and some of our data also suggest to the fact that they might be less developed as compared to mTLS in NSCLC. Overall, this might be explained by the low level of neoantigens in HGSOCs, leading to low level of T_{FH} cells, insufficient TLS development without fully activated GCs and hence the generation of B cells unable to preserve progenitor T cell phenotype. These observations may at least partially explain the limited sensitivity of patients with HGSOC to conventional ICIs targeting the PD1/PD-L1 axis, in line with available preclinical and clinical findings (*Chalabi et al, Nat Med, 2020 ; Gao et al, Nat Med, 2020 ; van Dijk et al, Nat Med, 2020 ; Cottrell et al, Ann. Oncol, 2018 ; Helmink et al, Nature, 2020 ; White et al, Ann.Oncol, 2023 ; Petitprez et al, Nature, 2020* and many others).

The qualitative differences between mTLS development might result from low levels of CXCR5⁺PD1⁺FOXP3⁻CD4⁺CD68⁻ T_{FH} cells, CD21⁺CD23⁺ fDCs which in turn correlate with significantly reduced level of GZMB⁺CD8⁺, GZMB⁺CD4⁺ T cells as well as TCF1⁺PD1⁺CD8⁺ T cells within mTLS in HGSOE as compared to mTLS in NSCLC samples, as documented by our previous and recent findings presented in Figure 3, 4 of revised version of manuscript and Figure 3 of this rebuttal letter. Besides the qualitative differences in mTLS contexture, our findings also point to key quantitative difference between mTLS in HGSOE and NSCLC (Figure 3 of revised version of manuscript), which might be crucial for immune profiling and T cell differentiation within tumor microenvironment. Thus, in line with previous studies our findings document that formation of eTLS and mTLS in HGSOE correlate with increased T cell infiltrate. Globally, our data mainly point to low number of mTLS in HGSOE tumor samples (located especially in marginal zones of tumor samples) which might be insufficient to drive and/or maintain the TCF1⁺PD1⁺CD8⁺ T cells phenotypes within the entire tumor microenvironment as compared to NSCLC with high frequency of both eTLS and mTLS developed within entire tumor microenvironment. Taken together, we believe that both the maturity and frequency of ectopic lymphoid organs within tumors might be the key for successful differentiation and/or preservation of T cells within so called progenitor ICI-responsive TCF1⁺PD1⁺CD8⁺ T cells phenotype.

And raised these major critiques:

1. The patient cohorts are not uniform with regard to important clinicopathological features. For example, the authors state that eTLS are more prevalent in early-stage tumors, which are fairly abundant (28%) in cohort 1.

Could this be why eTLS appear to be prognostically favorable compared to mTLS?

It seems that cohort 2 received a DC vaccine? This should disqualify this as a validation cohort.

Not all cases in cohorts 1 & 2 are HGSOE; other EOC histologies are included, which are known to have distinct immunological and clinical properties.

It's not clear if some of the cases may have received neoadjuvant chemo? If so, that would also be an issue, as NACT has a profound influence on TIL patterns.

In general, the cohorts lack a uniform reporting structure such that it is impossible to identify all the ways in which they may differ.

A final point is that multi-variate analysis needs to be performed to rule out these and other potentially

confounding issues.

The above issues are critical for establishing whether or not eTLS are genuinely more prognostically favorable than mTLS.

Our response: Inspired by the constructive critique raised by Reviewer #1 related to uniformity of clinicopathological features of patients involved in our cohorts, we have performed additional analyses of early (stage I+II) and late (III+IV) stage patients survival using uniform cohort of chemotherapy naïve and DC-based therapy naïve HGSOC patients. Other histological subtypes were removed for the final survival analyses. Similar to our previous findings, the presence of eTLSs within cluster 2 patients correlate with favorable disease outcome in both early and late stage HGSOC as documented by Kaplan-Meier analyses (early stage: RFS: $p=0.0004$; OS: $p=0.047$; late stage: RFS: $p=0.003$; OS: $p=0.015$) (**Fig. 1A of this rebuttal letter**) and both univariate (Table 1 of this rebuttal letter) and multivariate COX regression analyses (Table 2 of this rebuttal letter). Moreover, the presence of mTLSs in cluster 3 patients is linked to poor disease outcome as compared the cluster 2 patients with only eTLS development (RFS: $p=0.005$; OS: $p=0.034$) (**Fig. 1 of this rebuttal letter**). These analyses were included in Figure 2, Suppl. Figure 6, Suppl. Table 8 and 9 of revised version of manuscript. Finally, as per Reviewer #1 suggestion, we have replaced the supplementary tables commenting on clinicopathological characteristics of patients with single uniform reporting table as Suppl. Table 1 of revised version of manuscript.

Figure 1. The clinical relevance of TLS formation and maturation in early (stage I+II; n=38) and late (stage III+IV; n=171) stage HGSOC patients. (A, B) Relapse-free survival (RFS) and overall survival (OS) of HGSOC patients based on stratification into 3 clusters with not TLS development (CL1), only eTLS development (CL2) and both eTLS and mTLS development (CL3) in 4 independent cohorts of early (stage I+II) (A) and late (stage III+IV) (B) HGSOC patients.

A	RFS			OS		
	Variable	HR (95%CI)	p value	HR (95%CI)	p value	
TLS clusters	CL 1	1		CL 1	1	
	CL 2	0.2 (0.08-0.53)	0.001	CL 2	0.53 (0.17-1.68)	0.279
	CL 3	0.85 (0.26-2.78)	0.783	CL 3	1.55 (0.41-5.85)	0.521
Stage	I	1		I	1	
	II	1.47 (0.63-3.45)	0.371	II	0.71 (0.24-2.07)	0.524
Debulking	R0	1		R0	1	
	R2	2.77 (0.64-12.05)	0.174	R2	2.89 (0.63-13.13)	0.171
Age		1.07 (1.02-1.12)	0.003		1.047 (1.01-1.13)	0.015

B	RFS			OS		
	Variable	HR (95%CI)	p value	HR (95%CI)	p value	
TLS clusters	CL 1	1		CL 1	1	
	CL 2	0.55 (0.37-0.81)	0.002	CL 2	0.58 (0.37-0.90)	0.014
	CL 3	1.07 (0.66-1.73)	0.797	CL 3	1.07 (0.64-1.78)	0.803
Stage	III	1		III	1	
	IV	1.03 (0.54-1.96)	0.939	IV	0.9 (0.42-1.94)	0.785
Debulking	R0	1		R0	1	
	R1	1.65 (0.95-2.88)	0.076	R1	1.84 (1.02-3.34)	0.043
	R2	1.63 (1.12-2.38)	0.01	R2	1.49 (0.98-2.25)	0.062
Age		1.02 (1.00-1.03)	0.03		1.03 (1.01-1.05)	0.004

Table 1. Univariate Cox proportional hazard analyses in early (n=38) (A) and late (n=171) stage (B)

HGSOC patients. *Abbreviations.* TLS, tertiary lymphoid structures

A	RFS			OS		
	Variable	HR (95%CI)	p value	HR (95%CI)	p value	
TLS clusters	CL 1			CL 1		
	CL 2	0.25 (0.08-0.77)	0.015	CL 2	0.8 (0.21-2.99)	0.736
	CL 3	0.63 (0.10-4.09)	0.627	CL 3	2.08 (0.22-19.84)	0.523

Debulking	R0			R0		
	R2	0.68 (0.13-3.54)	0.651	R2	1.59 (0.26-9.57)	0.614
Age		1.06 (1.00-1.12)	0.042		1.06 (1.00-1.13)	0.056

B

Variable	RFS			OS		
		HR (95%CI)	p value		HR (95%CI)	p value
TLS clusters	CL 1	1		CL 1	1	
	CL 2	0.58 (0.39-0.88)	0.011	CL 2	0.63 (0.45-1.16)	0.042
	CL 3	0.96 (0.51-1.79)	0.886	CL 3	1.58 (0.79-3.12)	0.193
Debulking	R0	1		R0	1	
	R1	1.49 (0.85-2.62)	0.166	R1	1.6 (0.87-2.92)	0.128
	R2	1.49 (1.01-2.19)	0.041	R2	1.38 (0.9-2.11)	0.135
Age		1.01 (1.00-1.03)	0.153		1.02 (1.00-1.04)	0.058

Table 2. Multivariate Cox proportional hazard analyses in early (n=38) (A) and late (n=171) stage (B) HGSOc patients. Abbreviations. TLS, tertiary lymphoid structures

2. The use of the term 'early' TLS is an over-simplification. Although this term is used loosely in some publications, stricter definitions should be applied, especially in a study such as this, where key findings rest on the definition of eTLS. There is only limited evidence in the literature that eTLS-like aggregates are all on their way to becoming TLS, and accumulating evidence that some of these aggregates may perform unique functions unrelated to TLS per se (discussed in PMID: 35393541). Thus, 'eTLS' might

actually refer to several distinct types of aggregate, each with potential prognostic significance.

Our response: We apologize for the lack of details focusing on early and mature TLS identification in human cancer tissues. TLSs were defined as lymphoid aggregates of CD20⁺, CD4⁺, CD8⁺, DC-LAMP⁺ cells. Only TLS made up of more than 50 cells were included in the analysis. In the absence of CD21⁺ and CD23⁺ positivity, the TLS was identified as early TLS (**Fig. 2A of this rebuttal letter**). TLSs were defined as “mature” when CD21⁺ and CD23⁺ dendritic cells were co-detected in the TLS (**Fig. 2B, C of this rebuttal letter**). Inspired by the constructive comment of Reviewer #1 we have provided further details on early TLS (eTLS) and mature TLS (mTLS) identification within material/methods and results section of revised version of manuscript.

Figure 2. Spatial distribution of early TLS (eTLS) and mature TLS (mTLS). (A, B) Representative images of eTLS and mTLS using immunofluorescence of CD4, CD8, CD20, CD21, CD23, DC-LAMP and GZMB staining. (C) Representative image of TLS areas quantification in HALO software.

3. The comparison of TLS in HGSOE and NSCLC is key to the author's main conclusions. However, this is only demonstrated in 4 panels (Figs 3E, 3F, 4B and 4C), which do not address this issue head-on. Fig 3E shows representative images only. Fig 3F shows the numbers of mTLS, which is useful information but does not address size or functional status. Fig 4B and 4C appear to show the densities of Tfh cells and fDCs in total tumor tissue from HGSOE vs. NSCLC. It's not surprising that NSCLC has higher levels of these cells, given that these tumors are generally hotter and (according to the authors) have bigger and more abundant TLS. If the authors wish to demonstrate that TLS are less mature in HGSOE than NSCLC, this would require a careful, detailed analysis of large numbers of TLS from both tumor types. Ideally, the authors would find a tangible difference in TLS structure or composition. I don't think size is a sufficient distinction on which to rule one TLS as less functional than another; I'm not aware of evidence that large mTLS are more functional than smaller mTLS.

Our response: Inspired by constructive critique by Reviewer #1, the density of CD4⁺CD68⁻, CD8⁺ T cells, CD68⁺CD4⁻ TAMs, CXCR5⁺PD1⁺FOXP3⁻CD4⁺CD68⁻ T cells, GZMB⁺CD4⁺ and GZMB⁺CD8⁺ and PD1⁻FoxP3⁺CD4⁺ was determined within eTLS, mTLS and entire TME of HGSOE and NSCLC samples to extend our current findings on T cells differences displayed in Figure 3 of current version of manuscript. Supporting our previous findings documenting higher density of CXCR5⁺PD1⁺FOXP3⁻CD4⁺CD68⁻ T_{FH} cells and CD21⁺CD23⁺ fDCs in mTLS areas of NSCLC samples as compared to HGSOE, mTLS development in NSCLC is associated with higher density of CD4⁺CD68⁻ T cells, CD8⁺ T cells, GZMB⁺CD4⁺ and GZMB⁺CD8⁺ cells as compared to HGSOE (**Fig. 3 of this rebuttal letter**), as per digital pathology analyses only in predefined mTLS areas (**Fig. 2C of this rebuttal letter**). Moreover, we further determined the density of CD68⁺ TAMs and PD1⁻FOXP3⁺CD4⁺ T cells in mTLS areas in HGSOE and NSCLC samples. We did not observe any significant difference in frequency of potentially immunosuppressive immune subsets between mTLS in HGSOE and NSCLC tumor samples (**Fig. 3 of this rebuttal letter**). Thus, these additional results support our previous findings documenting the limited maturation of TLS in the TME of HGSOE, which may be associated with limited capacity of TLS to develop GCs with fully mature follicular CD21⁺CD23⁺ DCs. These novel analyses were included in Figure 4 and Suppl. Figure 8 of revised version of manuscript. Altogether, we believe that not only qualitative difference but mainly quantitative difference in mTLS frequency between HGSOE and NSCLC are two important aspects to maintain clinically relevant TCF1⁺PD1⁺CD8⁺ T cells within tumor microenvironment.

Figure 3. Immune composition in distinct maturation types of tertiary lymphoid structures (TLS) in HGSOc and NSCLC patients. Density of CD4⁺CD68⁺ T cells, CD8⁺ T cells, CD68⁺CD4⁺ tumor associated macrophages, GZMB⁺CD8⁺ T cells, GZMB⁺CD4⁺ T cells and PD1⁺FoxP3⁺CD4⁺ T cells within complete tumor microenvironment (TME, including TLS), eTLS and mTLS of HGSOc (Study Group 1) and NSCLC samples (Study Group 6) as determined by immunofluorescence. Box plots: lower quartile, median, upper quartile; whiskers, minimum, maximum. Statistical significance was calculated by the Mann–Whitney test. p values are indicated.

4. As mentioned in my first review, I don't think the mouse models add to this story. Mouse models are notoriously poor for recapitulating phenomena related to TIL-B and TLS. While I recognize there are some exceptions to this rule (e.g. when a given tumor line was engineered to have higher TMB, PMID: 31730857), one can't draw rigorous conclusions by comparing cell line A which naturally has a high TMB to cell line B which naturally has a low TMB. Cell lines differ by a plethora of other factors that could contribute to any of the observed effects.

Our response: We agree with Reviewer #1 that development of mouse experimental model with *Apobec* overexpression as previously employed by *Hollern et al, Cell, 2019* might be suitable for TLS development

analyses based on TMB values. Inspired by this study, we have originally performed the *Apobec3* transfection for ID8 and BR5 cell line upon discussion with corresponding author of *Hollern et al* study. Although, the transfection was clearly successful as shown in Figure 4A of this rebuttal letter, we failed to observe a significant increase in TMB driven by *Apobec3* overexpression in ID8 and BR5 ovarian cancer lines (Fig. 4B of this rebuttal letter). Thus, further driven by these findings, we evaluated the TMB in parental ID8 and *Brca1^{-/-} Trp53^{-/-}* ID8 cell line generated by CRISPR/Cas9 gene editing and kindly provided by prof. Ian McNeish (*Walten et al, CCR, 2016*). However, against expectations, this genetic modification also did not lead to significant increase in TMB levels (Fig. 4C of this rebuttal letter). Driven by these experimental findings, we selected another mouse model of ovarian cancer that is also syngeneic to C57BL/6J mice (as per initial request of Reviewers of Nature Communications) but exhibits high TMB (*Brca1^{-/-} Trp53^{-/-}, Myc, Hras SO1* cells). Thus, TMB^{Lo} ID8 and TMB^{Hi} SO1 mouse ovarian cancer cell lines were used to generate tumors in immunocompetent female syngeneic C57BL/6J mice for subsequent TLS analyses. We agree with Reviewer #1, that provided mouse experimental model has limitation related to origin of cell line which might be further discussed within revised version of manuscript. However, we hope that Reviewer #1 will agree that heterogeneity which might come from the origin of cell lines might partially reflect the high variability and heterogeneity also among individual patients with HGSOC. Moreover, also driven by the fact that the TLS field lack for mouse models designed for TLS analyses, we believe that our current findings might support the development of optimal models for TLS analyses as also confirmed by consent statements provided by Reviewers #2 and #3 of *Nature Communications*. Our choice to preserve mouse data in the revised version of the manuscript was supported by the Editor of *Nature Communications*.

Figure 4: Mutational impact of *Apobec3* in ID8 and BR5 mouse ovarian cell lines. (A) Immunoblot of *Apobec3* following transduction of ID8 and BR5 cell lines. GAPDH is loading control. (B) Somatic mutation prevalence (mutations per megabase) in ID8 and BR5 parental cell lines and cell lines with overexpression of

Apobec3. (C) Somatic mutation prevalence (mutations per megabase) in ID8, *Brca1*^{-/-} *Trp53*^{-/-} ID8 and *Brca*^{-/-} *Trp53*^{-/-}, *Myc*, *Hras* SO1 cell lines. Statistical analyses were calculated by Mann-Whitney test. p values are indicated.

5. Finally, I think the authors overstate the possibility that mTLS in HGSOc may give rise to less fit CD8 TIL populations and this in turn explains why HGSOc responds poorly to ICI. While this is a reasonable hypothesis that could be stated once or twice in the paper, at present it is overemphasized, giving the impression that the paper provides more evidence in favor of this notion than it actually does.

Our response: We apologize for the strong phrasing within the original version of manuscript. Inspired by constructive critique by Reviewer #1 we have toned down our statements related to the possibility that mTLS in HGSOc may give rise to less functional CD8⁺ T cells within revised version of manuscript.

6. Minor points:

- in Fig 1D, how did they stain for so many markers simultaneously? The methods section does not describe such highly multi-plexed methodology.

Our response: We apologize for the lack of details focusing on sequential IHC on multiplex immunofluorescence staining within the original version of manuscript. Briefly, sequential IHC consisting of iterative cycles of staining, scanning, and antibody and chromogen stripping was performed according to a modified protocol based on previous reports (*Tsujikawa et al, Cell Reports, 2017; Glass et al, J.Histochem.Cytochem,2009*). Primary antibodies, horseradish peroxidase (HRP)-conjugated polymer and chromogenic detection were serially added in the indicated order and condition shown in Figure 5 of this rebuttal letter. Two forms of negative controls were used during analyses: 1) slides for conventional negative controls were treated with 2.5% goat serum in PBS without primary antibodies; 2) slides for sequential IHC negative controls were used for confirmation of complete antibody and signal stripping. AEC color signals were extracted from each digitized single-marker image by color deconvolution, followed by pseudo-coloring and image fusions in HALO INDICA. Further details can be found in Supplemental material and methods of revised version of manuscript.

As per suggestion of Reviewer #1, we have provided more details on sequential immunostaining protocol in the revised version of manuscript.

Figure 5. Experimental design of sequential immunostaining.

[Editorial Note: Figure created with BioRender.com]

- in human tissues, CD4 is expressed by macrophages as well as T cells -- how were these distinguished?

Our response: Inspired by the Reviewer #1 comment, we have included sequential CD68 staining (methodology explained within Fig. 5 of this rebuttal letter) as marker of tumor associated macrophages (TAMs) in the final version of panel for T_{FH} detection to avoid any false positivity of potential CD68⁺ TAMs. Thus, representative immunostaining and the density of CXCR5⁺PD1⁺FoxP3⁻CD68⁻CD4⁺ T_{FH} cells is currently displayed Figure 6 of this rebuttal letter and in the Figure 4 of revised version of manuscript.

Figure 6. Detection of CXCR5⁺PD1⁺FOXP3⁻CD4⁺CD68⁻ T_{FH} using multiplex immunofluorescence staining. (A-C) Representative image (A) and box plot showing the density of CXCR5⁺PD1⁺FOXP3⁻CD4⁺CD68⁻ T_{FH} cells (B) in the complete tumor microenvironment (TME), eTLS and mTLS of HGSOC and NSCLC patients and (C) box plot showing density of CD23⁺CD20⁺ follicular dendritic cells (fDCs) in TLS^{Lo} and TLS^{Hi} HGSOC and NSCLC patients from study group 1 and study group 6, respectively. Box plots: lower quartile, median, upper quartile; whiskers, minimum, maximum. Statistical significance was calculated by the Mann–Whitney test. *p* values are indicated.

POINT-BY-POINT REPLY TO REVIEWER N° 2

I know that you have put a large amount of work into the current submission and congratulate you. I do believe that this paper will be of significance to the TLS field. You have generally addressed my questions. However, I have a few minor questions remaining.

Our response: We are indebted to Reviewer #2 of Nature Communications for the positive evaluation of our revised version of manuscript.

1. Did you also evaluate for PNAD+ vasculature, a marker for high endothelial venules commonly associated with TLS?

Our response: We thank Reviewer #2 for bringing this point to our attention. As we mainly focused on distinction of eTLS and mTLS, we have implemented a novel well-define 8-plex immunofluorescence panel focusing on CD8, CD4, CD20, CD21, CD23, DCLAMP and GZMB, including marker for CD4⁺ T cells, follicular DCs, B cells and mature DCs are recently used within key studies in the field (*Vanhersecke et al, Nat Cancer, 2021; Helmink et al, Nature, 2020; Petitprez et al, nature 2020*). We agree that detection of PNAD⁺ high endothelial venules might further improve the detection of TLS structures within tumor samples. Inspired by Reviewer #2 suggestion, we implemented PNAD⁺ staining into our current 8-plex immunofluorescence staining using sequential immunohistochemistry (methodology was described in detail within Figure 5 of this rebuttal letter). However, using PNAD⁺ staining in HGSOC and NSCLC tumor tissues did not improve our current strategy for distinction of early lymphoid aggregates, early TLS (eTLS) and mature TLS (mTLS) based on previous established multiplex, as PNAD positivity was observed within all predefined structures in both HGSOC and NSCLC tumor samples (Fig. 7A, B of this rebuttal letter). Moreover, besides TLS areas, PNAD expression was also observed within tumor islets in distinct tumor samples of HGSOC and NSCLC samples (Fig. 7C of this rebuttal letter). Thus, we hope that Reviewer #2 will agree that our previously developed multiplex immunofluorescence panel represents power tool for TLS detection within human tumor samples.

Figure 7. Representative immunostaining for TLS structures using immunofluorescence of PNAd, CD4, CD20, CD21 and CD23 versus single PNAd immunostaining in NSCLC (A) and HGSOC (B) tumor samples. (C) Representative image of PNAd staining in tumor islets of NSCLC and HGSOC samples.

2. I saw images segmenting out TLS via Halo. However, how were the T-cell areas, associated with the B cell follicles, determined in these analyses? Was that done by a measurement, or something else?

Our response: The quantification of TLS areas was done in HALO software with manual depiction of TLS based on described criteria. We currently prefer the manual depiction with further confirmation by two independent pathologists from the team, as we do not believe that the current artificial intelligence (AI) module on HALO perfectly covers the TLS regions as compared to experienced scientists/pathologists. However, we agree that the use of AI for TLS detection on H&E slides has gained attention (*Barmpoutis et al, Plos One, 2021; Hu et al, Cell Syst, 2023*) and may represent a future tool for pathology laboratories.

3. Did you evaluate the murine advanced lymphoid aggregates for additional TLS markers such as PNAd, CD21, CD23 etc.?

Our response: Inspired by the constructive comment provided by Reviewer #2, we have implemented anti-CD21 antibody to our current sequential IHC protocol for LAs detection in mice studies. Confirming our previous observation, CD21⁺ cells were localized within large LAs and thus confirm our previous findings (Figure 8 of this rebuttal letter). The novel IHC sequential protocol and novel representative figures were included in the revised version of manuscript (Figure 6 or revised version of manuscript).

Figure 8: Representative immunostaining for CD4, CD8, CD20 and CD21 lymphoid aggregates within TMB^{Lo} ID8 and TMB^{Hi} SO1 ovarian tumors. Scale bars 1 and 2.5 mm.

POINT-BY-POINT REPLY TO REVIEWER N° 3

Many of the comments have now been addressed satisfactorily. The Authors are commended for their thoughtful and responsive rebuttal. Notwithstanding, please address the following:

1) In several locations in this revised manuscript, the Authors mention the use of a TLS-like chemokine profile, which is not adequately described. This is neither described in the Materials and Methods in the Main content nor in the Materials and Methods in the Supplemental content. Moreover, in Figure 2C, no mention is made of it in the legend. Similar for the legend of Supplemental Figure 5D.

Our response: We thank Reviewer #3 for bringing this point to our attention. The TLS-like chemokine signature (*CCL2, CCL3, CCL4, CCL5, CCL8, CCL18, CCL19, CCL21, CXCL9, CXCL10, CXCL11, and CXCL13*) as previously published by *Coppola et al, Am J Patol, 2011; Messina et al, Sci Rep, 2012; Li et al, Med Oncol, 2022; Sautes-Fridman et al, Front Immunol, 2016* was employed to confirm our analyses within manuscript. We apologize for the oversight, the methodology and reference are provided within materials and methods section, main text and figure legends in the revised version of manuscript.

2) Given 1), in the Introduction section, reference is made to melanoma and is cited as reference #17. However, there is a published study by Messina et al., Scientific Reports volume 2, Article number: 765 (2012) that should be considered for its relevance here.

Our response: Inspired by constructive comment of Reviewer #3, we have included the reference Messina et al, 2012 in the final version of manuscript.

3) It would be worthwhile for the Authors to consider adding text to the Discussion section to reflect their responses to 7)a and 7)b.

Our response: Inspired by suggestion provided by Reviewer #3, we have amply discussed in more details these findings in the discussion of revised version of manuscript.

4) The Authors could comment on the issue of reliability of analyses done on just single slices off a block of tissue given concerns for the heterogeneity of sampling of a tumor mass (e.g., Gerlinger et al., NEJM 336; 883-892, 2012).

Our response: We totally agree with Reviewer #3 that TLS analyses on one single slide from the entire FFPE block might impact the reliability of data and of course the entire analyses would benefit from analysing more respective slides from individual patients. However, we believe that the potential risk of misinterpretation of our data are compensated by the facts that: 1) we analysed the TLS presence in large retrospective independent cohorts, counting 209 tumor samples; 2) for each individual patient, we stained two tissues slides: first using double CD20-DC-LAMP IHC staining and second using 6 plex immunofluorescence staining for CD4, CD8, CD20, CD21, CD23, DC-LAMP and GZMB to also confirm the number of TLS structures; 3) the TLS presence was analyzed in large tumor tissues from surgery which, as compared to routinely used biopsies within multiple studies cover large representative areas of at least 200 mm².

REVIEWER COMMENTS

Reviewer #1 (Remarks to the Author):

The authors have made significant improvements to this manuscript, however several concerns remain:

1. In the first sentence of the Abstract, "reflecting the key role of TLSs in the initiation of tumor targeting immunity" is an overstatement. This hypothesis is widely discussed in the field, and remains an important issue, but has not been definitively established. The same concern applies to Lines 87-88 and 475.
2. As previously stated by this reviewer, the term "early TLS" is still too loosely defined. According to the Methods section, it seems that early TLS were defined as any lymphoid aggregate of 50 or more cells that lacked CD21+ and CD23+ positivity. How close together did the cells need to be? What minimal cell type compositions were required? Moreover, in accordance with current practice in the field, primary vs. secondary follicles should not be grouped together as mTLS but treated as separate categories; others have shown that only the latter have prognostic significance, so this should be considered here as well.
3. The use of a DC vaccine in one group (study group 2) has been obscured by now simply referring to NCT02107937 study (Line 152), which is not helpful to readers.
4. 19 NSCLC cases is an insufficient sample size to make the potentially important conclusion that TLS are more numerous and mature in NSCLC vs. HGSOC. Moreover, the 19 cases remain incompletely described. Which types of NSCLC, grade, smoking status, etc. One cohort did not have NACT, the other may have? Supplementary Fig 1 is a schematic diagram, not a proper description of clinical cohorts.
5. Lines 437-438: they state that TLS in NSCLC are "more developed" than in HGSOC, but it's unclear how this was defined. Fig 3E refers to some sample images but leaves the reader to do their own comparisons.
6. On Pg. 3 of the rebuttal, they state that "Other histological subtypes were removed for the final survival analyses". Were they also removed from all other analyses? They should be, given the focus of this study on HGSOC.
7. Lines 320-325: How many mTLS's were evaluated in support of these statements? With only 3 cases, it seems likely that this number was small, especially given their statement that

TLS's were "at various maturation stages".

8. Line 331: do they mean lymphoid aggregate (which is not defined) or TLS?

9. There seems to be a contradiction between Lines 346-348 "improved disease outcome is largely associated with immune cell infiltration in non-TLS tumor areas" versus Lines 349-350 "mTLS formation is associated with effector CD8+ T cells and development of antitumor immunity", as well as the data presented in that section (e.g. Lines 383-385). Overall, this study still leaves one confused as to whether mTLS are favorable/unfavorable/neutral with respect to prognosis. The authors promote the idea that mTLS are unfavorable, but this seems contradicted by some of their data (Lines 384-385). It may be that a tie breaker study using an independent cohort is needed to resolve these contradictions.

10. Lines 442-444 seem to have another contradiction. If mTLS in HGSOc are associated with a higher density of TCF1+PD1+CD8+ T cells, which the authors argue are prognostically favorable, then shouldn't mTLS also be favorable? This contradiction is re-stated in Lines 446-448. I realize that biology is not always "rational", but such contradictions add to my confusion about the take-home messages from this study.

11. Line 507: one cannot claim a "positive correlation" if it is not significant.

12. Line 520: "direct link" is an overstatement given the correlative nature of the data.

Reviewer #2 (Remarks to the Author):

Dear Author,

Thank you my questions were addressed.

From a personal perspective, I am still a bit curious about the PNA⁺ status of the murine tumors as they may represent TLS vs lymphoid aggregates, and murine models of TLS are currently limited. However, you have done a large body of work, and enough to show that the murine lymphoid aggregates resemble the human TLS in many ways, and I will not belabor this point, as in this manuscript you have carefully called them LA and not TLS. From the perspective of the field having murine TLS models that resemble human TLS will be important to move TLS studies ahead.

Reviewer #4 (Remarks to the Author):

The Authors have adequately addressed my comments in their revised manuscript.

POINT-BY-POINT REPLY TO REVIEWER N° 1

Reviewer n° 1 commented:

The authors have made significant improvements to this manuscript, however several concerns remain:

Our response: We are indebted to Reviewer #1 for acknowledging the major improvements that were implemented into the revised version of the manuscript. This seems perfectly aligned with the evaluation of the other Reviewers, who were fully satisfied with the latest version of the article.

And raised these major critiques:

1. In the first sentence of the Abstract, "reflecting the key role of TLSs in the initiation of tumor targeting immunity" is an overstatement. This hypothesis is widely discussed in the field, and remains an important issue, but has not been definitively established. The same concern applies to Lines 87-88 and 475.

Our response: Inspired by the comment provided by Reviewer #1, we have toned down our statements regarding the key role of TLSs in initiation of tumor targeting immunity.

2. As previously stated by this reviewer, the term "early TLS" is still too loosely defined. According to the Methods section, it seems that early TLS were defined as any lymphoid aggregate of 50 or more cells that lacked CD21⁺ and CD23⁺ positivity. How close together did the cells need to be? What minimal cell type compositions were required? Moreover, in accordance with current practice in the field, primary vs. secondary follicles should not be grouped together as mTLS but treated as separate categories; others have shown that only the latter have prognostic significance, so this should be considered here as well.

Our response: We thank Reviewer #1 for bringing this point to our attention. As the field currently lack unique marker for eTLS identification, eTLS within our study were defined as lymphoid aggregates of minimum size of 250 μm , consist of 50 or more cells that lacked CD21⁺ and CD23⁺ positivity, with majority of cells being CD20⁺ B cells in close proximity contacts (min 3 μm) forming early clusters, with presence of CD4⁺ and CD8⁺ T cells. Inspired by the comment of Reviewer #1 we have provided further details on early TLS (eTLS) identification within material/methods of revised version of manuscript.

In addition, using multiplex immunofluorescence analyses (CD4, CD8, CD20, CD21, CD23, DCLAMP, GZMG) we have determined the frequency of CD4⁺CD8⁺CD20⁺CD21⁺CD23⁻ pTLSs and CD4⁺CD8⁺CD20⁺CD21⁺CD23⁺ sTLSs in tumor samples of 209 HGSOC patients as shown in Figure 1A and B of this rebuttal letter. Interestingly, only limited number of TLSs develop solely primary follicle, with vast majority of TLSs developing directly secondary follicle (Fig. 1B of this rebuttal letter). Importantly, these findings are different in NSCLC. As per request of Reviewer #1 raised in Point 4 (see below), we have determined the eTLS, pTLS and sTLS in total 31 adenocarcinoma NSCLC without neo-adjuvant chemotherapy (Fig. 1C of this rebuttal letter). We have observed high prevalence of both pTLS and sTLS in NSCLC patients (Fig. 1C of this rebuttal letter). Importantly, in line with Reviewer #1 comment, using median stratification we demonstrate that high prevalence of sTLS significantly correlate with superior survival benefit in patients with NSCLC ($p=0.034$) (Fig. 1D of this rebuttal letter). On contrary, we failed to observe similar survival benefits for eTLS and pTLS development in NSCLC patients (Fig. 1D of this rebuttal letter). Taken together, our analyses, immunostaining and data confirm and expand previously published findings about the prognostic impact of sTLS in uniform cohort of stage III adenocarcinoma NSCLC patients however the same does not hold true in HGSOC as further discussed in response to the point 4 of Reviewer #1.

We hope that Reviewer #1 will agree with our decision to classify pTLS and sTLS in HGSOC as mature TLS (mTLS), as pTLS are limited in HGSOC, a nomenclature that has been consistently used in previous preclinical and clinical studies in the field (*Vanhersecke et al, Nat Cancer, 2021; Meylan et al, Immunity, 2022; Vanhersecke et al, Lab. Invest., 2023; Ruffin et al, Nat Commun, 2021*).

Figure 1. Distribution of primary (pTLS) and secondary follicle (sTLS) like TLSs in HGSOc and NSCLC.

(A) Representative image of immunofluorescence of CD4, CD8, CD20, CD21, CD23, DC-LAMP and GZMB staining (immunofluorescence panel 1) for CD4⁺CD8⁺CD20⁺CD21⁺CD23⁻ pTLSs and CD4⁺CD8⁺CD20⁺CD21⁺CD23⁺ sTLSs identification in HGSOc. (B, C) Distribution of early TLS (eTLS), pTLS and sTLS across 209 HGSOc patients (B) and 31 adenocarcinoma NSCLC patients (C). (D) Overall survival (OS) of 31 adenocarcinoma NSCLC patients based on median stratification of sTLS and/or presence of pTLS and eTLS. Survival curves were estimated by the Kaplan-Meier method, and differences between groups were evaluated using log-rank test. Number of patients at risk and *p* values are reported.

3. The use of a DC vaccine in one group (study group 2) has been obscured by now simply referring to NCT02107937 study (Line 152), which is not helpful to readers.

Our response: Based on previous Reviewer #1 concern, original findings commenting the impact of TLSs in patients with DC-based vaccination were removed from the final version of manuscript. In the current version of manuscript, solely standard of care patients (control arm, study group 2) from NCT021007937 with no DC-based therapy were employed as validation cohort for TLS quantification in tumor samples within our study. In summary, no study group reported in this manuscript received DC-based vaccination.

4. 19 NSCLC cases is an insufficient sample size to make the potentially important conclusion that TLS are more numerous and mature in NSCLC vs. HGSOE. Moreover, the 19 cases remain incompletely described. Which types of NSCLC, grade, smoking status, etc. One cohort did not have NACT, the other may have? Supplementary Fig 1 is a schematic diagram, not a proper description of clinical cohorts.

Our response: Inspired by constructive comment raised by Reviewer #1, we have extended the original analyses of 19 adenocarcinoma NSCLC patients with additional 12 tumor samples from the same cohort of stage III adenocarcinoma NSCLC who underwent primary surgery in the absence of neo-adjuvant chemotherapy between 2014 and 2022 at University Hospital Hradec Kralove. The main clinicopathological characteristics (including histology subtype, grade, smoking status) of patients were summarized in Supplementary Table 5 within the revised version of manuscript. Using multiplex immunofluorescence analyses (CD4, CD8, CD20, CD21, CD23, DCLAMP, GZMG) we have determined the frequency of both eTLSs, pTLSs and mTLSs in additional 12 NSCLC patients (similarly as previously reported for 19 NSCLC patients) (Fig. 1C of this rebuttal letter). The frequency of both eTLS and mTLS is significantly higher as compared to HGSOE patients (n=209) as shown in Figure 2 of this rebuttal letter and Figure 3F of revised version of manuscript.

Figure 2. Distribution of eTLS and mTLS within HGSOE (n=209) and NSCLC (n=31).

5. Lines 437-438: they state that TLS in NSCLC are "more developed" than in HGSOE, but it's unclear how this was defined. Fig 3E refers to some sample images but leaves the reader to do their own comparisons.

Our response: We agree with Reviewer #1 that based on provided data we cannot claim that TLSs in NSCLC as compared to HGSOE were more developed. We apologize for the strong phrasing which was removed from the revised version of manuscript.

6. On Pg. 3 of the rebuttal, they state that "Other histological subtypes were removed for the final survival analyses". Were they also removed from all other analyses? They should be, given the focus of this study on HGSOE.

Our response: We thank Reviewer #1 for bringing this note to our attention. Only HGSOE tumor samples were employed for biological and survival analyses provided in the revised version of manuscript.

7. Lines 320-325: How many mTLS's were evaluated in support of these statements? With only 3 cases, it seems likely that this number was small, especially given their statement that TLS's were "at various maturation stages".

Our response: We agree with Reviewer #1 that that spatial transcriptomics findings presented in Figure 1D, E were performed on relative small number of HGSOC tumor samples. However, these findings were later supported by multiplex immunofluorescence analyses on 68 HGSOC patients presented in Figure 1F of revised version of manuscript, and they confirmed/expanded previously published data by Meylan et al, Immunity, 2022.

8. Line 331: do they mean lymphoid aggregate (which is not defined) or TLS?

Our response: We apologize for the oversight, the sentence was corrected accordingly in the revised version of manuscript: “In line with transcriptomic data, CD20⁺ B cells, DC-LAMP⁺ DCs, CD8⁺ T cells and GZMB⁺CD8⁺ T cells were largely localized within individual TLS as compared to nTLS areas.”

9. There seems to be a contradiction between Lines 346-348 "improved disease outcome is largely associated with immune cell infiltration in non-TLS tumor areas" versus Lines 349-350 "mTLS formation is associated with effector CD8+ T cells and development of antitumor immunity", as well as the data presented in that section (e.g. Lines 383-385). Overall, this study still leaves one confused as to whether mTLS are favorable/unfavorable/neutral with respect to prognosis. The authors promote the idea that mTLS are unfavorable, but this seems contradicted by some of their data (Lines 384-385). It may be that a tie breaker study using an independent cohort is needed to resolve these contradictions.

Our response: We apologize for the potential source of confusion. Taken together, our findings demonstrate that the presence of TLSs within the TME correlate with increased immune infiltrate, including effector CD8⁺ T cells as comprehensively documented within Figure 2 of revised version of manuscript. Moreover, using digital pathology analyses we have determined the clinical relevance of immune components (counting for CD8⁺, GZMB⁺CD8⁺ T cells and CD20⁺ B cells) within outside TLS areas (as determined by digital pathology after TLS areas exclusion) as compared to inner TLS areas presented within Figure 1G, H of revised version of manuscript. These unique digital pathology analyses suggest that although TLSs represent a unique site in the TME of HGSOC exhibiting a dense accumulation of CD20⁺ B cells, T_{FH} cells and effector cells, improved disease outcome is largely with immune cell infiltration outside TLS tumor areas as demonstrated withing Figure 1F-H of revised version of manuscript.

Based on comprehensive survival analyses on 4 independent HGSOC cohorts (n=209 patients; additional cohort 4 was already added based on Reviewer #1 request), we propose that patients with overall high density of TLS (determined as cumulative number of eTLS and mTLSs) are associated with improved disease outcome (Figure 2G of revised version of manuscript). Moreover, HGSOC patients with only eTLS formation (cluster 2) are associated with improved disease outcome as compared to patients with no TLS formation (cluster 1) (Figure 2H of revised version of manuscript). Finally, we failed to observe similar clinical benefits in patients developing mature TLS (cluster 3) as shown in 209 HGSOC patients and independently in individual patients cohorts (Figure 2H, Supplemental Figure 6 of revised version of manuscript). The lack of clinical benefit of mTLS development in HGSOC patients might be associated with several factors including : a) limited number of mTLS within the entire TME of HGSOC patients as compared to NSCLC (Fig. 1 B, C of this rebuttal letter; Figure 1B of revised version of manuscript); b) limited number of patients (15%) developing mTLS within 4 independent cohorts; c) localization of mTLS dominantly in invasive margins of TME (Suppl. Figure 4A, B of revised version of manuscript); d) higher density of terminally exhausted TIM3⁺PD1⁺CD8⁺ T cells as compared to progenitor TCF1⁺PD1⁺CD8⁺ T cells associated with mTLS formation within this particular group of patients (cluster 3) (Figure 3D of revised version of manuscript). We have done our best to convey this message as clearly as possible in the revised version of the manuscript.

10. Lines 442-444 seem to have another contradiction. If mTLS in HGSOC are associated with a higher density of TCF1+PD1+CD8+ T cells, which the authors argue are prognostically favorable, then shouldn't mTLS also be favorable? This contradiction is re-stated in Lines 446-448. I realize that biology is not always "rational", but such contradictions add to my confusion about the take-home messages from this study.

Our response: We thank Reviewer #1 for bringing this to our attention. Our findings demonstrate that frequency of mTLSs correlate with higher density of TCF1⁺PD1⁺CD8⁺ T cells which is clearly prominent in NSCLC patients with higher mTLS frequency as compared to HGSOC patients associated with limited number of mTLS (Figure. 3G, J of revised version of manuscript). Supporting this notion, the median density of TCF1⁺PD1⁺CD8⁺ T cells within NSCLC was significantly higher in the entire TME ($p=0.0002$) (as well as eTLS and mTLS) as compared to HGSOC (Figure 3G of revised version of manuscript). Moreover, not only mTLS but also eTLS formation correlate with higher density of TCF1⁺PD1⁺CD8⁺ T within tumor core and tumor stroma of patients with only

eTLS development (cluster 2) as shown in figure 3D, which might also impact the survival benefit of this immune subset demonstrated by Kaplan-Meier analyses.

11. Line 507: one cannot claim a "positive correlation" if it is not significant.

Our response: We agree with Reviewer #1 and apologise for strong phrasing, which we have toned down.

12. Line 520: "direct link" is an overstatement given the correlative nature of the data.

Our response: We agree with Reviewer #1 and apologise for overstatement, which we have toned down.

POINT-BY-POINT REPLY TO REVIEWER N° 2

Reviewer n° 2 commented:

Thank you my questions were addressed.

POINT-BY-POINT REPLY TO REVIEWER N° 3

Reviewer n° 3 commented:

The Authors have adequately addressed my comments in their revised manuscript.

Our response: We are indebted to Reviewer #2 and #3 of Nature Communications for the positive evaluation of our revised version of manuscript.

REVIEWERS' COMMENTS

Reviewer #1 (Remarks to the Author):

The authors have made another round of significant revisions in response to reviewers' comments, and the manuscript is significantly improved. However, I am still concerned about the sample sizes of several of the analyses, and the fact that sample sizes seem to jump around depending on the analysis, in ways that are difficult to understand or follow.

1. Results in Figure 4B appear based on only 17 HGSOC and 10 NSCLC cases. In Fig 4C the numbers change to 16 and 14, respectively, with no explanation given. If these findings are considered important, they should be demonstrated with larger, uniform cohorts.
2. Throughout the manuscript, the authors emphasize their assertion that an increased density of TCF1+PD1+CD8+ T cells was associated with favorable prognosis. However, the Kaplan-Meier analysis in Suppl. Fig. 7E appears to be based on only 20 patients. If so, this is a grossly inadequate sample size.
3. In response to the request for an increased number of NSCLC cases, the authors added 12 cases to reach a total of 31. This is still a low number on which to base their conclusions. Moreover, they seem to have increased the number of HGSOC patients to n=209 (which is a favorable increase), yet this is not mentioned in the rebuttal letter. Is this the case, or is this a mistake?
4. For some cohorts, it remains unclear whether some patients had NACT. If so, such cases need to be analyzed and reported separately, rather than pooled with the others. If none of the cases in the entire study had NACT, this needs to be stated unequivocally for the benefit of readers.
5. There is no table for Study Group 7.
6. The titles for Supplemental Tables 1 and 2 are confusing:

"Supplemental Table 1. The main clinicopathological characteristics of 209 HGSOC patients from study group 1 (cohort 1 and 2) and group 2 employed within survival analyses."

"Supplemental Table 2. The main clinicopathological characteristics of 79 HGSOC patients from study from study group 2."

Does this mean that patients from Study Group 2 are reported in both tables?

Overall, I still have concerns about sample sizes and the ways in which cohorts have been assembled, analyzed and reported. Some findings appear well supported by large sample sizes, whereas others involve unacceptably low numbers. Yet one has to read the manuscript carefully to detect such issues. One is left uncertain as to which findings and conclusions are likely to be robust, and which are not.

POINT-BY-POINT REPLY TO REVIEWER N° 1

The authors have made another round of significant revisions in response to reviewers' comments, and the manuscript is significantly improved. However, I am still concerned about the sample sizes of several of the analyses, and the fact that sample sizes seem to jump around depending on the analysis, in ways that are difficult to understand or follow.

Our response : We thank Reviewer #1 for acknowledging the improvements we implemented into the revised version of the paper. Reviewer #1 is right in noticing that sample sizes are not constant throughout the paper, which largely emerges from the limited amount of material available from patients and/or the technological approaches. As a standalone example, we have used sequential IHC staining for some of the analysis. In this setting, the last rounds of staining are somehow prone for the tissue to detach and hence get lost, impeding the analysis of a specific marker, but not previous ones from the same sample.

1. Results in Figure 4B appear based on only 17 HGSOE and 10 NSCLC cases. In Fig 4C the numbers change to 16 and 14, respectively, with no explanation given. If these findings are considered important, they should be demonstrated with larger, uniform cohorts.

Our response: Please see point above for the discrepancy in number of samples, the technologies employed being described in detail in the M&M section. These findings, which were generated in response to previous requests from the reviewers of Nature Communication, lend additional support to conclusions that are based on the bulk of the data from several patient cohorts presented throughout the paper.

2. Throughout the manuscript, the authors emphasize their assertion that an increased density of TCF1+PD1+CD8+ T cells was associated with favorable prognosis. However, the Kaplan-Meier analysis in Suppl. Fig. 7E appears to be based on only 20 patients. If so, this is a grossly inadequate sample size.

Our response: We thank Reviewer #1 for bringing up this point. We indeed mentioned that an “an increased density of TCF1+PD1+CD8+ T cells was associated with favorable prognosis as determined by Kaplan-Meier analyses”, which to us does not seem over emphasized, while citing in the discussion several studies linking this cell population with improved patient prognosis in a variety of tumors. That said, our findings from Suppl Fig 6E

are 100% disposable for the central message of the paper and can be removed if the Editors of Nature Communications deem this necessary.

3. In response to the request for an increased number of NSCLC cases, the authors added 12 cases to reach a total of 31. This is still a low number on which to base their conclusions. Moreover, they seem to have increased the number of HGSOC patients to n=209 (which is a favorable increase), yet this is not mentioned in the rebuttal letter. Is this the case, or is this a mistake?

Our response: Indeed, the total number of HGSOC patients is 209, we apologize if this was not clear from the rebuttal letter. As for NSCLC, we respectfully disagree that a total of 31 is a low number in this specific context for at least the two following reasons: (1) this is a very homogeneous cohort of stage 3 adenocarcinoma patients that was gathered rapidly in response to the Reviewers' request, and (2) this cohort is used as a comparator for the main findings of the papers, which are instead based on >200 HGSOC patients, with statistically significant results

4. For some cohorts, it remains unclear whether some patients had NACT. If so, such cases need to be analyzed and reported separately, rather than pooled with the others. If none of the cases in the entire study had NACT, this needs to be stated unequivocally for the benefit of readers.

Our response: We apologize for this oversight, no patients received NACT. We have revised the description of the cohorts to make sure this is clear within revised version of manuscript.

5. There is no table for Study Group 7.

Our response : Inspired by Reviewer #1 comment we have added supplemental table for study group 7 in the revised version of manuscript.

6. The titles for Supplemental Tables 1 and 2 are confusing:

"Supplemental Table 1. The main clinicopathological characteristics of 209 HGSOC patients from study group 1 (cohort 1 and 2) and group 2 employed within survival analyses."

"Supplemental Table 2. The main clinicopathological characteristics of 79 HGSOC patients from study from

study group 2."

Does this mean that patients from Study Group 2 are reported in both tables?

Our response: We apologize for the confusion, we have revised manuscript and supplemental tables to improve clarity on specific patients cohorts and groups within the entire study.

Overall, I still have concerns about sample sizes and the ways in which cohorts have been assembled, analyzed and reported. Some findings appear well supported by large sample sizes, whereas others involve unacceptably low numbers. Yet one has to read the manuscript carefully to detect such issues. One is left uncertain as to which findings and conclusions are likely to be robust, and which are not.

Our response: We hope that the explanations provided above and related corrective measures are sufficient for this paper to be accepted for publication in Nature Communications, as recommended previously by two addition Reviewers